# Left–right-alternating theta sweeps in entorhinal–hippocampal maps of space

Abraham Z. Vollan[1,2 ✉], Richard J. Gardner[1,2], May-Britt Moser[1] & Edvard I. Moser[1 ✉]

Place cells in the hippocampus and grid cells in the entorhinal cortex are elements of a neural map of self position[1–5]. For these cells to benefit navigation, their representation must be dynamically related to the surrounding locations[2]. A candidate mechanism for linking places along an animal's path has been described for place cells, in which the sequence of spikes in each cycle of the hippocampal theta oscillation encodes a trajectory from the animal's current location towards upcoming locations[6–8]. In mazes that bifurcate, such trajectories alternately traverse the two upcoming arms when the animal approaches the choice point[9,10], raising the possibility that the trajectories express available forward paths encoded on previous trials[10]. However, to bridge the animal's path with the wider environment, beyond places previously or subsequently visited, an experience-independent spatial sampling mechanism might be required. Here we show in freely moving rats that in individual theta cycles, ensembles of grid cells and place cells encode a position signal that sweeps linearly outwards from the animal's location into the ambient environment, with sweep direction alternating stereotypically between left and right across successive theta cycles. These sweeps are accompanied by, and aligned with, a similarly alternating directional signal in a discrete population of parasubiculum cells that have putative connections to grid cells via conjunctive grid × direction cells. Sweeps extend into never-visited locations that are inaccessible to the animal. Sweeps persist during REM sleep. The sweep directions can be explained by an algorithm that maximizes the cumulative coverage of the surrounding manifold space. The sustained and unconditional expression of theta-patterned left–right-alternating sweeps in the entorhinal–hippocampal positioning system provides an efficient 'look around' mechanism for sampling locations beyond the travelled path.

Grid cells are position-tuned cells with firing locations that rigidly tile the environment in a hexagonal lattice pattern[4,11,12]. Two decades of investigation have proposed a mechanism for grid cells in which their firing emerges when an activity bump is translated across an internally generated toroidal continuous attractor manifold, on the basis of external speed and direction inputs[5,6,13–19]. However, the role of grid cells in navigation remains poorly understood. One clue is that the spatial coordinate system defined by grid cells allows position offsets to be expressed as vectors[11,13–15]. Not only does this make it possible to continuously update the neural position representation by path integration, but it also allows the network to relate the current position estimate to target locations such as goals[20–24]. Grid cells have been suggested in computational models to probe the environment dynamically by using linear 'look-ahead' trajectories[21–25]. These proposed trajectories are similar to the theta-paced forward sweeps recorded in hippocampal place-cell ensembles when animals run on linear tracks or in mazes[6–10]. The sweeps observed on tracks and in mazes in previous studies have limited navigational utility, however, because they are constrained to the travelled path. Here, by recording from the medial entorhinal cortex

(MEC) and hippocampus using high-site-count Neuropixels probes[26,27], we searched for a more general mechanism for rapid sampling of the ambient environment, including places the rat had never visited, in the population activity of many hundreds of grid, place and direction cells.

## Grid and place cells sample space with alternating sweeps

We recorded neural activity in 16 rats by using Neuropixels probes targeting MEC and parasubiculum (384–1,522 cells per session; Extended Data Fig. 1) while the rats foraged for scattered food in an open-field arena of 1.5 m × 1.5 m. As expected, the activity was patterned by the theta rhythm (Fig. 1a, top), which discretizes population activity in MEC and hippocampus into successive packets of around 125 ms (refs. 28–30). To examine the dynamics of spatial coding in individual cycles of the rhythm, we decoded position in 10-ms bins by correlating the instantaneous firing-rate population vectors (PVs) from all MEC–parasubiculum cells with the session-averaged PV at each position in the environment (that is, the stack of firing-rate maps). Over

[1]Kavli Institute for Systems Neuroscience and Centre for Algorithms in the Cortex, Norwegian University of Science and Technology, Trondheim, Norway. [2]These authors contributed equally: Abraham Z. Vollan, Richard J. Gardner. ✉e-mail: abraham.z.vollan@ntnu.no; edvard.moser@ntnu.no

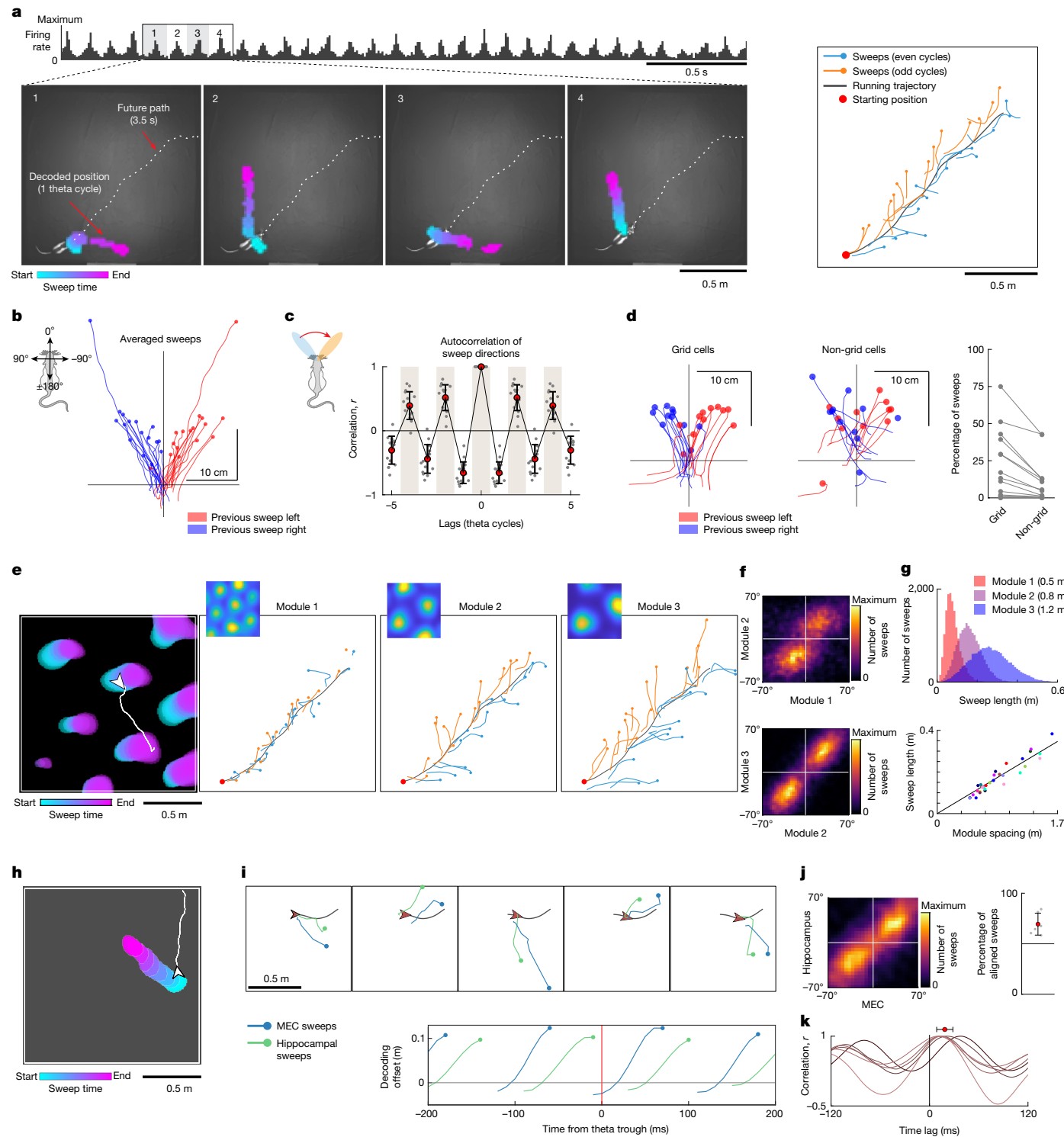

**Fig. 1** | See next page for caption.

the course of each theta cycle, the decoded position generally swept in a straight trajectory outwards from the animal's location into the nearby environment (Fig. 1a and Supplementary Video 1). In general, such trajectories, referred to as sweeps, moved forwards at an angle from the animal's head axis, with direction alternating between left and right on successive theta cycles. Left–right alternating sweeps were particularly prominent during fast, straight running (Fig. 1a). The sweeps were not coupled to rhythmic lateral or vertical head movements, or to the frequency of footsteps, although these behaviours

occurred at frequencies similar to the theta rhythm (Extended Data Fig. 2).

Sweeps were counted and measured by a sequence-detection procedure that looked for near-linear trajectories of decoded positions spanning at least four successive time bins of the theta cycle, with analysis restricted to locomotion periods (more than 15 cm s$^{-1}$). Sweeps were detected in all 16 animals with recording sites in the MEC or parasubiculum (Fig. 1b). The number of identified sweeps increased linearly with the number of recorded cells and the number of spikes

**Fig. 1 | Grid cells and place cells sample ambient space with alternating sweeps. a**, Left–right-alternating theta sweeps in ensembles of MEC and parasubiculum cells. Top, summed spike counts of all co-recorded cells during a 3.5 s epoch, showing 8–10 Hz theta-rhythmic population activity (rat 25843, session 1). Bottom, sweeps decoded from joint activity of these cells during the four successive theta cycles highlighted above. Images show snapshots of the recording arena; white dashed lines show the animal's future trajectory. The decoded position throughout each theta cycle is plotted as coloured blobs (position bins in which correlation between instantaneous PVs and reference maps exceed the 99th percentile), with colour indicating time within the sweep. Right, decoded sweeps during the whole 3.5 s period (one sweep per theta cycle). Colours correspond to even (blue) and odd (orange) theta cycles. **b**, Sweeps rotated to head-centred coordinates (head orientation is vertical) and averaged across theta cycles in which the preceding sweep went left (red) or right (blue); data are for 16 animals, with one pair of red and blue lines per animal. Note the consistent alternation of sweep direction (no red sweeps on the left and no blue sweeps on the right). **c**, Temporal autocorrelograms of angles between successive sweep directions, for discrete theta cycles, across all recording sessions (individual animals, $n = 16$, grey dots; mean and s.d., red dots with whiskers). Note that there are peaks at alternate theta cycles. **d**, Sweeps decoded separately from grid cells or a size-matched random selection of non-grid cells (within the same recording region, the same session and the same animal). Left, plots like those in **b**. Note the more-uniform alternations and directions of sweeps in grid cells. Right, fractions of theta cycles with detected sweeps. Dots connected by lines show data from the same session in the same animal ($n = 16$). **e**, Grid modules express sweeps at different scales. Left, coloured blobs plotted for a single grid module, as in **a**. Right, sweeps of

three co-recorded grid modules plotted on top of the running trajectory (grey) during a 3.5 s period, as in **a**; blue and orange indicate even and odd theta cycles. Note the coordinated left–right alternation and increasing sweep length. **f**, Heat maps showing the joint distribution of sweep directions (head-centred) for pairs of simultaneously recorded grid modules during the recording session shown in **e**. **g**, Sweep length is proportional to grid spacing. Top, histograms showing the distribution of sweep lengths in the three modules shown in **e** (module spacing is indicated). Bottom, mean sweep lengths for all modules (one session per animal; each dot shows one module; colour indicates animal identity). **h**, Example sweep decoded from hippocampal ensemble activity (all cells) in an open field arena (as in **a** and **e**). **i**, Hippocampal sweeps follow MEC–parasubiculum sweeps. Top, decoded sweeps from co-recorded hippocampal cells (green) and MEC–parasubiculum cells (blue) over five successive theta cycles. Current location (arrowhead) and future path (grey line) are shown. Note the alignment of hippocampal and MEC–parasubiculum sweeps. Bottom, progression of decoded sweeps along the head direction axis as a function of time from the beginning of the theta cycle (for an example recording session). Note that hippocampal sweeps are delayed relative to MEC sweeps. **j**, Left, heat map showing alignment of sweep directions decoded from co-recorded MEC and hippocampal cells, plotted as in **f**. Right, fraction of theta cycles in which sweeps from both regions pointed to same side of the head axis (all six animals with paired hippocampus/MEC–parasubiculum recordings; red dot, mean; whiskers, s.d.). **k**, Temporal cross-correlation of decoded positions in MEC and hippocampus (one session per animal; red dot, mean peak location; whiskers, s.d.). Note the peak correlation at positive lags, indicating that sweeps in the hippocampus are delayed. Credit: rat, scidraw.io/Gil Costa.

recorded per theta cycle (Extended Data Fig. 3a). Sweeps were detected in 72.9 ± 5.8% (mean ± s.e.m.) of the theta cycles in recordings with more than 1,000 cells (3 rats). In the full sample (16 rats, mean of 769 cells), sweeps were detected in 48.0 ± 5.2% of theta cycles. Sweep directions alternated from side to side across theta cycles in all rats (Fig. 1b,c and Extended Data Fig. 3c,d), with left–right–left or right–left–right alternation occurring in 79.8 ± 2.24% (mean ± s.e.m.) of successive sweep triplets, significantly more often than when sweep directions were shuffled (61.1 ± 0.2%; higher than the 99.9th percentile in all animals). Alternation was also significant with wider windows (more than three theta cycles; Extended Data Fig. 3i). Sweeps were directed forwards at an angle of 23.9 ± 2.7° (mean ± s.e.m. across 16 animals) to either side of the animal's head direction (Fig. 1b and Extended Data Fig. 3c,d), approaching around 30° to either side in the animals with most cells (Extended Data Fig. 3d). The average length of a sweep was 22.5 ± 1.4 cm (mean ± s.e.m.). Left–right alternating sweeps were preserved when other decoding methods were used (Extended Data Fig. 4a and Supplementary Video 1).

After observing sweeps in the combined activity of all MEC–parasubiculum cells, we next investigated whether sweeps were preferentially expressed in spatially modulated neurons such as grid cells (24.8% of the recorded cells; range 8–42% across animals). Restricting the analysis to grid cells revealed a stronger presence of sweeps than for non-grid cells: 28.5 ± 6.7% of theta cycles in grid cells compared with only 13.4 ± 4.6% in a similar number of randomly sampled, simultaneously recorded non-grid cells ($P = 0.001$, Wilcoxon signed-rank test with 11 rats and a criterion of 100 cells or more for decoding; Fig. 1d). In two rats with more than 300 grid cells, the corresponding numbers were 63.1 ± 11.9% and 42.7 ± 8.4%. The contribution of subtypes of grid cells[5,31–33] was assessed by omitting either burst-firing (bursty) grid cells or non-bursty grid cells from the decoder (Extended Data Fig. 5). The number of detected sweeps was decreased when bursty grid cells (located in MEC layer II or the parasubiculum) were excluded from the decoder, but not when non-bursty grid cells (located in MEC layer III) were excluded (Extended Data Fig. 5c). Individual bursty grid cells exhibited phase precession along the sweep axis and, to a weaker extent, along the axis of movement (Extended Data Fig. 5h–k). Non-bursty

grid cells were not affected by the choice of reference position. These findings point to burst-firing grid cells as the strongest carriers of the entorhinal sweep signal.

Grid cells are organized functionally in modules, each consisting of cells with common grid spacing and field size[16,34]. To determine whether sweeps are coordinated across modules, we identified 36 grid modules, with 1–4 modules per animal and 13–215 cells per module, and decoded position separately for each module with more than 40 grid cells (31 modules in 15 animals). Sweeps in individual grid modules were directed forwards at an angle of 19.3 ± 3.5° (mean ± s.e.m.; alternation in 74.4 ± 1.9% of theta cycle triplets, greater than the 99.9th percentile of shuffled values in 28 of 31 modules) to either side of the head axis (Fig. 1e and Extended Data Figs. 3e and 4b). Sweep directions of co-recorded grid modules were mutually aligned (circular correlation between sweep directions, $r = 0.60 ± 0.038$, $P < 0.001$ for all 23 module pairs), pointing to the same side of the head axis in 70.3 ± 2.0% of theta cycles (significantly above the 50% chance level for all module pairs, $P < 0.001$, binomial test) (Fig. 1f). Sweep lengths scaled with the spacing of the grid modules (Fig. 1e and Supplementary Video 2; Pearson correlation between grid spacing and sweep length, $r = 0.95$, $P = 2.9 × 10^{-16}$, $n = 31$ modules from 15 animals) (Fig. 1g), spanning 19.7 ± 0.5% of the module spacing and rarely exceeding half of the spacing (8.8 ± 1.8% (mean ± s.e.m.) of sweeps; mean sweep lengths from 0.07 to 0.38 m and module spacing from 0.46 m to 1.62 m). This proportional relationship makes sweep lengths approximately equal across modules when mapped onto the toroidal unit tile of the modules.

To investigate whether sweeps are propagated from the MEC to downstream regions, we decoded position from place cells in the hippocampus of eight animals (six of which also had MEC–parasubiculum implants). Place cells mirrored grid cells by expressing sweeps forwards to alternating sides of the animal (Fig. 1h,i and Extended Data Fig. 3f). Sweeps were detected in 43.7 ± 5.9% (range 23.9–69.4%) of identified theta cycles when position was decoded from the joint activity of all hippocampal cells (157–747 cells). Identified sweeps had average lengths of 0.27 ± 0.03 m (mean ± s.e.m.) and offsets of 20.8 ± 5.1° from the head axis. Simultaneously decoded hippocampal and entorhinal sweeps were co-aligned (left or right with respect to the head axis) in

70.2 ± 4.1% of the theta cycles in which sweeps were detected in both brain regions (range 60.7–84.3%, $P < 0.001$ compared with chance in all animals, one-tailed binomial test). The absolute mean angle between sweep directions in the two cell populations was only 5.5 ± 1.9° (correlation, $r = 0.46 ± 0.089$, $n = 6$ rats; Fig. 1i,j). Place-cell sweeps were delayed compared with MEC sweeps (temporal cross-correlation of decoded positions, lag of 19.4 ± 3.9 ms (mean ± s.e.m.), 6 rats, all cells in each region) (Fig. 1k and Extended Data Fig. 3g), raising the possibility that hippocampal place cell sweeps are inherited from entorhinal grid cells[24,35].

## Alternating direction signal in a separate cell population

The shared direction of sweeps across grid modules implies an overarching coordination mechanism, such as a common directional input signal[11,13]. To search for a direction signal that matches the direction of sweeps, we examined head-direction cells in the MEC–parasubiculum[36–38]. A total of 29.5% (3,632 of 12,300) of the cells displayed significant and stable head-direction tuning (16 animals, 1 session per animal; Fig. 2a and Extended Data Fig. 6a). As in previous studies[37,38], head-direction cells in this region were often more broadly tuned than their counterparts in the anterior thalamus[39] or presubiculum[36] (tuning width, 111.7 ± 23.9° (mean ± s.d.); Fig. 2a). Most of the head-direction-tuned cells in our sample, including conjunctive grid × direction cells (referred to as 'conjunctive grid cells' hereafter), were strongly modulated by the local theta rhythm (Fig. 2a and Extended Data Fig. 6a). These cells were anatomically segregated from non-conjunctive ('pure') grid cells, with most theta-rhythmic directional cells located in the parasubiculum (85.6% (1,699 of 1,984) in 14 rats) (Fig. 2b and Extended Data Fig. 1a,b).

At the population level, there was strong correspondence between sweep direction in grid cells and the direction encoded by the theta-modulated head-direction cells. Although the population activity of the latter cells roughly tracked the animal's head direction (Fig. 2c, top), sub-second analyses revealed discrete, theta-paced packets of coordinated activity that flickered from left to right of the head axis on successive theta cycles (Fig. 2c, bottom). To quantify these switches, we decoded instantaneous direction by correlating firing-rate PVs at the centre of each theta cycle with the session-averaged PVs for each head direction. The decoded signal, referred to as internal direction, alternated from side to side of the head axis in 86.1 ± 1.8% of theta cycle triplets (higher than the 99.9th percentile of shuffled values in all 16 animals) (Fig. 2c–e and Extended Data Fig. 4c,d), with peak offsets at 19.9 ± 2.9° on either side of the head axis (27.9 ± 1.6° in the 3 animals with most cells). Sweeps in grid cells were aligned with decoded internal direction (circular correlation, $r = 0.66 ± 0.04$, $P < 0.01$ in all animals; absolute mean angle 4.4 ± 0.9° (mean ± s.e.m.)) (Fig. 2f,g and Supplementary Video 3). Sweep and direction signals pointed to the same side of the head axis in 72.5 ± 2.3% of theta cycles ($P < 0.01$ compared with chance in all animals, one-tailed binomial test). The rigid left–right alternation of the population signals explains the phenomenon of theta-cycle skipping[10,40–42] in individual cells (predominant firing on every other theta cycle), which was present in most of the internal direction and grid cells (Fig. 2a and Extended Data Figs. 3b, 5l,m and 6b,g–i).

To determine whether cells that participate in the alternating direction signal are distinct from classical head-direction cells, we compared the relative tuning strength of each cell with tracked head direction versus decoded (internal) direction (Extended Data Fig. 6c). Theta-rhythmic directional cells, including conjunctive grid cells, were more strongly tuned to internal direction, whereas the smaller sample of non-rhythmic directional cells, often in deep layers or neighbouring regions, consistently followed the tracked head direction (classical head-direction cells; Extended Data Fig. 6c–e). The coupling between

sweeps and internal direction persisted when conjunctive grid cells were excluded from the analysis (Extended Data Fig. 6f).

## A microcircuit for directing sweeps

The correlations between internal direction and sweep signals, along with the strong projections from the parasubiculum to layer II of the MEC[43], point to internal direction cells as a possible determinant of sweep direction in grid cells. To examine whether this is reflected in the functional connectivity of the circuit, we cross-correlated the spike times of all recorded cell pairs (1,421,107 pairs, 16 animals; Fig. 3a). Pairs in which one cell consistently fired ahead of the other with short latency were defined as putatively connected[44,45]. Putative connections were found within and between functional cell classes (Extended Data Fig. 7a–d). They included connections from internal direction cells to conjunctive grid cells (144 of 130,506; 0.11% of cell pairs) and from conjunctive grid cells to pure grid cells (188 of 49,091; 0.38%). Internal direction cells and conjunctive grid cells had more-frequent functional connections to bursty grid cells (in MEC layer II; Extended Data Fig. 5b) than to non-bursty grid cells (in MEC layer III; Extended Data Fig. 5b) (420 of 131,173 (0.32%) versus 5 of 73,985 (0.007%) internal direction and conjunctive grid cells combined; $P = 3.0 × 10^{-73}$, Fisher's exact test) (Extended Data Fig. 7a).

In theoretical models, the grid-cell position signal is translated across the attractor manifold by directional input that causes a phase shift of grid-cell activity in the direction of the input signal, through a layer of conjunctive grid cells[11,13,15,25]. Consistent with this prediction, putative connections from internal direction cells to conjunctive grid cells were primarily between cells with similar directional tuning (Fig. 3b–d and Extended Data Fig. 7c,f; angle between tuning directions of 14.1 ± 52.1° (mean ± s.d.); circular correlation between tuning directions, $r = 0.41$, $P = 6 × 10^{-7}$, $n = 144$ connected cell pairs from 13 animals). Moreover, putative connections from conjunctive grid cells to pure grid cells targeted cells with a slightly shifted grid phase (Fig. 3e and Extended Data Fig. 7d; magnitude of grid phase offset, 26.0 ± 12.8% (mean ± s.d.) of the grid spacing; $n = 85$ connected cell pairs from 12 animals). The direction of the spatial phase offset closely matched the preferred internal direction of the conjunctive grid cells (Fig. 3e,f and Extended Data Fig. 7f; correlation between grid-phase offset direction and preferred direction, $r = 0.62$, $P < 5.1 × 10^{-8}$; angle between directions, 0.4 ± 55.3° (mean ± s.d.)). This directional and positional alignment of connected cell pairs differed substantially from that of randomly selected non-connected cell pairs (Extended Data Fig. 7g). Taken together, the findings support the notion that sweep direction is determined by activation of internal direction cells via a layer of conjunctive grid cells (Fig. 3d,g).

## Sweeps extend to never-visited locations

In place cells, forward-projecting sweeps before the choice point in a maze are thought to reflect a deliberation over behavioural options[9,10]. If sweeps are involved in planning, we would expect their direction to correlate with subsequent movement. The sweeps recorded during foraging in this study showed a high degree of stereotypy that is inconsistent with a role in goal-oriented navigation. However, because most of these sweeps corresponded to navigable and previously travelled paths, a role in trajectory planning is not ruled out. This limitation led us to record in environments in which the navigational opportunities were constrained to 1D paths. The rats ran on one of the following: an elevated 2.0 m linear track with reward delivered at the ends (5 rats; Fig. 4a, left); an elevated 'wagon wheel' track consisting of a circle, 1.5 m in diameter, with two diagonal cross-bridges[5] (2 rats; Fig. 4a, right); or an m-shaped extended T-maze with a left–right choice point at the end of the central stem (1.1 m x 1.2 m) (Extended Data Fig. 8). Directional signals were decoded from all MEC–parasubiculum cells as before,

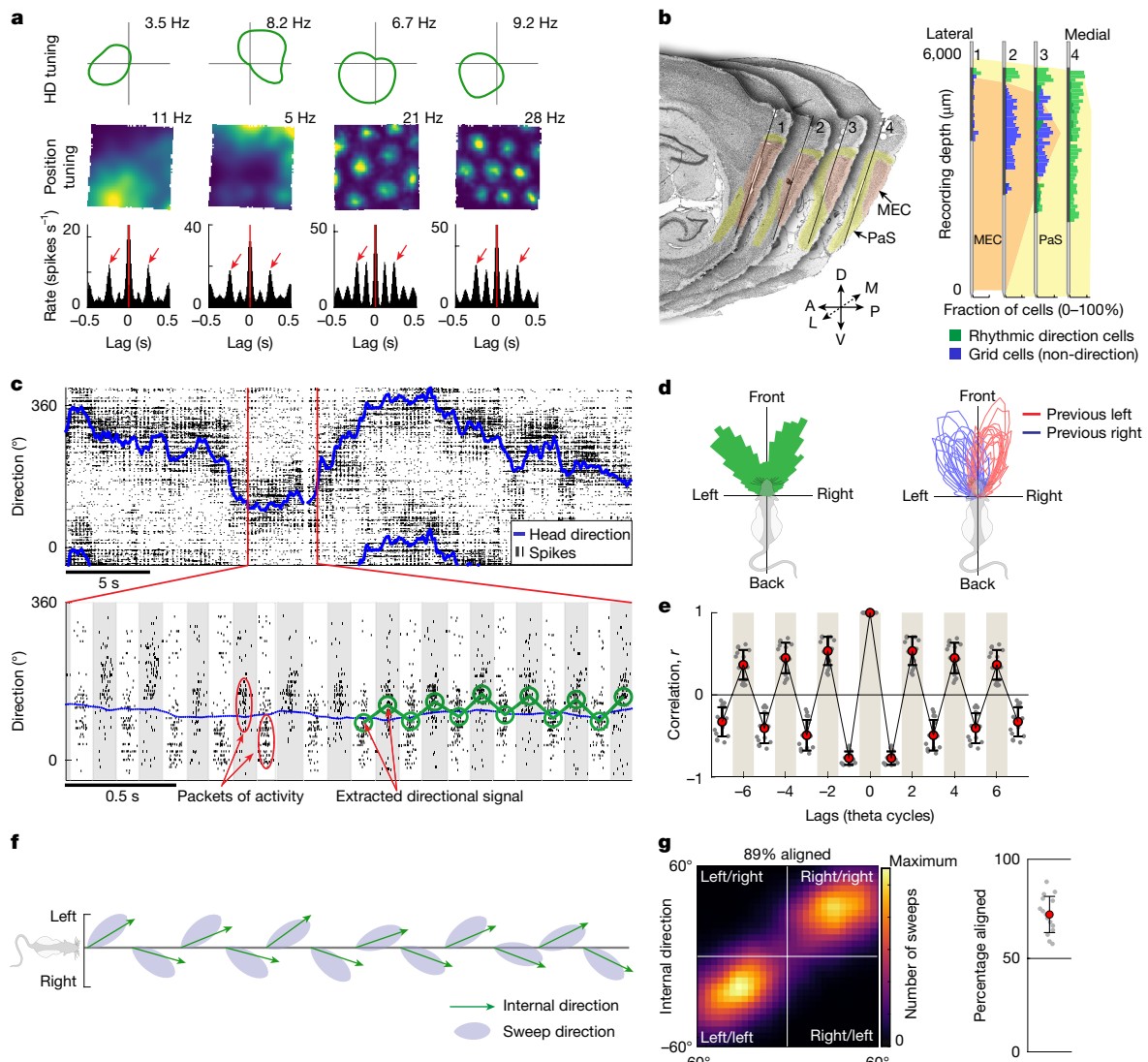

**Fig. 2 | Sweeps are aligned with an alternating direction signal in the parasubiculum. a**, Top, circular firing-rate plots showing broad tuning to head direction (HD) for four example cells in the parasubiculum. Middle, 2D position rate maps for the same cells. Bottom, temporal autocorrelograms of the cells' firing rates. Arrows indicate temporal lags corresponding to the second theta peak (about 250 ms). Note the strong theta-cycle skipping in the first two cells, and conjunctive grid tuning in the last two. **b**, Theta-rhythmic directional cells are localized primarily in the parasubiculum. Left, serial sagittal sections (lateral to medial) from rat 25953 showing tracks from a four-shank Neuropixels 2.0 probe in the parasubiculum (PaS, yellow) and MEC (orange). Each section contains the track from one shank (about 250 μm apart). Directions: L, lateral; M, medial; A, anterior; P, posterior; D, dorsal; V, ventral. Right, fraction of theta-rhythmic 'internal direction' cells (green) and pure grid cells (blue) along the probe shanks (proportion of all cells at corresponding depths). Black segments on the shanks show active recording sites (combining seven sessions with different site configurations). **c**, Raster plot showing flickering of direction-tuned activity around the animal's head direction on successive theta cycles. Top, head direction of rat 25843 during 30 s of running in an open arena. Black rasters indicate spikes fired by 533 co-recorded internal direction cells, mostly from the parasubiculum. Cells (rows) are sorted by preferred firing direction. Bottom, population activity for a 4 s extract from the top panel. Theta cycles are indicated by alternating grey and white backgrounds. Discrete, theta-paced 'packets' of population activity alternate left and right of the animal's head direction (blue line). Green circles show the decoded direction in each theta cycle. **d**, Left, offset between head direction and decoded direction plotted in a polar histogram (rat 25953, session 4). Decoded direction is distributed at two principal angles on either side of the head direction. Right, distributions of decoded direction (relative to head direction) for all 16 animals (1 session per animal). Each red or blue line shows the distribution of decoded directions when the previous decoded direction was oriented to the left (red) or right (blue). **e**, Temporal autocorrelograms of angles between decoded directions in successive theta cycles, for all 16 animals (1 open-field session per animal). Red dots, means; whiskers, s.d. **f**, Sweeps are aligned to the direction signal. Decoded direction (green arrows) and sweep direction (grey patches) are shown over 12 successive theta cycles (about 1.5 s, same session as in **c**) plotted with reference to the animal's head direction (horizontal line). Theta cycles are evenly spaced along the horizontal axis. **g**, Left, heat map showing alignment of decoded direction and sweep direction (both in head-centred coordinates) across theta cycles throughout the recording session in **c** and **f**. Right, percentage of theta cycles in which decoded direction and sweeps point to the same side of the animal's head axis (16 rats, 1 open-field session per animal). Red dot, mean; whiskers, s.d. Credit: rat, scidraw.io/Gil Costa.

by correlating instantaneous PVs with head-direction tuning curves computed over the whole session, which was long enough for all directions to be sampled. In all tasks, the decoded direction signal pointed consistently to the sides of the tracks, towards places never navigated,

in an alternating pattern resembling that observed in the open arena (Fig. 4a and Extended Data Fig. 9a). As before, alternating direction signals were accompanied by sweeps in grid cells, decoded from the phase relationships of these cells in a previous open-field session.

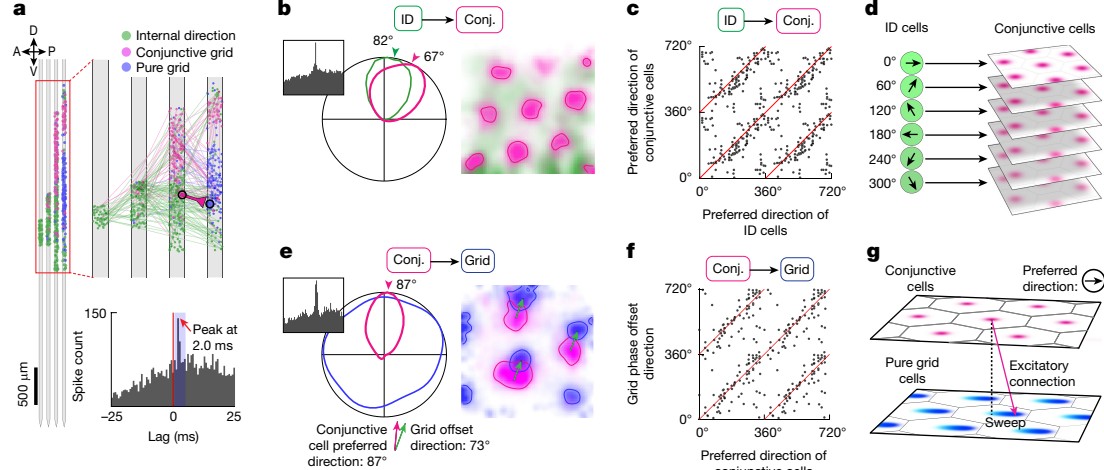

**Fig. 3 | A microcircuit for directing sweeps. a**, Identification of putative monosynaptic connections. Top, dots show locations of co-recorded internal direction cells (green), conjunctive grid cells (pink) and pure grid cells (blue) along each recording shank during an example session. Lines show detected connections between pairs of cells; the colour indicates the functional identity of the presynaptic cell. One example pair of connected cells (conjunctive grid to pure grid) is highlighted. Bottom, cross-correlogram between the firing rates of the two highlighted cells in the top panel. The blue rectangle indicates the time window (0.7–4.7 ms) used to detect putative monosynaptic connections. **b**, A cell pair consisting of an internal direction cell (ID, green) with a putative projection to a conjunctive grid cell (Conj., pink). Temporal cross-correlogram of firing rates (left), directional tuning (middle) and position tuning (right) (rate expressed by colour intensity, ranging from the 10th to the 99th percentile of each cell's firing-rate map) for the putative pre- and postsynaptic cells. Note the similar preferred direction of the two cells (middle). **c**, Preferred directions for all pairs (dots) of putatively connected internal direction to conjunctive grid cells (144 pairs from 13 animals). **d**, Schematic of inferred connectivity

between internal direction cells and conjunctive grid cells. Internal direction cells relay the direction signal to conjunctive grid cells with similar directional tuning. **e**, A cell pair consisting of a conjunctive grid cell (pink) projecting to a pure grid cell from the same grid module (blue). Panels are as in **b**. The firing fields of the pure grid cell are shifted along the preferred direction of the conjunctive cell. The direction and magnitude of this offset (green arrow) was found by cross-correlating the spatial rate maps of the cells. **f**, Preferred direction of presynaptic conjunctive grid cells and grid phase offset direction between pre- and postsynaptic cells for all pairs (dots) of putatively connected conjunctive grid to pure grid cells (85 pairs from 12 animals). **g**, Schematic of inferred connectivity between conjunctive grid cells and pure grid cells. Conjunctive grid cells project asymmetrically to pure grid cells, and excite pure grid cells with a grid phase offset that is aligned to the preferred direction of the conjunctive cell. In this connectivity scheme, activating a set of conjunctive cells with a particular preferred direction leads to a sweep-like trajectory in that direction in pure grid cells.

Sweeps extended into the previously unvisited space along the sides of the tracks, as for the direction signals (Fig. 4b).

Representations of unvisited locations were similarly obtained when a 'latent manifold tuning' (LMT) framework[46] was used to decode direction and position directly from the track data, instead of referring to data from a different session (Fig. 4c and Extended Data Fig. 10). The remaining analysis of unvisited space was therefore performed with LMT analyses. Two latent variables (one in 1D and one in 2D) were respectively initialized with tracked head direction and position, and were then iteratively fitted to capture the fast dynamics of internal direction cells and grid cells. After some iterations, the latent position signal started to travel, in alternating directions on successive theta cycles, into the inaccessible open space surrounding the animal's path, either beside the edges of the elevated tracks (Fig. 4c–f and Extended Data Fig. 9c,e) or beyond the opaque, high walls of the open field (Extended Data Fig. 9d). Sweep directions correlated strongly with decoded internal direction, regardless of whether the sweeps terminated within or outside the environmental boundaries (Fig. 4f). LMT analysis further showed that individual grid cells were tuned to unvisited locations within a sweep's length outside the open-field box or within the interior holes of the wagon-wheel track, in agreement with a continuation of the periodic grid pattern (Fig. 4g,h and Extended Data Fig. 9f,h). The same pattern of results was obtained when using a single-cell Poisson generalized linear model (GLM)-based model to infer out-of-bounds tuning for each cell independently (Extended Data Fig. 9g,h).

LMT analyses showed that sweeps also extended beyond the walls of the open-field arena in hippocampal place cells, in coordination with out-of-bounds sweeps in the MEC (Extended Data Fig. 9i). Individual place cells similarly showed tuning to locations beyond the arena walls

(Extended Data Fig. 9j). Taken together, these findings show that a seamless map of ambient space, including grid cells, internal direction cells and place cells, is created independently of whether the animal ever visits the locations covered by the sweep signals.

## Sweeps and internal direction signals persist in sleep

If sweeps are a fundamental feature of the grid-cell system generated entirely by local circuit properties, they might be present regardless of sensory input, past experience or behavioural state. In agreement with this idea, sweeps maintained their stereotypic alternation profile during foraging in darkness and in novel environments (Extended Data Fig. 9k,l and Supplementary Video 4), as well as during sleep in a resting chamber (Fig. 5, Extended Data Fig. 11 and Supplementary Video 5). Data from the sleep sessions were analysed further. Considering that head-direction cells and grid cells traverse the same low-dimensional manifolds during sleep as in the awake state[5,18,19,47,48], we decoded direction and position during sleep using fitted LMT tuning curves from a same-day open-field session (Fig. 5a,b).

Sleep sessions were segmented into epochs of REM sleep and slow-wave sleep (SWS; Extended Data Fig. 11a). We recorded on average $99.1 \pm 34.6$ min (mean ± s.d.) of sleep per rat in 9 rats, of which $11.3 \pm 4.5\%$ was classified as REM sleep and $88.7 \pm 4.5\%$ as SWS (average epoch durations, $85 \pm 54$ s and $232 \pm 223$ s (mean ± s.d.), respectively). During REM sleep, the population dynamics of internal direction cells and grid cells was similar to when awake. Spiking activity was highly theta-rhythmic (Fig. 5a and Extended Data Fig. 11b,c), and internal direction decoded from neighbouring peaks of activity alternated from side to side (alternation in $70.1 \pm 1.8\%$ of triplets of neighbouring peaks; mean alternation after shuffling peaks, 50.0%; angles between decoded direction at

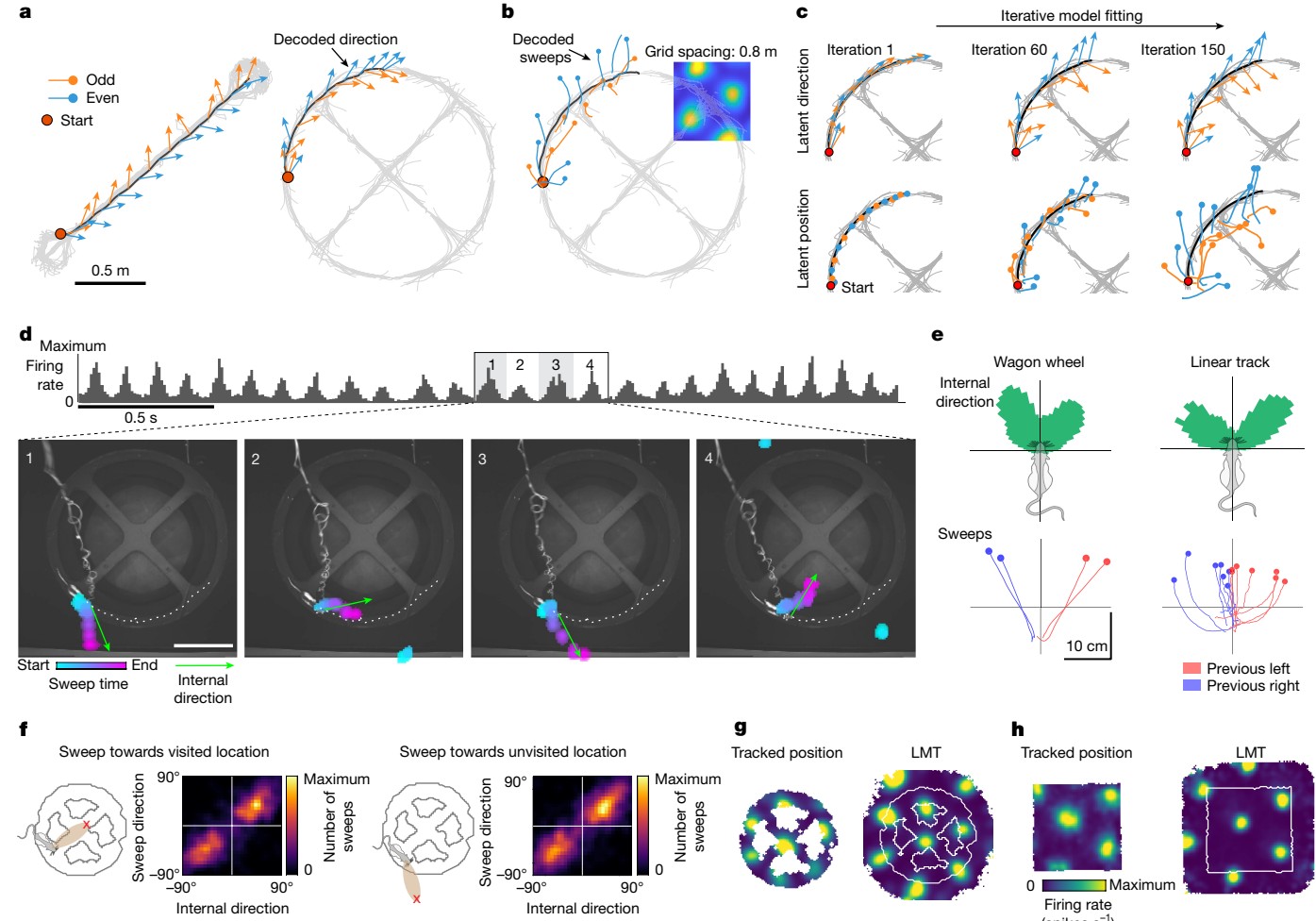

**Fig. 4 | Sweeps extend to never-visited locations. a**, Decoded (internal) direction signals (PV correlation, based on head-direction tuning curves from the same session, all MEC–parasubiculum cells) point towards inaccessible and never-visited locations along an elevated linear track (left) or a wagon-wheel track (right). The decoded direction (arrows of constant length) is shown for successive theta cycles over the course of 3-s running segments. Alternating theta cycles are shown in different colours; black, running trajectory from the segment; grey, full trajectory. **b**, Sweeps on the wagon-wheel track decoded from a single grid module based on rate maps from a preceding open-field session (inset shows an example rate map). The decoded position during each theta cycle is plotted on top of the animal's running trajectory. **c**, The LMT model allows decoding to include never-visited locations. The LMT model is fitted to the neural data by iteratively updating a latent 1D and 2D trajectory. Orange and blue arrows and lines show latent direction (top) and position trajectory (bottom) during 17 theta cycles from a wagon-wheel session at different stages of the model-fitting (left to right). The full running trajectory is shown in light grey. The latent direction and position signals are initialized with the rat's actual head direction and running trajectory (iteration 1) but evolve into sweep-like trajectories that cover the 2D space surrounding the maze (iteration 150). **d**, Sequence of four successive sweeps (1–4) and concurrent internal direction during navigation on an elevated wagon-wheel maze (Bayesian decoder, based on fitted LMT tuning curves for all MEC–parasubiculum cells). Each video frame shows the internal direction (green arrow, length is fixed) and high-probability positions (coloured blobs, as in Fig. 1a, bottom) during one sweep. Note that sweeps travel into the inaccessible space inside and outside

neighbouring peaks, 30.5 ± 2.7° (mean ± s.e.m.) to the opposite side of the previous pair of peaks; mean shuffle, 0.04°, 9 rats; Fig. 5a,c and Supplementary Video 5). Individual grid modules expressed rhythmic sweeps that reset at each theta cycle, alternating in direction across successive theta cycles (Fig. 5a,d and Supplementary Video 5). Sweep

the navigable track. Scale bar, 0.5 m. **e**, Top, circular histogram showing head-centred distribution of fitted LMT internal direction values from one example session on the wagon-wheel track (left) and one example session on the linear track (right). Note that internal direction is bimodally distributed around the animal's head direction (wagon wheel, 38.2° and 9.1° to either side, left–right alternation in 83.4% and 68.3% of theta cycles, $n = 2$ rats; linear track, 28.3° ± 8.3° to either side, left–right alternation in 78.1 ± 2.3% of theta cycles, mean ± s.e.m. from 7 rats), as in the open field (Fig. 2). Bottom, lines show sweeps averaged across each recording session during theta cycles that followed a left (red) or right (blue) sweep (wagon-wheel, 2 rats; linear track, 7 rats). **f**, Out-of-bounds sweeps consistently coincide with internal direction pointing towards the same location. Colour-coded 2D histograms of conditional occurrences of the two LMT latent variables (internal direction and sweeps in head-centred coordinates) for sweeps that terminate inside (top) or outside (bottom) visited portions of the environment in one animal (rat 25843, the same session as **a**–**c**). The circular correlation coefficient between head-centred internal direction and sweep directions was similar when analysis was confined to theta cycles in which sweeps terminated inside versus outside the wagon-wheel track: $r = 0.83$ versus $r = 0.82$. Similar results were obtained for a second rat with fewer cells (not shown): $r = 0.34$ versus $r = 0.35$. **g**, Firing-rate maps of a grid cell on the wagon wheel, based on either the original position coordinates (left) or the latent position fitted by the LMT model (right). **h**, Firing-rate maps of the same grid cell in an open-field session recorded on the same day. Note that the LMT model infers the continuation of grid-like periodic tuning to locations beyond the environment boundaries. Credit: rat, scidraw.io/Gil Costa.

directions were aligned with decoded internal direction (absolute mean angle between sweeps and internal direction, 18.4 ± 9.6°; offset in shuffled data, 91.6 ± 5.3°; Extended Data Fig. 11h). Sweep lengths were comparable with those recorded in the open field (25.4 ± 0.62% (mean ± s.e.m.) of grid module spacing). The sweeps were nested on

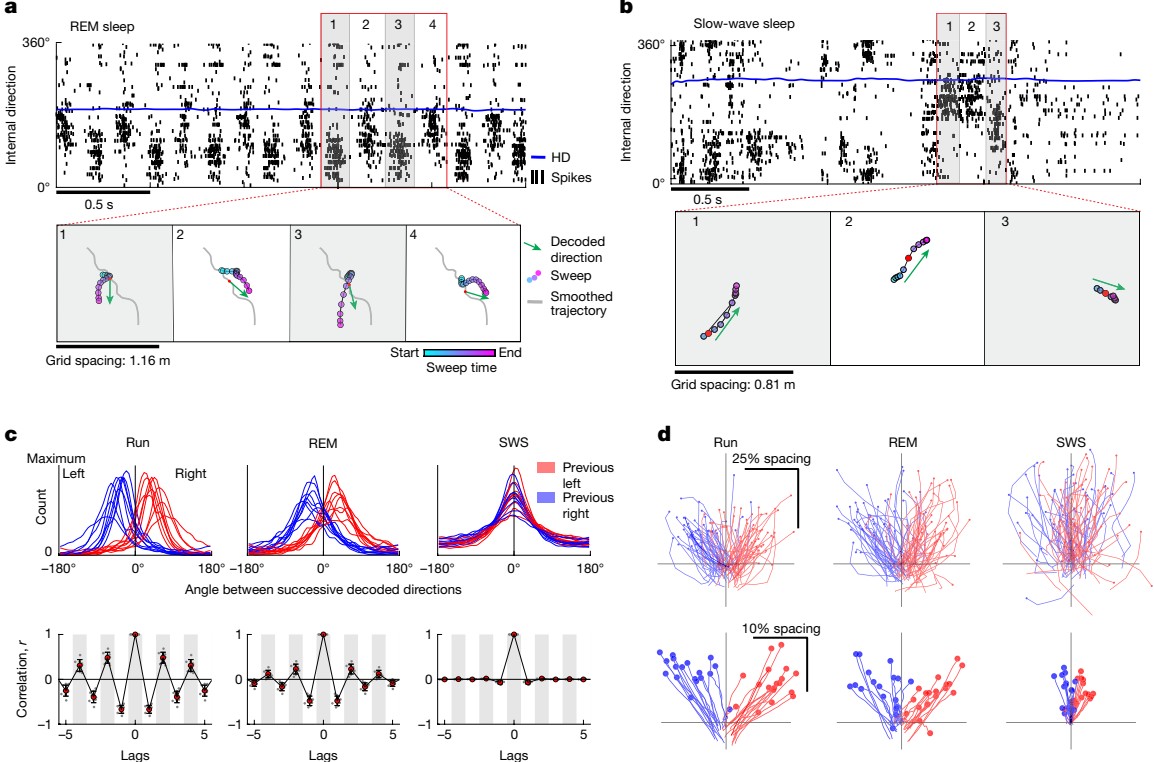

**Fig. 5 | Sweeps and internal direction signals persist during sleep. a**, Sweeps and alternating internal direction pulses are preserved during REM sleep. Top, raster plot with spike times (black ticks) of internal direction cells sorted by preferred firing direction, and tracked head direction (blue line), during a 2.5 s extract from an epoch of REM sleep. Note the theta-paced, left–right alternating packets of direction-correlated activity. Bottom, decoded sweeps (filled circles, colour-coded by time) from grid cells of one module and decoded internal direction from direction-tuned cells (green arrows) during four successive theta cycles (1–4 in the top panel). The grey line shows the reconstructed 2.5 s trajectory after smoothing with a wide gaussian kernel ($\sigma$ = 100 ms). For both position and internal direction, a Bayesian decoder was used, using tuning curves estimated by the LMT model during a preceding open-field foraging session. **b**, Internal direction-aligned, non-rhythmic trajectories during SWS. Top, raster plot (as in **a**) showing activity of internal direction cells during a 3 s extract from SWS. Note the sharp transitions between the up and down states. Bottom, decoded position from a single grid module (as in **a**) during each of three highlighted segments of an up state in the top panel. Note the sweep-like position trajectories aligned with the decoded internal direction (green arrows)

in each segment. **c**, The decoded direction alternates from side to side during awake open-field running (Run) and REM, but not during SWS. Top, distribution of angles between decoded direction at successive peaks of activity ($n$ = 9 rats, 1 session per rat) when the previous decoded direction was directed to the left (red) or right (blue). Bottom, autocorrelogram of decoded direction across brain states. Coloured dots, mean; whiskers, s.d. Note the rhythmic alternation during awake and REM. **d**, Top, decoded trajectories for 100 example sweeps from one example grid module (same decoding method as in **a**), coloured according to whether the previous decoded direction pointed to the right (blue) or left (red). Because spatial representations are decoupled from physical movement during sleep, sweep trajectories are referenced to the low-pass-filtered decoded trajectory (smoothed with a 100 ms gaussian kernel) and aligned to a 'virtual head direction' (low-pass-filtered decoded direction). Individual sweeps are plotted as separate lines. Bottom, averaged sweeps across brain states for all 18 grid modules of all 9 animals. Sweeps were referenced, rotated and sorted as in the top panel, normalized by the spacing of the grid module, and then averaged across all theta cycles. Each pair of red and blue lines corresponds to one module.

top of behavioural timescale trajectories extending several metres over the course of tens of seconds (Extended Data Fig. 11g), mirroring how sweeps extend outwards from a slower running trajectory during awake exploration (Fig. 1a).

During SWS, the population dynamics were less regular, and spiking activity was confined to brief bursts during up states, followed by silent down states (Fig. 5b and Extended Data Fig. 11b,c). There was no periodic left–right alternation of the internal direction signal between successive local maxima in the summed population activity, neither within nor across up states (alternation on only 52.4 ± 0.3% of triplets of neighbouring peaks; mean shuffle, 50.0%; average angle between neighbouring activity peaks, only 5.9 ± 0.9° (mean ± s.e.m.); mean shuffle, 0.01°; Fig. 5b,c). During SWS, bursts of direction-tuned population activity were often accompanied, in grid cells, by sweep-like trajectories aligned to the decoded internal direction signal (sweeps were observed in 30.0 ± 1.9% (mean ± s.e.m.) of identified peaks in internal direction cell activity; absolute mean angle between sweep-like trajectories and internal direction, 13.7 ± 7.2°; corresponding offset in shuffled data,

86.7 ± 8.1°; Fig. 5b,d and Extended Data Fig. 11d–f). As with the internal direction signals, the sweep trajectories were not rhythmic (Extended Data Fig. 11i). Thus, coordinated direction and sweep signals can exist in all states, but theta activity is required to maintain the rigidly coupled rhythmic side-to-side pattern.

## Sweeps cover nearby space with optimal efficiency

After dissociating sweep and direction signals from navigational goals and spatial decision processes, we hypothesized that the alternation of sweep directions instead signifies an efficient strategy[49] for covering ambient space. To test this hypothesis, we simulated an ideal sweep-generating agent that chooses sweep directions that tile space with optimal efficiency. We created a simple model of the spatial footprint of a sweep (Extended Data Fig. 12a,b), based on our previous observation that sweep lengths are proportional to grid module spacing (Fig. 1g). At each time step, the sweep-generating agent was tasked with choosing a sweep direction that minimizes overlap with the area

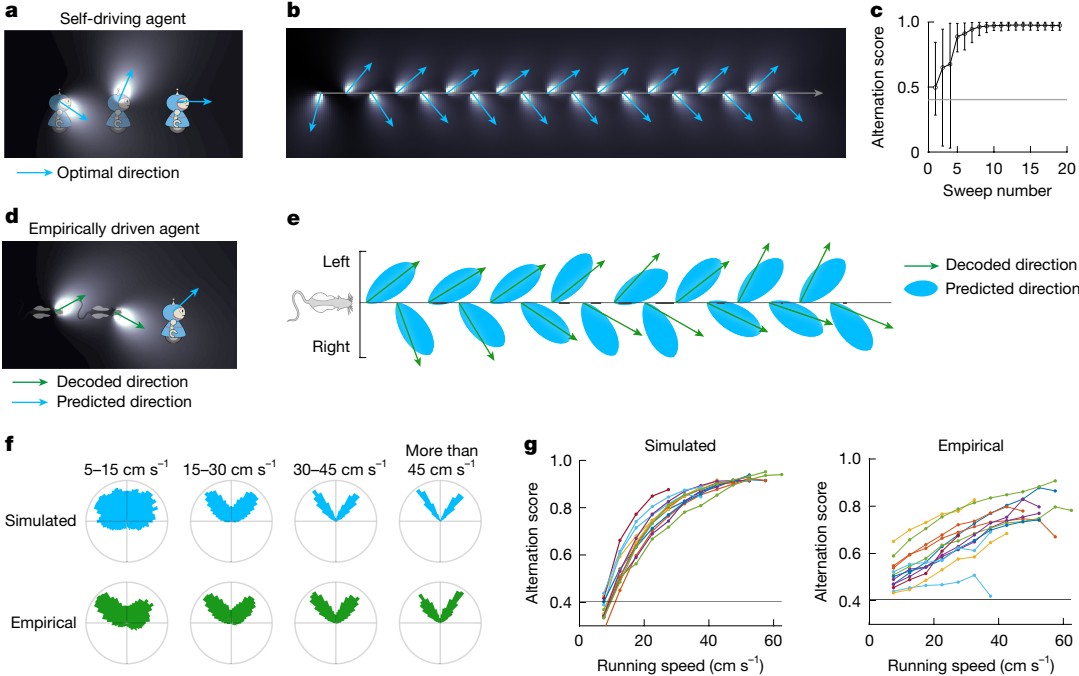

**Fig. 6 | Sweeps cover nearby space with optimal efficiency. a**, Illustration of an autonomous sweep-generating artificial agent. The agent is shown at a series of three positions as it moves from left to right. At each time step, it generates a beam-shaped 'sweep' (white illuminated regions) in a particular direction (blue arrows). The sweep footprints persist in time and serve as a memory trace of previously covered locations. At the current time step (right), the agent calculates the direction that minimizes overlap with the coverage trace; this will become the direction of its next sweep. **b**, The agent is moved along a scale-free linear path (grey horizontal line) and generates a sweep at each time step (sweeps as shown in **a**). The selected sweep directions (blue arrows) spontaneously alternate between two directions relative to the direction of movement. **c**, Alternation of sweep directions in 1,000 runs of the simulation in **b**. Circles show the mean alternation score (range 0–1) for each time step; error bars indicate the 5th and 95th percentiles across runs. The horizontal line indicates the expected alternation score for random, uniformly distributed angles. **d**, Illustration of an alternative 'empirically driven' agent that predicts observed sweep directions. As in **a**, the agent chooses the overlap-minimizing sweep direction at every time step. However, here the sweeps in the previous coverage trace are set at empirically decoded positions and directions, instead of the agent's previous choices. Hence, this model variant predicts the optimal next sweep direction, given the empirical data up to the current time. **e**, Sweep directions predicted by the empirically driven agent shown in **d** (blue shapes) and internal direction fitted by the LMT model (green arrows) during 16 theta cycles in an open field (both plotted in a head-centred reference frame). Theta cycles are evenly spaced along the horizontal axis. Note the alignment between decoded and predicted directions. **f**, Running speed modulates the distribution of sweep direction. Top row, distribution of head-centred sweep directions chosen at different running speeds by the agent in an open-field session (same session as in **e**, but the simulation was now run in 'self-driving' mode without the influence of decoded direction). Bottom row, distributions of head-centred internal direction decoded from empirical data (in the same session). In both cases, the bimodality of the distribution increases with running speed. **g**, Alternation of sweep directions increases with speed. Left, average alternation score of the sweep directions chosen at different speeds by the agent in self-driving mode during open-field foraging. Each set of connected coloured dots shows data from a different rat (*n* = 13). The horizontal line indicates the expected alternation score for random, uniformly distributed angles. Right, the same as the left graph, except using decoded (empirical) internal direction (with the same animals). Credits: rats, scidraw.io/Gil Costa; robots, openclipart.org/annares.

covered cumulatively by previous sweeps, without foresight of upcoming sweep directions (Fig. 6a). When the agent was moved along a linear path at a constant speed, it generated sweeps that alternated between two characteristic directions, 33.0 ± 0.25° (mean ± s.e.m.) to either side of the movement direction across 1,000 runs with random initial conditions (Fig. 6b and Extended Data Fig. 12c), resembling empirical sweep directions on a linear track (Fig. 4a). The prevalence of alternation was quantified using a score that measured alternation in a sliding window of three successive sweep directions, with scores ranging from 0 (no alternation) to 1 (perfect alternation). The agent reliably converged on an alternating pattern that exceeded chance level from the third sweep (alternation score at third sweep, 0.66 ± 0.34 (mean ± s.d.), $P = 2.1 \times 10^{-56}$ with respect to chance level of 0.40, one-tailed sign test; Fig. 6c) and approached perfect alternation towards the end of the run (alternation score, 0.97 ± 0.020). The simulation results point to alternation as a stable regime for minimizing the overlap of successive sweeps (Fig. 6b and Extended Data Fig. 12c). Robust alternation was obtained across a range of sweep widths (Extended Data Fig. 12d).

To determine whether the agent could predict the angles of individual sweeps in the empirical neural data, we moved the agent along the animal's recorded locomotor path in the open-field arena, and for each sweep detected in the empirical data, we determined the optimal sweep direction at the animal's position based on previous empirical directions (using the fitted LMT internal direction variable; Fig. 6d). To prevent the agent from being influenced by sweeps that occurred at similar locations far in the past, we introduced a temporal decay to the cumulative coverage trace of previous sweeps (Extended Data Fig. 12e). The agent's chosen directions aligned with empirical directions in all 13 animals with sufficient numbers of internal direction cells to reliably extract internal direction (correlation simulated versus decoded direction, $r = 0.36 \pm 0.08$, $P < 0.001$ in all animals; offset from the decoded direction, 0.27 ± 1.1° (mean ± s.e.m.); Fig. 6e). The sweep directions of the agent alternated in step with the decoded directions in 68.8 ± 0.04% (mean ± s.e.m.) of the theta cycles, which is more often than expected by chance ($P < 0.001$ for all sessions in all animals with respect to a chance level of 50%, binomial test; Fig. 6e and Extended Data Fig. 12h). This alignment was maintained across a range of temporal decay factors, with the highest accuracy obtained with rapid forgetting (mean decay constant, $\tau = 0.04$, $n = 13$ animals, corresponding to a halving of sweep trace intensity every 215 ms; Extended Data Fig. 12e). In both

the simulated and the empirical data, the egocentric distribution of sweep and direction signals became increasingly bimodal during fast and straight running, with alternation scores increasing with running speed and path straightness (correlation of speed versus alternation, $r = 0.92 \pm 0.01$ (mean $\pm$ s.e.m.) for simulated and $r = 0.96 \pm 0.014$ for decoded directions; straightness versus alternation, $r = 0.917 \pm 0.014$ and $r = 0.862 \pm 0.045$, respectively; $P < 0.05$ in 12 of 13 animals; Fig. 6f,g and Extended Data Fig. 12g). Alternation was maintained in a multimodule version of the model in which sweep directions were chosen independently for each module (Extended Data Fig. 12i).

## Discussion

We show that grid cells encode trajectories that, in each theta cycle, sweep outwards from the animal's location, with direction alternating between left and right across successive theta cycles. These trajectories are particularly prominent in burst-firing grid cells. The sweeps were aligned to a similarly alternating directional signal in a separate population of direction-tuned neurons. Sweeps were identified also in hippocampal place cells, but these were delayed compared with sweeps in grid cells, indicating that they were propagated from the MEC[24,35]. Collectively, the findings point to a specialized circuit of space-coding neurons that on alternating theta cycles generates paths to locations on the left and the right side of a navigating animal's trajectory. The findings provide a framework for understanding corollaries of sweep sequences in individual cells, such as theta phase precession[6,50] and theta cycle skipping[10,40–42].

The sustained expression of directionally alternating sweeps and direction signals, and the invariance of sweep geometry on the grid-cell manifold, indicate that sweeps have a fundamental role in subsecond mapping of the surrounding environment. Grid cells were shown to sweep with directional offsets that maximize coverage of the ambient space, at lengths traversing a substantial fraction of the periodic grid-cell manifold. Sweeps may enable grid cells to link locations in the proximal environment into a continuous two-dimensional map without extensive behavioural sampling[51,52], allowing maps to be formed (or updated) faster and more effectively than if animals had to physically run on each of those trajectories. In familiar environments, sweeps may enable animals to retrieve existing representations of the surrounding space, one sector at a time, mirroring the alternating sonar beams of echolocating bats[53]. The rigid lengths and directions of sweeps indicate that sweeps are defined with reference to the internal toroidal manifold, independently of the external environment.

However, the presence of a hardwired side-shifting mechanism does not exclude the possibility that, with training in a structured environment, sweeps may be aligned with paths towards goal locations or around obstacles, in agreement with claims from studies observing forward-directed sweeps on linear or T-shaped mazes in the hippocampus[9,10,54] or downstream of it (see, for example, ref. 55). In those studies, sweeps, decoded only with respect to visited positions on the track or maze, extended down possible future paths, with alternations on the maze stem reflecting upcoming bifurcations. In the present study, we were able to decode representations in the full ambient 2D space, beyond the animal's path, either by leveraging the invariance of grid phase relationships from a different condition or by using a latent-variable model that characterizes position tuning to any location in the nearby environment. The finding that sweeps invariantly alternated between left and right in the full 2D space raises the possibility that forward-looking sequences reported previously in linear environments reflect projections of left–right-alternating sweeps onto the animal's running trajectory, with sweeps towards unvisited lateral locations going undetected because the decoding procedures used could match activity only with locations that had been physically visited.

Our findings provide some clues to the mechanisms of sweep formation. The functional connectivity analyses raised the possibility that internal direction cells, directly or indirectly through conjunctive grid cells, drive the generation of grid-cell sweeps in the same direction[21–23,25]. The projections from conjunctive grid cells to pure grid cells were asymmetric, in the sense that they preferentially activated cells with grid phases displaced in the direction of the internal direction signal, mirroring a vector-integration mechanism proposed for bump movement in continuous attractor network models for grid cells[11,13,15,21]. The scalar component of the vector computation remains to be identified, however. Sweep lengths are not a direct reflection of running speed, because sweeps persist during sleep. Instead, or additionally, sweep length may be influenced by factors such as the intensity or duration of the internal direction input. Our observations also leave open the mechanism of directional alternation. Two classes of alternation mechanisms can be envisaged. First, alternation may be hardwired into the connectivity of the circuit. Rhythmic alternations reminiscent of those observed here have been described in many brain systems: in left–right shifting spinal-cord circuits for locomotion[56]; in inspiration–expiration circuits in the medulla[57]; and in the hemisphere-alternating REM sleep circuits of reptiles[58]. Alternations between opposing states in these networks rely on central pattern generator mechanisms in which activity is switched periodically between two internally coupled subcircuits[58–61]. A central pattern generator might also underlie left–right alternations in the cortical navigation circuit. Second, and alternatively, alternating sweep directions may emerge spontaneously through a spatial overlap-minimizing rule, without explicit implementation of alternation, as shown in the present artificial-agent simulations. A plausible mechanistic substrate for such a rule exists in single-cell firing-rate adaptation[25,62–67], which penalizes repeated activation of the same neural activity patterns.

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

## Methods

### Subjects

The data were obtained using 19 Long Evans rats (18 males and 1 female; 300–500 g at the time of implantation). Data from five of the animals have been used for other purposes in published data[5,68]. The rats were group-housed with three to eight of their littermates before surgery and were thereafter housed singly under enriched circumstances in large two-storey metal cages (95 × 63 × 61 cm) or in smaller Plexiglas cages (45 × 44 × 30 cm). They were kept on a 12 h:12 h light:dark schedule in humidity- and temperature-controlled rooms. Experiments were approved by the Norwegian Food Safety Authority (FOTS ID 18011 and 29893) and done in accordance with the Norwegian Animal Welfare Act and the European Convention for the Protection of Vertebrate Animals used for Experimental and Other Scientific Purposes.

### Surgery and electrode implantation

In 19 rats we implanted Neuropixels silicon probes targeting either the MEC–parasubiculum region (ten rats, of which three were implanted bilaterally), the hippocampus (two rats) or both regions (seven rats). Neuropixels prototype phase 3 A single-shank probes[26] were used in eight of the rats, and prototype 2.0 multi-shank probes[27] were used in the other 11 rats. Probes targeting the MEC–parasubiculum were implanted 4.2–4.7 mm lateral to the midline and 0.0–0.3 mm anterior to the transverse sinus, at an angle of 18-25° in the sagittal plane, with the tip of the probe pointing in the anterior direction. Probes were lowered to a depth of 4,100–7,200 μm. Hippocampal probes were positioned vertically mediolaterally 1.4–3.0 mm from the midline and anteroposteriorly 1.9–4.0 mm posterior to bregma. In rats with probes in both the MEC–parasubiculum and hippocampus, the two probes were implanted in different hemispheres (except for one animal). Two of the rats with probes in the hippocampus had also received injections of an adeno-associated virus (AAV8) carrying the fluorescent marker mCherry and the HM4Di DREADD receptor bilaterally in the hippocampus 3–4 weeks before probe implantation. The recordings included in the present study were obtained before any administration of the DREADD agonist Descholoroclozapine. The implants were secured with dental cement. A jeweller's screw in the skull above the cerebellum was connected to the probe ground and external reference pads with an insulated silver wire. The detailed procedure for chronic Neuropixels surgeries has been described elsewhere[27]. Postoperative analgesia (meloxicam and buprenorphine) was administered during the surgical recovery period. Rats were left to recover until they resumed normal foraging behaviour, at least 3 h after surgery.

### Electrophysiological recordings

Instruments and procedures were similar to those described for Neuropixels recordings used in the lab[5,26,27,68]. In brief, neural signals were amplified (gains of 500 for phase 3A and 80 for 2.0 probes), filtered (0.3–10 kHz for phase 3A and 0.005–10 kHz 2.0 probes) and digitized at 30 kHz by the probe's on-board circuitry. Signals were multiplexed and transmitted to the recording system along a tether cable. SpikeGLX software (https://billkarsh.github.io/SpikeGLX/) was used to control acquisition and configure the probes. A motion-capture system, based on retroreflective markers on the implant, OptiTrack Flex 13 cameras and Motive recording software, was used to track head position and orientation in 3D. The 3D tracking coordinates were subsequently projected onto the horizontal plane for estimation of 2D position and head-direction azimuth. Back markers were attached in one animal. Another camera (Basler acA2040-90umNIR) was used to capture overhead infrared video in a subset of the recordings. Overhead video frames were aligned to OptiTrack tracking data with an affine transformation between corresponding points in the video and tracking data. Timestamps from each data stream were synchronized as previously described[5,68], by generating randomized sequences of digital pulses with an Arduino microcontroller and sending them to the Neuropixels acquisition system as direct TTL input and to the OptiTrack system and video camera by means of infrared LEDs placed on the edge of the arena.

### Behavioural procedures

Recordings were obtained while the rats foraged in an open field, while they navigated for rewards on a linear track or on a wagon-wheel circular track, or during sleep. Owing to the previously reported gradual decay in signal quality over the first 7–14 days after probe implantation[70], most recordings were made in the first week after surgery (full range, 0–151 days postoperatively). All behavioural tasks for a particular animal were performed in the same recording room (except for one recording in a novel room; Extended Data Fig. 9k,l), often consecutively on the same day. Recording sessions were sometimes interrupted to remove twists from the Neuropixels tether cable. During presurgical training, some of the rats were food restricted, maintaining their weight at a minimum of 90% of their free-feeding body weight. Food restriction was not used in any of the animals at the time of recording.

### The open-field foraging task

Eighteen of the rats foraged for randomly scattered food crumbs (corn puffs or vanilla cream cookies) in a square open-field box with a floor size of 150 × 150 cm and a height of 50 cm. The floor was made of black rubber, and the walls were made of black expanded PVC plastic. The arena was placed on the floor centrally in a large room (16 or 21 m²) with full visual access to background cues. A large white cue card was affixed to one of the walls (at the same height as the wall, a width of 41 cm, and a horizontal placement at the middle of the wall). In all illuminated trials, at the time of the surgery, each rat was already highly familiar with the environment and the task (having experienced 10–20 training sessions before surgery, each lasting at least 20 min). Recording sessions lasted 23–141 min.

In one exceptional case, a rat foraged during recording in a dark room encountered for the first time (Extended Data Fig. 9k,l). In this experiment, a circular arena 150 cm in diameter was used, as described in a previous study[68]. The arena was encircled by thick, dark-blue curtains. All light sources in the recording room were turned off or occluded before the recording started.

In another exceptional case, a rat foraged in a 1 m x 1 m square environment with a transparent plexiglass floor. As well as the overhead camera, a Basler camera was placed under the floor to record the rat's footsteps. DeepLabCut[69] was used to track the positions of each paw, snout and tail base. Body-part coordinates were interpolated in time to match the 10-ms bins of marker-based tracking and neural data, and were spatially aligned with the marker-based tracking data using an affine transform between snout coordinates from the video and head coordinates from the marker-based tracking.

### The linear track task

Five rats with MEC–parasubiculum implants, of which two also had hippocampal implants, shuttled back and forth on a 200 cm linear track with liquid rewards delivered at each end (chocolate-flavoured oat milk dispensed by tube). When the rat consumed a reward at one end of the track, the reward port at the opposite end was refilled. Before surgery, rats were trained on the track task until they consistently completed around 40 laps in one training session. Recording sessions lasted 45–66 min, with 18–48 min of running between reward sites.

### The wagon-wheel task

Two rats with MEC–parasubiculum implants were tested on a wagon-wheel track, an elevated circular track 10 cm wide with two perpendicular cross-linking arms spanning the diameter of the circle[5]. The track was fitted with eight reward wells, placed halfway between each of the five junctions. Each of the wells could be filled with chocolate oat milk by attached tubing. At any given time, a pseudorandom

subset of 1–4 of the wells was filled to encourage steady exploration of the entire maze. Before surgery, the rats were trained to asymptotic performance levels, for which they obtained at least 30 rewards per 30-min session.

### The m-maze task
One rat with probes implanted in the hippocampus and the MEC–parasubiculum was tested on an m-shaped maze, consisting of a central arm connected to two side arms through a T junction (arm lengths of 1.2 m, a T length 1.1 m and a track width of 10 cm)[10,71]. Reward wells were placed at the ends of each arm and were remotely filled with a liquid reward. The rat was trained to visit each of the side arms alternately, returning to the central arm between each visit to the side arms (centre–right–centre–left–centre–right and so on), receiving rewards at the end of each successful trajectory. Before being exposed to the m-maze, the rat was trained to shuttle back and forth on a linear track daily for around a week with multiple sessions each day. After 2–3 days with repeated exposure to the m-maze, the rat displayed almost perfect task performance and was implanted. The data shown in Extended Data Fig. 8 were obtained during the night after surgery.

### Natural sleep
Sleep recordings, including both REM and SWS epochs, were obtained from nine rats with MEC–parasubiculum implants (two of them had combined MEC–hippocampal implants). Sleep was promoted by putting the rat in a black acrylic box (40 cm × 40 cm floor, 80 cm high), lined with towel on the floor, before recording. The box walls were transparent to infrared, allowing the rat's position and orientation to be tracked through the walls. Water was available ad libitum. During recording, the room lights were on and pink noise was played through the computer speakers to mask any background sounds. Sleep sessions typically lasted 2–3 h (see Extended Data Fig. 11a for an example recording).

### Spike sorting and single-unit selection
Spike sorting was done using KiloSort 2.5 (ref. 27), with customizations as previously described[5]. To exclude low-firing units, fast-firing interneurons and contaminated clusters, units were excluded if they had a mean spike rate of less than 0.1 Hz or greater than 10 Hz (less than 0.025 Hz or more than 5 Hz for hippocampal units), or if their waveforms had a large spatial footprint (a similar waveform amplitude across a wide range of channels), because this seemed to be a reliable indicator of poor cluster quality[27]. The waveform footprint was expressed as the anatomical spread of recording channels in which the unit was detected (with detection defined as at least 10% of the maximal amplitude of the unit across all channels), weighted by its waveform amplitude on each channel[27]:

$$\sum_{i=1}^{N} \mathbf{w}_i((\mathbf{x}_i - \overline{\mathbf{x}})^2 + (\mathbf{y}_i - \overline{\mathbf{y}})^2),$$

where $\mathbf{x}_i$, $\mathbf{y}_i$ refers to the $x$, $y$ location of the $i$th recording channel, $\overline{\mathbf{x}}$ and $\overline{\mathbf{y}}$ are the centres of mass of the recording channel positions where the unit was detected, $\mathbf{w}_i$ is the amplitude of the waveform at the $i$th recording channel, and $N$ is the number of channels where the unit was detected. Units were excluded if the spatial footprint of their waveforms exceeded 35 μm (or 50 μm for hippocampal units). Units that were recorded on sites located outside the regions of interest (the MEC–parasubiculum or hippocampus) were excluded from further analysis.

### Preprocessing and temporal binning
During awake sessions, only time epochs in which the rat was moving at a speed greater than 5 cm s$^{-1}$ were used for spatial analyses. Spike times were binned in 10-ms time bins for all population analyses (unless otherwise specified), and tracking data were resampled at the same time intervals to align them with the spike-count data. For computational

reasons, awake sessions were truncated in length to the nearest multiple of 100 s (given by the chunk size for the LMT model) by trimming the last part of the behaviour session.

### Rate maps and angular tuning curves
To generate 2D rate maps for the open-field arena and wagon-wheel track, position estimates were binned into a square grid of 2.5 × 2.5 cm bins. For each bin, we calculated the firing rate of each cell (the number of spikes in the bin divided by time spent in the bin). Rate maps were smoothed with a cross-validated smoothing procedure. In brief, the recording was split into ten folds of equal duration, and the firing rate $y$ during each fold was compared with the expected firing rate $\hat{y}$ based on the rate map calculated over the remaining nine folds and smoothed with a gaussian kernel of width $\sigma$. The value of $\sigma$ (1 cm < $\sigma$ < 50 cm) that minimized the mean squared error of the firing-rate prediction (using the MATLAB function fminbnd) was chosen to smooth the rate map. The same procedure was used to compute spatial rate maps with respect to latent position signals from the LMT model (see the 'The LMT model' section). For PV decoding analyses, a fixed-width gaussian kernel was used to smooth the rate maps ($\sigma$ = 7.5 cm). Spatial autocorrelations and grid scores were calculated as described previously[37], based on the individual cells' rate maps.

Angular tuning curves with respect to head direction, theta phase or internal direction (see the 'Decoding of internal direction based on PV correlations' section) were calculated by binning the angular variable into 60 evenly spaced angular bins. For each 6° bin, the spike rate was calculated as the number of spikes divided by time spent in the bin. Angular tuning curves were smoothed with the same cross-validated smoothing procedure as the spatial rate maps (0.01 rad < $\sigma$ < 1 rad), except in PV-decoding analyses, for which a fixed-width gaussian kernel of $\sigma$ = 12° was used.

### Identification of grid cells and grid modules
Grid cells were detected as groups of cells corresponding to grid modules by finding clusters of co-recorded cells that expressed similar spatially periodic activity in the open field, based on a similar procedure described previously[5]. In brief, 2D autocorrelograms were calculated from the coarse-grained spatial rate maps of each cell (10 cm × 10 cm bins, no smoothing across bins). Autocorrelogram bins within a central radius of two bins or beyond an outer radius corresponding to the rate-map size were masked, before vectorizing and concatenating the autocorrelograms in a matrix. Considering the spatial autocorrelograms of all cells as a point cloud, for which each point (autocorrelogram) has an $N$-dimensional position representing the value of each of its $N$ spatial bins, the Manhattan distances between all points were calculated, and each point's 30 nearest neighbours were identified. The resulting neighbourhood graph was given as input to the Leiden clustering algorithm, which was used to partition the spatial autocorrelograms into clusters, using a resolution parameter of 1.0 (1.5 for sessions with more than 1,000 units). Clusters that contained cells with clear and consistent grid patterns were classified as candidate modules of grid cells. For each cluster, grid periodicity was measured by the grid score of the median autocorrelogram across all cells in the cluster. Grid-pattern consistency was measured by computing the Pearson correlation between the average autocorrelogram of the cluster and the autocorrelogram of each individual cell. The median consistency across all cells in the cluster was defined as the grid consistency of the cluster. For a cluster to be classified as a grid module, three criteria had to be fulfilled: a cluster grid score greater than 0.3; a grid pattern consistency greater than 0.5; and the cluster needed to contain a minimum of ten cells. In some recording sessions, single grid modules seemed to be split into two clusters with similar spacing and orientation. For this reason, we added a step that merged grid clusters if the correlation between their average autocorrelograms was greater than 0.7.

## Subclasses of grid cells defined by differential bursting

To measure the tendency of cells to fire in bursts, we devised a burst score (BS) based on the firing-rate autocorrelograms of each cell (time range ±50 ms, bin width 1 ms). The autocorrelogram value of the centre bin was set to zero, and the autocorrelogram counts were normalized by the mean. Next, we compared the autocorrelogram values at short time lags (2–10 ms) with those found at longer time lags (13–50 ms):

$$BS = \frac{1}{b-a}\sum_{i=a}^{b}\mathbf{y}_i - \frac{1}{d-c}\sum_{i=c}^{d}\mathbf{y}_i,$$

where $\mathbf{y}_i$ is the value of the $i$th bin of the mean-normalized autocorrelogram, and $a$, $b$, $c$ and $d$ are the indices of autocorrelogram bins corresponding to time lags of 2, 10, 13 and 50 ms. Thresholds for classification of bursty versus non-bursty cells were determined by inspection of the distribution of burst scores across the sample (bimodal with a trough around −0.2; Extended Data Fig. 5a). Cells with a burst score greater than 0 were classified as bursty, cells with burst scores less than −0.4 were classified as non-bursty, and cells with burst scores between −0.4 and 0 were left unclassified.

Grid cells could be classified into bursty and non-bursty subclasses on the basis of their burst scores, but a third subclass of grid cells was revealed by closer inspection of autocorrelogram shape or theta phase modulation (Extended Data Fig. 5a). To reliably identify these three subclasses in an unsupervised manner, we applied an unsupervised clustering algorithm to the temporal autocorrelograms of all grid cells[5]. Temporal autocorrelograms were computed for each cell by calculating a histogram of the temporal lags between every spike and all surrounding spikes within a ±100 ms window, using 2-ms bins. The histogram was then divided by the mean value and concatenated in a matrix, discarding any cells with less than 100 counts in the autocorrelogram. Principal components analysis (PCA) was applied to the matrix of autocorrelograms, treating individual cells as observations and keeping the first ten components for further analysis. Next, a neighbourhood graph was constructed by computing the Manhattan distance between all pairs of points in the 10D point cloud and finding each point's 150 nearest neighbours. The graph was used as input to the Leiden clustering algorithm (resolution parameter 0.2). The clustering algorithm detected three distinct clusters with unique autocorrelogram shapes, in agreement with the three subclasses of grid cells identified on inspection of the theta phase modulation and burst scores across all grid cells (Extended Data Fig. 5a). Most cells in the first two clusters had positive burst scores (referred to as bursty type I and bursty type II), but cells in the third cluster had negative burst scores (referred to as non-bursty).

## Classification of direction-tuned cells

Cells were classified as tuned to HD if their HD tuning curves differed significantly from a uniform distribution ($P < 0.001$, Rayleigh test for non-uniformity) and was stable across the first and second half of the recording session ($P < 0.01$, Pearson correlation between tuning curves from the first and second half). No further criteria on tuning strength were applied to classify HD tuning, because many internal direction-tuned cells display weak HD tuning and there was no clear cut-off between HD-tuned cells and non-HD-tuned cells (Extended Data Fig. 6a). Cells were classified as internal direction cells if their tuning curves, with respect to the decoded internal direction (see the 'Decoding of internal direction based on PV correlations' section), passed the above criteria for non-uniformity and stability, and had a mean vector length greater than 0.3. Tuning width was defined as two standard deviations of the tuning curves.

## Theta phase estimation

Theta phase was extracted from the population spiking activity of all units (including fast-firing putative interneurons) in the MEC–parasubiculum region. Spike times were binned into 10-ms bins and the resulting spike counts were bandpass-filtered with a second-order Butterworth filter in the theta range (5–10 Hz). PCA was applied to the matrix of bandpass-filtered spike counts (with units as variables and time bins as observations). The first two principal components typically contained an oscillating circular representation corresponding to the theta rhythm. Theta phase at time $t$ was defined as the direction of the projection of the PV at time $t$ onto the plane defined by PC1 and PC2. The phase with minimal firing activity was defined as zero. In the six rats with dual entorhinal–hippocampal implants, hippocampal sweeps were referenced to the theta phase extracted from MEC–parasubiculum activity. In the two rats with probes only in the hippocampus, the theta phase was estimated from the activity of the hippocampal population.

## Theta-cycle skipping

Theta skipping (the tendency for cells to fire only on alternate theta cycles) was quantified by computing a theta-skipping index (TSI), as in previous descriptions of the phenomenon[10,41,42]. In brief, firing-rate autocorrelograms were generated for each cell (time range ±500 ms, bin width 5 ms, gaussian smoothing $\sigma = 10$ ms), and the relative height of the second theta peak, $p_2$, compared with the first theta peak, $p_1$, was determined as

$$TSI = \frac{p_2 - p_1}{\max(p_1, p_2)},$$

where $p_1$ is defined as the maximum autocorrelogram value between lags 90 and 170 ms, and $p_2$ is the maximum value between lags 180 and 300 ms. Cells were classified as theta skipping if the TSI was positive, that is, if the second theta peak was higher than the first theta peak. The TSI has a clear interpretation only for cells that are theta-rhythmic and is therefore reported for cells that were classified as modulated by theta phase. Cells were classified as theta-phase modulated if the theta-phase tuning curves differed significantly from a uniform distribution ($P < 0.001$, Rayleigh test for non-uniformity) and were stable across the first and second halves of the recording session ($P < 0.01$, Pearson correlation between tuning curves from the first and second halves). Cells were classified as non-rhythmic if they did not meet the above criteria and had a theta-phase mean vector length of less than 0.2.

## Decoding sweeps according to PV correlations

This section describes how we visualized and quantified sweeps. We first decoded the position from the activity of the entire set of MEC–parasubiculum neurons. For each temporal bin, we correlated the instantaneous population activity (the PV) with the session-averaged population activity for each location in the environment (a reference PV, rPV). The inputs to the PV correlation decoder were as follows: first an $N \times T$ matrix of temporally smoothed firing rates (gaussian kernel $\sigma = 10$ ms), with $T$ columns corresponding to time bins and $N$ rows corresponding to neurons; and second, an $N \times M$ matrix of spatial tuning curves, with $M$ columns corresponding to position bins and $N$ rows corresponding to neurons. The spatial tuning curves were normalized by dividing the tuning curve of each neuron by its mean value. To decode position across successive time bins, we computed the Pearson correlation coefficient between each PV (columns of the spike-count matrix) and each rPV (columns of the tuning-curve matrix). This yielded a vector of correlation values for each time step, with elements corresponding to each [$x$, $y$] location in the environment. The decoded position was taken as the position bin with the highest correlation value, and the resulting decoded trajectory was smoothed with a $\sigma = 8$ ms gaussian kernel. To filter out unreliable estimates, the peak correlation value for each time bin was compared with a shuffled distribution of PV correlation values (computed by shuffling rows of the tuning-curve matrix). Decoded estimates were discarded if their PV correlation did not exceed the 99th

percentile of the shuffled distribution. Decoding estimates were also discarded for time points for which fewer than five cells were active.

In each theta cycle, the decoded position swept outwards from a location slightly behind the tracked head position of the animal (Fig. 1). Thus, the starting locations for a series of sweeps formed a slowly evolving trajectory that roughly followed the animal's running trajectory. This trajectory, referred to as the lowpass-filtered decoded trajectory and estimated by decoding position from spikes emitted in the beginning of each theta cycle, was used as a reference signal for measuring sweeps. Spike counts from the first half of each theta cycle were smoothed with a wide gaussian kernel ($\sigma = 1.7$ theta cycles) before the PV-correlation decoding method was used to decode position across all bins. The resulting decoded trajectory was smoothed with a $\sigma = 10$ ms gaussian kernel. Sweeps seemed to be more reliably anchored to the lowpass-filtered decoded trajectory than to the tracked head position of the animal. This difference was particularly evident when rats navigated in darkness, when sweeps and lowpass-filtered signals could deviate substantially from the rat's actual trajectory (Extended Data Fig. 9k,l).

### Extracting sweep trajectories
Individual sweeps, defined as smooth spatial trajectories in each theta cycle, were extracted from the decoded position trajectory by a simple sequence-detection algorithm. In each theta cycle, candidate sweeps were identified as the longest sequence (the highest number) of consecutive valid time bins for which the decoded position jumped less than 20 cm and changed direction less than 90° between consecutive 10 ms bins. Candidate sweeps were truncated to maximize the net Euclidian distance from beginning to end, which effectively removed folds at either end of the sweep. A sweep vector, **s**, was defined as the vector between the lowpass-filtered decoded position at the beginning of the theta cycle and the most distal point of the candidate sweep. To identify sweep trajectories that were fairly straight and well described by the sweep vector, we measured, for each candidate sweep, the goodness of fit, $r^2$, between the sweep-vector axis and the collection of $[x, y]$ points in the sweep, represented by the vectors **x** and **y**:

$$r^2 = 1 - \frac{\mathrm{var}(\mathbf{e})}{\mathrm{var}(\mathbf{x}) + \mathrm{var}(\mathbf{y})},$$

where **e** is a vector of residuals of all points with respect to the sweep-vector axis, such that $\mathbf{e}_i$ is the residual of point $[\mathbf{x}_i, \mathbf{y}_i]$. Candidate sweeps were kept for further analyses if they included at least four samples and had $r^2 > 0.5$ (meaning that most of the variance is explained by the sweep axis). Sweep direction and sweep length were defined as the direction and magnitude of the sweep vector. Sweep prevalence was computed as the fraction of theta cycles (where the animal moved faster than 15 cm s$^{-1}$) containing sweeps that passed the selection criteria. When comparing sweep prevalence in different populations of cells (grid cells versus non-grid cells), the number of cells in the largest population was down-sampled (repeated 100 times) to match the number of cells in the other population.

For several analyses and visualizations (for example, Fig. 1b), sweeps (in allocentric coordinates) were transformed to head-centred coordinates. This was done by first subtracting the tracked position (for example, Fig. 1b) or lowpass-filtered decoded trajectory (for example, Extended Data Fig. 3e) and then negatively rotating each $[x, y]$ coordinate by the animal's head direction. Session-averaged sweeps (for example, Fig. 1b) were computed by first interpolating each head-centred sweep trajectory at 50 time points linearly spaced from the beginning to the end of the trajectory. Interpolated sweeps were grouped into those that followed a right sweep or left sweep, before an average sweep was computed for each group by taking the median position at corresponding time points within the sweep (Figs. 1b, 4e and 5d and Extended Data Fig. 3e,f).

### Temporal delay between hippocampal and entorhinal sweeps
To quantify the temporal delay between hippocampal and entorhinal sweeps, we first decoded the position independently from activity in the hippocampus and the MEC–parasubiculum. Next, we computed, for each region, the offset between decoded position and the rat's position, projected onto the rat's head axis. Finally, we computed the temporal cross-correlation between head-centred decoded position from each region. The peak of the cross-correlogram nearest zero was taken as the temporal delay between hippocampal and entorhinal sweeps. MEC theta troughs were used as a reference for plots in Fig. 1i and Extended Data Fig. 3g, but only for visualization purposes. Similar delays were observed when the number of cells was balanced across regions by down-sampling cells in the region with more cells to match the number in the other region (data not shown).

### Decoding of internal direction on the basis of PV correlations
Internal direction was decoded from the activity of all MEC–parasubiculum neurons using the same PV-correlation procedure described for position decoding except using angular tuning curves for head direction instead of spatial-position rate maps. Because we observed that internal direction cells were activated in discrete pulses during each theta cycle (Fig. 2c), the time bin corresponding to the theta phase with maximal activity was used to express internal direction in each theta cycle. The decoded internal direction $\alpha$ at time $t$ was taken as the circular mean of all possible decoding angles weighted by the correlation values for each directional bin at time $t$:

$$\alpha = \mathrm{angle}\left( \sum_{j=1}^{M} \mathbf{r}_j \exp(i \times \boldsymbol{\theta}_j) \right),$$

where $\boldsymbol{\theta}_j$ denotes the angular value of the $j$th bin, $i$ is the imaginary unit, $\mathbf{r}_j$ is the correlation value for the $j$th angular bin and $M$ is the total number of angular bins. The presence of negative correlation values did not affect the location of the mean. Similar results were obtained with Bayesian decoding for which posterior probabilities are always positive (Extended Data Fig. 4c,d).

For some analyses and visualizations (for example, Fig. 2d), we rotated the decoded internal direction (in allocentric coordinates) to a head-centred reference frame by subtracting the animal's head direction.

### Left–right alternation of sweeps and internal direction
The extent of directional alternation across successive sweeps and internal direction signals was characterized in a head-centred reference frame (see above) during periods when the animals moved faster than 15 cm s$^{-1}$. The data were generally thresholded at 15 cm s$^{-1}$ because alternation was more reliable during running (see Fig. 6f,g, which also includes data for speed thresholds down to 5 cm s$^{-1}$). The prevalence of directional alternation was computed by counting triplets of theta cycles for which sweep direction or internal direction alternated in a left–right–left or right–left–right pattern (detected as sign inversions in the angles between successive directions) divided by the total numbers of theta-cycle triplets for which sweeps or internal direction were detected. The fraction of theta-cycle triplets with directional alternation was compared with a shuffled distribution of scores for which head-centred directions were randomly shuffled (1,000 iterations). In Extended Data Fig. 3i, we repeated the same analysis except with windows of alternation extending from 3 to 20 theta cycles. Directional alternation was visualized in temporal autocorrelograms of angles between successive head-centred sweep direction or internal direction. Autocorrelograms were computed as the circular correlation between the original trace of head-centred directions and a series of lagged versions of the signal (lags from −7 to 7 theta cycles). To find and visualize directional modes of sweep and internal direction angles, we computed

histograms of head-centred directions that were conditioned on the decoded direction in the previous theta cycle. First, decoded directions were classified as left- or right-directed according to the sign of their angular offset from the previous cycle. Next, two histograms were computed, one for decoded directions that followed a left-directed angle and one for decoded angles that followed a right-directed angle. Because sweeps and internal direction angles alternate consistently, this procedure resulted in two unimodal distributions, one on either side of the reference head direction. The directional modes of sweep or internal direction were taken as the peak of each of the two conditional distributions.

In the artificial-agent simulation, a time-resolved measure of directional alternation was used to quantify instantaneous alternation of the agent's chosen sweep direction. A three-sweep sliding window was used, such that at the $i$th timestep, a triplet of sweep directions $\alpha_{i-1:i+1}$ was selected. In the three-sweep sliding window, the two angles between consecutive sweep pairs were calculated: $a = \alpha_i - \alpha_{i-1}$ and $b = \alpha_{i+1} - \alpha_i$. The alternation score $s$ was therefore computed as:

$$s = \frac{|a - b|}{2 \max(|a|, |b|)},$$

where $||$ denotes the absolute value. The alternation score ranged from 0 to 1, where 1 indicates perfect alternation.

## Single-module decoding

To decode position from individual grid modules, the PV-correlation decoding analysis (first applied on all cells) was next applied to subsets of cells that belonged to individual grid modules. The decoded position from each grid module was mapped onto the hexagonal unit tile of the grid module using a cross-correlation procedure. First, we computed a template grid pattern for each grid module by averaging the spatial rate-map autocorrelograms across all cells in the module. The template grid pattern was cross-correlated with the 2D distribution of PV-correlation values at each time step. Peaks in the resulting spatial cross-correlogram were detected and, for each cross-correlogram, the peak nearest to the origin was taken as the decoded position. To ensure that the decoded position was within the bounds of the central unit tile, the decoded trajectory was wrapped around the three grid axes of the template grid pattern. Single-module sweeps were detected as described for whole-population decoding (by finding consecutive time bins within each theta cycle for which the decoded trajectory formed a smooth trajectory), except that spatial and directional offsets between consecutive decoded positions were computed with periodic boundary conditions derived from the template grid pattern. For some visualizations (for example, Figs. 1e and 4b), the single-module sweeps were aligned to the behavioural trajectory of the rat by subtracting the offset between the animal's tracked position and the lowpass-filtered decoded trajectory at the beginning of each theta cycle.

## Theta phase precession

To show and quantify phase precession in grid cells during 2D open-field foraging, we adapted approaches used in previous studies[72,73]. In brief, we first detected vertices of the grid pattern as local maxima in the spatial rate maps. Next, the vector from the rat's position to the nearest grid vertex was computed for each time step. The resulting vectors were projected onto the decoded internal direction (which serves as a reliable measure of the sweep axis), yielding a scalar 'in-field position' at each time step. In-field position is negative when the animal is approaching a grid field and positive when it is heading out of the grid field. Phase precession was measured as the circular–linear correlation between theta phase and in-field position at the time of each spike, following ref. 74. Cells with a negative correlation coefficient and $P < 0.01$ were defined as phase precessing.

## Phase precession and theta skipping in simulated neurons

To investigate whether alternating sweeps and internal direction signals were sufficient to explain phase precession and cycle skipping in individual cells, we generated spikes from firing-rate models of grid cells and internal direction cells.

Firing-rate predictions for grid cells were generated on the basis of: first, the spatial rate map of an idealized grid cell (implemented as a hexagonal tessellation of 2D gaussian bumps with a spacing of 75 cm, $\sigma = 0.5$ cm and an amplitude of 30 Hz); and second, a time-varying position signal (a recorded trajectory of a rat in the open field, with or without added sweep-like spatial offsets with length up to 35 cm). The firing-rate prediction at each time step was taken as the value of the rate map at the location of the time-varying signal divided by the sample rate (100 Hz). Poisson spike trains were generated from the predicted firing rate. Phase-precession plots were made as described in the previous section. Only spikes from the second half of the theta cycle (corresponding to the outwards phase of each sweep and the active phase of bursty grid cells) were used to compute firing-rate autocorrelograms, because any sweep-induced theta skipping would be visible only during this part of the theta cycle.

Firing-rate predictions for internal direction cells were generated on the basis of tuning curves for internal direction and theta phase, and a time-varying internal direction signal and a time-varying theta-phase signal. In Extended Data Fig. 6h, the tuning curves from a recorded cell and decoded internal direction were used to generate firing-rate predictions. In Extended Data Fig. 6i, simulated tuning curves from a recorded cell and simulated internal direction were used to generate firing-rate predictions. Tuning curves, internal direction and theta phase were simulated as Von Mises distributions with means of 60° and 180°, and concentrations of 6 and 1.5, respectively. Internal direction was simulated by adding ±30° offsets to the animal's head direction in an alternating or non-alternating pattern. The firing-rate prediction, $\lambda$, was estimated by combining the contributions of theta phase and internal direction tuning following a linear–nonlinear Poisson model:

$$\lambda = \exp(\beta_0 + \beta_{id}\mathbf{X}_{id} + \beta_\theta \mathbf{X}_\theta)/\mathrm{d}t,$$

where $\beta_0$ is the log mean firing rate, $\beta_{id}$ and $\beta_\theta$ are the log tuning curves for internal direction, and theta phase and data matrices $\mathbf{X}_{id}$ and $\mathbf{X}_\theta$ represent the time-varying internal direction and theta phase signals. Poisson spike trains were drawn from the firing-rate prediction.

## Identification of putative excitatory connections between cells

Putative excitatory monosynaptic connections between pairs of co-recorded neurons were identified by detecting short-latency, short-duration peaks in the firing-rate cross-correlograms (CCGs), following previous procedures[44,45,75–77]. CCGs were computed with 1 ms bins over a ±50 ms window. A baseline CCG was computed by convolving raw CCGs with a hollowed gaussian kernel ($\sigma = 5$ ms, hollow fraction 60%), which has been shown to approximate a jittered CCG[75]. Neuron pairs recorded on the same probe, for which each of the neurons emitted at least 2,000 spikes and the raw CCG contained at least 1,000 counts, were tested for functional connections. The baseline CCG was subtracted from the raw CCG and subsequent peak detection was performed on the baseline-corrected CCG. Neuron pairs were classified as putatively connected if the highest positive CCG peak satisfied the following five conditions: first, the peak occurred within an asymmetric time range consistent with monosynaptic excitation (0.7–4.7 ms), in agreement with experiments in which the discharge latencies of synaptically coupled cell pairs were measured more directly[76,78,79]; second, the peak height exceeded five standard deviations of the baseline-corrected CCG; third, the $P$-value of the peak was less than 0.001 (estimated from a Poisson distribution with continuity correction, as in ref. 75); fourth, the width of the peak (defined as the set of

bins contiguous with the peak whose values exceeded half of the peak height or two standard deviations of baseline and had $P < 0.01$) was less than 3 ms (consistent with the precise spike timing expected from a monosynaptic connection); and fifth, the width of the peak did not overlap with the zero-lag bin (suggestive of common input). Candidate connections were also discarded if any of the non-peak bins exceeded 2.5 standard deviations of the baseline-corrected CCG or if any of the bins in the anticausal direction had $P < 0.01$.

Overall connection probability was computed by dividing the total number of putatively connected neurons by the total number of pairs that were checked for connections. Similarly, target-specific connectivity rates were computed by dividing the number of connections from one functional cell class to another by the number total number of pairs.

## Decoding internal direction with PCA or UMAP
The internal direction signal was also decoded in an unsupervised manner, with PCA or uniform manifold approximation and projection[80] (UMAP). In this approach, the high-dimensional neural activity was projected down to a 2D subspace to characterize the trajectory of population activity on a low-dimensional manifold. Only rhythmic direction-tuned cells (with head-direction mean vector length greater than 0.3 and theta-phase mean vector length greater than 0.3) were included in the analysis, to avoid interference between the ring-like manifold of interest and other ensemble representations, such as spatial signals from pure grid cells. Spike counts from $n$ neurons were binned into $t$ time bins corresponding to individual theta cycles. Theta-cycle time bins were used instead of 10-ms time bins to prevent global within-cycle firing rate fluctuations from driving the results. PCA was done on the resulting $t$-by-$n$ matrix. Next, internal direction was read out either directly from the PCA output or by applying a second dimensionality-reduction step on the data using UMAP (Extended Data Fig. 4c,d).

In the PCA decoder, internal direction was read out by projecting the neural data onto the first two eigenvectors and taking the arctangent of each [$x$, $y$] coordinate in the resultant 2D projection. Because these angles were arbitrarily rotated with respect to the environment, they were aligned to the environment before further analysis. We assumed that internal direction and head direction had equal mean directions, and subtracted the average difference between the two signals from the decoded signal to align it with respect to the environment.

In the UMAP decoder, scores of the 20 principal components with the largest explained variance were used as input to the nonlinear dimensionality-reduction algorithm UMAP with these hyperparameters: n_components=3, metric=correlation, n_neighbors=199, min_dist=0.3, init=spectral. This yielded a 3D embedding of the high-dimensional population activity. The 3D UMAP point cloud typically showed a clear circular shape. The 3D point cloud was then collapsed to 2D by projecting the points onto the best-fit 2D plane. Hence, the 2D points were converted into angles by taking the arctangent of each $x$ and $y$ coordinate. The resulting angles were aligned to the environment by subtracting the average offset from head direction, as described above.

## Bayesian decoding of position
Sweeps could also be decoded using Bayesian reconstruction (Extended Data Fig. 4a). Position was decoded at 10-ms time steps from the matrix of firing rates and tuning curves from $N$ neurons, with an assumption of Poisson firing and a flat position prior[81]:

$$P(\mathbf{x}|\mathbf{y}) \propto \exp\left( \sum_{i=1}^{N} \mathbf{y}_i \log(f_i(\mathbf{x})) - \mathrm{d}t \sum_{i=1}^{N} f_i(\mathbf{x}) \right),$$

where $P(\mathbf{x}|\mathbf{y})$ is the conditional probability for the rat's 2D location $\mathbf{x}$, given the observed spike count $\mathbf{y}$ and position tuning curves $f(\mathbf{x})$. The decoded position was taken as the position bin that maximized $P(\mathbf{x}|\mathbf{y})$.

## The LMT model
The PV-correlation method has two limitations. First, it can only be used to decode positions and directions that the animal has physically sampled, because it relies on tuning curves with respect to the animal's tracked position and head direction. This is a problem for hippocampal data in particular, because the neural correlation structure varies between environments and brain states[82–84], making it impossible to decode position in one environment based on reference tuning curves from a different context. A second challenge is that the sharp spatial and directional tuning of grid cells and internal direction cells is obscured in standard time-averaged reference tuning curves, because sweeps and internal direction deviate from the tracked position and head direction (Extended Data Figs. 5f and 6d). To simultaneously extract sweeps through unvisited space and characterize spatial tuning directly from neural activity, we therefore adapted the LMT model in ref. 46. In this framework, sweeps may be considered as hidden or 'latent' trajectories on a neural manifold that is not directly observable. The goal of the LMT model is to infer both the latent trajectory and each cell's tuning to locations on the manifold, based on neural population activity. The model assumes that the latent variable evolves smoothly with time, that individual neurons are smoothly tuned to locations on the manifold, and that neurons fire according to a Poisson process (see Extended Data Fig. 10 for a schematic and ref. 46 for details). For a multidimensional (vector-valued) latent variable $\mathbf{x}(t)$, the temporal evolution of component $j$ is modelled as a Gaussian process:

$$\mathbf{x}_j(t) \sim \mathcal{GP}(0, k_t),$$

where $k_t$ is a temporal covariance function, $k(t, t') \triangleq \mathrm{cov}(\mathbf{x}_j(t), \mathbf{x}_j(t'))$. In this case, the exponential kernel $k(t, t') = r\exp(-|t-t'|/l)$ is used, with variance $r$ and length-scale $l$, respectively controlling the amplitude and smoothness of the latent variable. The log tuning curves $f(\mathbf{x})$ are also modelled as Gaussian processes, with the log tuning of the $i$th neuron expressed as:

$$f_i(\mathbf{x}) \sim \mathcal{GP}(0, k_\mathbf{x}),$$

where $k_\mathbf{x}$ is a spatial covariance function, in this case a Gaussian kernel $k_x(\mathbf{x}, \mathbf{x}') = \rho \exp(-\|\mathbf{x} - \mathbf{x}'\|_2^2/2\delta^2)$ with variance $\rho$ and length scale $\delta$. Furthermore, an L-1 penalty parameter, $\lambda$, was included to enforce the sparsity of the fitted tuning. The value of the latent variable $\mathbf{x}(t)$, in conjunction with each cell's tuning curve, $f_i(\mathbf{x})$, predicts the cell's log-firing rate, which is then transformed with an exponential nonlinearity into a Poisson-distributed spike count $y_{i,t}$ as:

$$y_{i,t}|f_i, \mathbf{x}_t \sim \mathrm{Poiss}(\exp(f_i(\mathbf{x}_t))).$$

The predicted and observed spike trains were compared, yielding a log-likelihood value. The model was fitted using an expectation-maximization algorithm that maximizes the log-likelihood of the spike trains by separately optimizing the latent variable and tuning curves in alternation. Over multiple iterations of this two-step optimization procedure, both evolve to capture the latent dynamics in the neural population activity, thus improving the prediction of the observed spikes.

The original LMT framework uses a single latent variable to predict the neural activity; however, as a form of Poisson regression, the LMT model can be trivially combined with other Poisson regression models, as a means of extracting other factors of interest, or to regress out noise. In the present study, we formulated the activity of each neuron as a sum of log firing-rate contributions from five input variables:

$$\mathbf{Y} \sim \mathrm{Poiss}(\exp(\mathbf{M}_{id} + \mathbf{M}_{pos} + \mathbf{M}_{theta} + \mathbf{M}_{hd} + \mathbf{M}_{pop})),$$

where **Y** is the time-by-neurons matrix of predicted spike counts and $\mathbf{M}_{varname}$ is a time-by-neurons matrix of log firing-rate contributions from model varname. The first two contributions correspond to the two latent variables of interest: internal direction ($\mathbf{M}_{id}$) and position ($\mathbf{M}_{pos}$). Internal direction was modelled as a 1D latent circular variable (hyperparameters: $\rho = 0.1$, $r = 100$, $\delta = 0.5$, $l = 0.1$ and $\lambda = 1$) that was initialized with the animal's tracked head direction. Position was modelled as a 2D latent variable (hyperparameters: $\rho = 0.1$, $r = 10$, $\delta = 6$, $l = 0.015$ and $\lambda = 1$) that was initialized with the animal's tracked position. Importantly, grid-like periodic mapping of space is not assumed by the model, which has no periodic constraints or boundary conditions.

The final three covariates (theta phase $\mathbf{M}_{theta}$, head direction $\mathbf{M}_{hd}$ and population firing rate $\mathbf{M}_{pop}$) are known to modulate MEC–parasubiculum activity[4,19,37,85], and are here included as 'noise' covariates to regress out their substantial contributions to the neural activity, hence reducing the likelihood of them influencing the extracted latent variables. Theta phase and head direction were modelled as circular 1D variables in the LMT framework; however, the variables were respectively fixed at the values of measured theta phase and head direction, and only tuning curves were optimized (hyperparameters: $\rho = 1,000$, $\delta = 2$ and $\lambda = 0$ for theta phase, and $\rho = 0.1$, $\delta = 0.5$ and $\lambda = 1$ for head direction). The population firing rate model was implemented as a generalized linear model $\mathbf{M}_{pop} = \mathbf{X}_{pop}\boldsymbol{\beta}_{pop}$, where $\mathbf{X}_{pop}$ is a time-by-neurons matrix of log population firing rates, and $\boldsymbol{\beta}_{pop}$ is a row vector of learnt coefficients for all neurons. Population firing rate was computed as the average instantaneous firing rate across neurons, smoothed with a $\sigma = 20$ ms gaussian kernel.

Each step of fitting in the composite model consisted of serially updating the parameters for each of the five submodels by maximizing the log likelihood of the model. Because the latent trajectories may be arbitrarily rotated and distorted with respect to the physical environment, an alignment procedure was performed after model fitting[46]. The latent internal-direction trajectory was aligned to the tracked head direction of the animal by subtracting the average angle between the two signals. The latent position trajectory was aligned to the tracked position of the animal with an affine transformation. The fitted latent variables were used for most LMT-based analyses of sweeps and internal direction. For some visualizations (for example, that in Fig. 4d) and analyses during sleep, position and direction were decoded from neural activity and fitted LMT tuning curves using the Bayesian framework described in the previous section.

### The GLM-based single-cell tuning model
Theta sweeps introduce offsets between the rat's current location and the location represented by place and grid cells, resulting in smeared receptive fields when a cell's spikes are plotted as a function of animal location (Extended Data Fig. 5e,f). To characterize the spatial tuning of individual cells independently, in a manner that accounts for sweeps, we used a GLM to model the spike train of a cell, **y**, as a function of a set of explanatory variables, **X**, parametrized by the learnt parameters $\boldsymbol{\beta}$:

$$\mathbf{y} \sim \mathrm{Poiss}(\exp(\boldsymbol{\beta}_{const} + \boldsymbol{\beta}\mathbf{X})).$$

### Construction of the data matrix X
The model included four explanatory variables: position, head direction, internal direction (from LMT) and theta phase. The variables were expressed by using a basis-expansion procedure, by which a set of smooth basis functions was used to decompose each single variable into multiple variables. The weighting of each basis function was given by a corresponding $\boldsymbol{\beta}$ parameter, giving the GLM the flexibility to fit any smooth function of the input variable in question. The 2D position variable was expressed by a set of 2D Gaussian basis functions ($\sigma = 2$ cm) arranged in a 10-cm-spaced triangular grid that tiled the open-field arena and a surrounding buffer zone. The angular variables head direction, internal direction and theta phase were expressed by a

set of 50 Von Mises functions ($\kappa = 10$) with equally spaced mean values from 0 to $2\pi$. Basis expansion was done by evaluating each of the basis functions for a given value of the input variable in question.

The basis-expanded representations are high dimensional and multicollinear (that is, the basis function values are correlated). In a regression model, these attributes tend to cause overfitting, so we used PCA to produce a low-dimensional, orthogonal representation of the basis-expanded data matrix. PCA acted as a form of regularization. For position, the principal components explaining 99% of the variance were retained, reducing the dimensionality from 527 to 92. For head direction and theta phase, the principal components explaining 80% of the variance were retained, reducing the dimensionality from 50 to 6. After performing basis expansion and dimensionality reduction for the input variables, the resultant matrices for all input variables were concatenated into the GLM design matrix **X**:

$$\mathbf{X} = [\mathbf{X}_{pos}, \mathbf{X}_{id}, \mathbf{X}_{hd}, \mathbf{X}_{theta}].$$

### Theta-phase-dependent shifting of the position covariate
To model the effect of sweeps on position-modulated firing, we added a preprocessing step that applied a theta phase-dependent shift to the animal's tracked position coordinates. Specifically, the 2D position coordinates (**x**) were parametrically shifted by a distance $\delta$ along the internal direction axis ($\alpha$):

$$\mathbf{x}'_t = \mathbf{x}_t + \delta_t[\cos(\alpha_t), \sin(\alpha_t)],$$

where **x**′ denotes the shifted position coordinates. The shift quantity, $\delta_t$, at each time point was modelled as a function of the current theta phase. Specifically, the shift quantity $\delta$ was modelled as a function shift parameter $\boldsymbol{\gamma}$, fitted by the model, and the basis-expanded theta phase $\mathbf{X}_{theta}$:

$$\boldsymbol{\delta} = \boldsymbol{\gamma}_{const} + \boldsymbol{\gamma}\mathbf{X}_{theta}.$$

The shifting parameters $\boldsymbol{\gamma}_{const}$ and $\boldsymbol{\gamma}$ were fitted together with the GLM $\boldsymbol{\beta}$ parameters using a gradient-based solver with the finite-difference method (MATLAB function *fminunc*).

Across cells and recordings, the GLM shift model yielded sharper receptive fields than standard rate maps with respect to tracked position (Extended Data Fig. 9h). The model's estimates of position tuning were more robust than those from the LMT model, because the latter depended on large numbers of co-recorded spatially modulated cells. Therefore, rate maps based on the GLM-shifted position were deemed most appropriate to use to identify grid cells and grid modules (see the 'Identification of grid cells and grid modules' section).

### Sweeps through unvisited space
For analyses of sweeps and spatial tuning to never-visited locations outside the bounds of the wagon-wheel track, we first defined the area of space that the animal had visited. This was achieved by binning the 2D environment in 2.5-cm bins and finding all the bins that the rat had visited (resulting in a binary 2D map with values of 1 for visited bins and 0 otherwise). The bounds of the animal's coverage was found by applying a binary dilation operation of the occupancy map (Matlab function *imdilate* with a disk-shaped structuring element, radius = 1), followed by a morphological closing operation (Matlab function *imclose* with a disk-shaped structuring element, radius = 1). The zero-valued bins in the resulting occupancy map were defined as never visited.

### Head oscillations
For analyses of coupling between left–right alternation in the internal direction signal and body movements (Extended Data Fig. 2), we extracted vertical and lateral head oscillations from OptiTrack tracking data during linear track running. Because head oscillations are most

prominent during fast, straight running[86], analyses were restricted to epochs with running speed greater than 40 cm s$^{-1}$. Vertical head speed was taken as the first derivative of the highpass-filtered vertical head position. Lateral head speed was computed by highpass-filtering 2D head position in the $x$–$y$ plane, rotating the highpass-filtered signal to align with the animal's movement direction and taking the first derivative in the lateral direction. Spectral analysis was done on 10-ms-binned head-speed signals and multi-unit activity using the multi-tapered Fourier transform, implemented by the Chronux toolbox (http://chronux.org/, function cohgramc). Non-overlapping 2 s windows were used, with a frequency band of 1 Hz and 3 tapers. All signals were bandpass-filtered between 1 and 20 Hz. The relationship between lateral head oscillations and internal direction signals was also investigated using phase-coupling analysis (Extended Data Fig. 2h). To this end, we extracted the phase of the lateral head-swing cycle from the Hilbert transform of the bandpass-filtered lateral head speed (pass band 2–10 Hz). Phases of 0° and 180° were set to match the times of right and left extremes in lateral head position, respectively. We next computed histograms of head-swing phase at times for which internal direction pointed to the left or right side of the animal's head axis. The mean vector of these histograms was used to measure the consistency and strength of phase locking across animals.

## Forelimb footsteps

For analyses of coupling between left–right alternation in the internal direction signal and forelimb footsteps (Extended Data Fig. 2i–k), we extracted forelimb footsteps from under-floor video. The position of paws, snout and tail base were extracted using DeepLabCut. The gait cycle, in which one cycle corresponds to the period from one right forelimb plant (phase 0) to the next, was extracted by projecting the vector between left and right forelimbs onto the head axis and computing the phase of the resulting signal using the Hilbert transform. Phase coupling between alternations in the internal direction signal and the gait cycle was examined using the same method as described for head oscillations.

## Sleep-stage classification

Sleep stages were identified as described in previous studies[5,19]. First, we identified periods of sustained immobility (longer than 120 s, locomotion speed below 1 cm s$^{-1}$, head angular speed below 6° s$^{-1}$). These periods were subclassified into SWS and REM on the basis of delta- and theta-rhythmic population activity in the recorded cells. Population firing rate was computed by summing the binarized 10-ms spike counts from each cell. The rhythmicity of this aggregated firing rate with respect to delta (1–4 Hz) and theta (5–10 Hz) frequency bands was quantified by applying a zero-phase, fourth-order Butterworth band-pass filter and then calculating the amplitude from the absolute value of the Hilbert transform of the filtered signal, followed by smoothing (Gaussian kernel with $\sigma = 5$ s) and standardization ($z$-scoring). Periods for which the ratio of the amplitudes of theta and delta activity (the theta/delta ratio) remained above 5.0 for at least 20 s were classified as REM. Periods during which the theta/delta ratio remained below 2.0 for at least 20 s were classified as SWS (Extended Data Fig. 11b).

## Detection of sweep and internal direction signals during sleep

To decode sweeps and internal direction from neural activity during sleep, we used tuning curves (LMT) from open-field sessions from the same recording day as the sleep session. Position was decoded separately for individual grid modules (see the 'Single-module decoding' section) because the correlation structure of grid cells across brain states may be preserved within but not across modules[5,19]. Because the theta rhythm is absent during SWS, we used local maxima in population activity as reference points for analysis of sweeps and direction signals in all brain states. To detect local maxima in the population

activity, regardless of brain state, the spike counts of all internal direction cells were summed and smoothed with a Gaussian kernel ($\sigma = 20$ ms) before applying the Matlab function *findpeaks* with default parameters to detect peaks in the summed activity. Local maxima occurred at theta-rhythmic intervals while awake and during REM, and irregularly during SWS (Extended Data Fig. 11). Pairs of local maxima 2–250 ms apart were used to quantify directional alternation in all brain states, and all detected maxima were used to measure alignment between sweep and direction signals. Internal direction was taken as the decoded direction at the time of local maxima. To extract sweeps, we identified smooth sequences of decoded positions (sequences for which decoded position jumped less than 15% of grid spacing and changed direction by less than 2 radians between successive time bins) that occurred in windows centred around each of the local maxima in population activity. The windows extended 50 ms to either side of local maxima or to the edge of the neighbouring window. Because spatial representations were decoupled from physical movement during sleep (Extended Data Fig. 11g), sweep trajectories were referenced to the low-pass-filtered decoded trajectory (smoothed with a 100 ms gaussian kernel) and aligned to a 'virtual head direction' (low-pass-filtered decoded direction, $\sigma = 1$ theta cycle gaussian smoothing).

## Simulation of an ideal sweep-generating agent

To test the hypothesis that alternating sweeps are controlled by an algorithm that maximizes the sampling of surrounding space, we simulated a sweep-generating agent that maximized environmental sampling by choosing sweep directions that minimized the overlap with previous sweeps.

First, we modelled the spatial coverage of a single sweep. Grid modules express sweeps at multiple spatial scales, so we reasoned that the total spatial coverage of a sweep may be considered as a sum of sweeps across individual grid modules (Extended Data Fig. 12a). A model sweep footprint was formulated, based on previous empirical observations of geometric relationships between single-module grid patterns[16] and the geometric properties of sweeps in the present results (Fig. 1e–g). In brief, we summed the grid patterns of five idealized grid modules, with an inter-module scale ratio of √2 and gaussian-shaped grid fields with $\sigma = 1/6$ of each module's spacing, at offsets from the origin corresponding to typical single-module sweep lengths of 1/3 of module spacing. The sum of sweeps across modules resembled a torch beam radiating outwards: as distance from the origin increased, the footprint broadened and decayed in intensity (Extended Data Fig. 12b). We approximated this shape by multiplying two simple spatial functions: an inverse distance function and an angular weighting function taken from a Von Mises distribution.

If we let $d$ and $\theta$ denote the distance and direction, respectively, from the agent's $[x, y]$ position, $\mathbf{x}_{agent}$, to a location $\mathbf{x}$ in the environment ($d = \|\mathbf{x} - \mathbf{x}_{agent}\|$ and $\theta = \arctan(\mathbf{x} - \mathbf{x}_{agent})$), the intensity of the sweep footprint at location $\mathbf{x}$ for a chosen sweep direction $\alpha$ becomes:

$$f(\mathbf{x}) = 1/d^2 \exp(\kappa \cos(\theta - \alpha)),$$

where $\kappa$ is the angular concentration parameter of a Von Mises distribution, which determines the angular width of the sweep footprint. A value of $\kappa = 5$ was used initially (Fig. 6b) to reflect the empirically derived sweep shape (Extended Data Fig. 12b), but a parameter search revealed that stable alternation emerged across a range of values of $\kappa$ (Extended Data Fig. 12d).

To run the simulation of sweeps on a linear track, we created an artificial scale-free 2D environment, binned into a 401 × 401 square grid. The agent was moved along a linear path at constant speed and instructed to generate a sweep every time step, by placing a sweep footprint in a specified direction. The cumulative trace of sweeps $h$ at time $t$ was computed by summing the footprints of previous sweeps:

$$h_t(\mathbf{x}) = \sum_{i=1}^{t-1} f_i(\mathbf{x}).$$

The optimal sweep direction $\alpha_{optimal}$ at time $t$ was chosen by finding the angle $\alpha$ that minimized the spatial overlap between the current sweep $f$ and the cumulative trace of previous sweeps $h$:

$$\alpha_{optimal} = \underset{\alpha}{\mathrm{argmin}} \sum_{i=1}^{N} f(\mathbf{x}_i, \alpha) h(\mathbf{x}_i),$$

where the spatial overlap was calculated by multiplying the current sweep footprint and the cumulative sweep trace and summing across all $N$ spatial bins.

Next, the simulation was run using the recorded behavioural trajectory of rats running in the open field. Real-world positions were mapped onto the agent's simulated environment by setting the agent's bin size to 1 cm and placing the open field at the centre of the bin grid. The agent's time steps were yoked to the times of theta cycles in the experimental data, and for each theta cycle, the agent deployed the above algorithm to select the optimal sweep direction. To prevent the agent from being influenced by sweeps that occurred at similar locations far in the past, we introduced a temporal decay factor $\tau$ (range 0–1) that exponentially discounted the intensity of the cumulative coverage trace at each time step:

$$h(\mathbf{x}) = h(\mathbf{x})\tau^{dt}.$$

The agent's sweep-direction choices are determined solely by its previous decisions, so we called this the self-driving version of the model. We also formulated an empirically driven version, for which the agent was tasked with predicting the optimal sweep at each time step, given the directions of previous sweeps decoded from neural data. This was implemented by using the LMT, instead of the agent's past sweep directions, to compute the cumulative sweep trace. Three rats were excluded from these analyses because internal direction could not be reliably estimated in these animals.

To investigate whether alternation also emerged when sweep directions were chosen independently for each grid module, we formulated a multimodule version of the simulation in which sweep footprints were simulated as three Gaussian functions (corresponding to three grid modules) whose width and sweep length increased with a geometric ratio of 1.5. At each time step, sweep directions for each module were chosen according to one of three updating rules: common, parallel or serial. In the common version, a single sweep direction was chosen for all modules, corresponding to the direction that minimized the overlap of the summed multimodule footprint with the summed multimodule trace. In the parallel and serial versions, sweep directions were chosen individually for each module, by minimizing the overlap between the single-module sweep footprint and the summed multimodule trace. In the parallel version, modules were updated simultaneously, meaning that modules were agnostic of each other's sweep directions within same time step. In the serial version, modules were updated sequentially, by updating the summed multimodule sweep trace with the chosen sweep of one module before choosing the sweep direction for the next module.

## Histology and recording locations

The rats received a lethal dose of pentobarbital, after which they were perfused intracardially with saline followed by 4% formaldehyde. The brains were extracted and stored in 4% formaldehyde, and were later cut in 30-μm sagittal or coronal sections using a cryostat. The sections were Nissl-stained with cresyl violet, and probe shank traces were identified in photomicrographs. In 14 animals, recording sites on the probes targeting the MEC–parasubiculum were aligned to the histological sections, as done previously[5], by using as reference points the tip of the probe shank and the intersection of the shank with the brain surface. The aligned shank map was then used to calculate the anatomical locations of individual recording sites (Extended Data Fig. 1). Estimates of anatomical locations are subject to some degree of measurement error owing to the limited accuracy of the alignment process and the fact that units may be detected some distance away from the recording site.

## Data analysis and statistics

Data analyses were performed with custom-written scripts in Matlab and Python. Clustering analyses of grid-cell modules and bursting subtypes of grid cells were conducted using the python package Scanpy[87] and its dependencies (including numpy, pandas, scipy, scikit-learn and matplotlib). The LMT model and the functional-connectivity analyses were implemented by adapting publicly available code from ref. 46 and ref. 77, respectively. Statistical analysis was done using Matlab. Circular statistics were computed using the Circular Statistics Toolbox[88]. Results are reported as mean ± s.e.m. unless otherwise indicated. Statistical tests were non-parametric and two-tailed, unless otherwise indicated. The Mann–Whitney $U$-test was used for unpaired comparisons, and the Wilcoxon signed-rank test was used for paired comparisons. Pearson correlations were used unless otherwise indicated. Power analysis was not used to determine sample sizes. For each animal, the recording session with the best unit yield and behavioural performance was included in the study. The study did not involve any experimental subject groups, so random allocation and experimenter blinding did not apply and were not performed.

## Reporting summary

Further information on research design is available in the Nature Portfolio Reporting Summary linked to this article.

## Data availability

The datasets generated during the current study will be available after publication at EBRAINS, https://doi.org/10.25493/R5FR-EDG. Source data are provided with this paper.

## Code availability

Code for reproducing the analyses in this article will be available after publication at Zenodo[89].

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

**Acknowledgements** We thank T. Waaga for sharing data and help with recording in darkness[68]; B. A. Dunn, M. Nau, Y. Burak and M. Witter for discussion; and S. Ball, K. Haugen, E. Holmberg, K. Jenssen, E. Kråkvik and H. Waade for technical assistance. The work was supported by a Synergy grant to E.I.M. from the European Research Council (KILONEURONS, grant 951319); FRIPRO grants to E.I.M. (grant 286225) and M.-B.M. (grant 300394/H10); two Centre of Excellence grants to M.-B.M. and E.I.M. (Centre of Neural Computation, grant 223262, and Centre for Algorithms in the Cortex, grant 332640); a National Infrastructure grant to E.I.M. and M.-B.M. from the Research Council of Norway (NORBRAIN, grant 295721); a grant from the Kavli Foundation to M.-B.M. and E.I.M.; grants from the K. G. Jebsen Foundation (grant SKGJ-MED-022), St Olav's Hospital and the Regional Health Authorities of Mid Norway (2020/7569); a gift from E. Solberg and family; a direct contribution to M.-B.M. and E.I.M. from the Ministry of Education and Research of Norway; and a PhD fellowship grant from the NTNU Faculty of Medicine and Health Sciences to A.Z.V. Clipart images were adapted with permission from scidraw.io (rat images by Gil Costa and Wenbo Tang) and openclipart.org/detail/190531/little-robot (robot image by annares).

**Author contributions** All authors planned and designed experiments, conceptualized and planned analyses, and interpreted data; A.Z.V. and R.J.G. performed experiments; A.Z.V. and R.J.G. visualized and analysed data; A.Z.V., R.J.G. and E.I.M. wrote the paper. All authors discussed and interpreted data. M.-B.M. and E.I.M. supervised and arranged funding for the project.

**Competing interests** The authors declare no competing interests.

**Additional information**
**Correspondence and requests for materials** should be addressed to Abraham Z. Vollan or Edvard I. Moser.

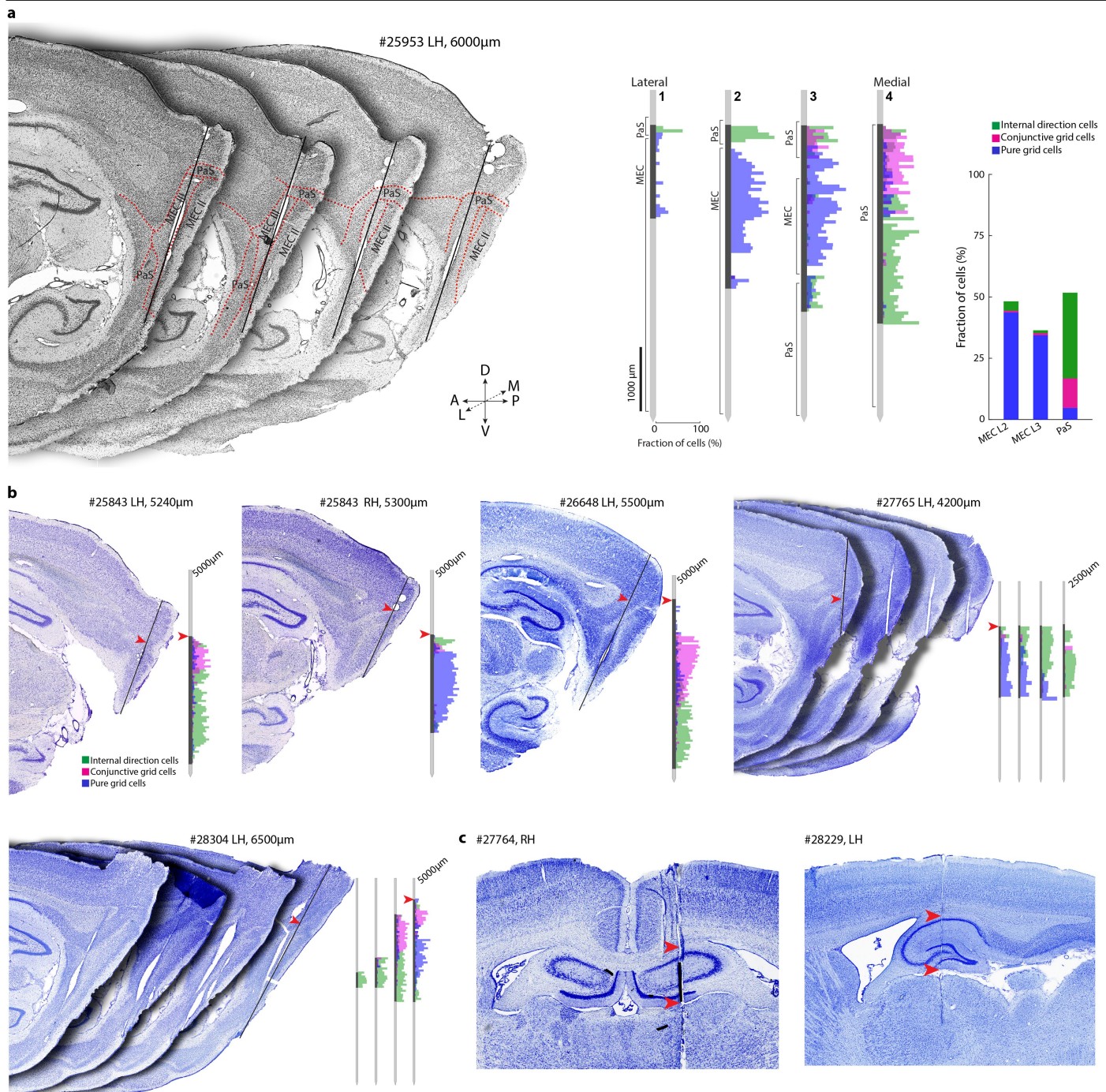

**Extended Data Fig. 1** | See next page for caption.

**Extended Data Fig. 1 | Recording locations and anatomical segregation of functional cell types. a**, Anatomical separation of rhythmic direction-tuned cells ('internal direction cells'), conjunctive grid cells and pure grid cells. Left: serial sagittal sections from rat 25953 (also shown in Fig. 2b) showing tracks from a 4-shank Neuropixels 2.0 probe. Sections are organized from lateral to medial and each section contains the track from one of four recording shanks (~250 μm apart; section with clearest track is shown). Borders between brain regions are indicated with dashed lines (MEC, medial entorhinal cortex; PaS, parasubiculum). Middle: fraction of internal direction cells (green), conjunctive grid cells (pink) and pure grid cells (blue) along each of the probe shanks, estimated by counting the number of functional cells recorded in 50 μm bins along the probe, divided by the total number of recorded cells in each bin. To maximize anatomical coverage of the distribution plot, cell counts are pooled across 7 recording sessions with different configurations of active recording sites (only 384 out of 5,120 sites can be recorded at any given time). Black portions of probe shanks show sites that were recorded from. Right: percentage of recorded cells in MEC and parasubiculum belonging to each of the three functional cell classes (MEC layer II: 43.7% pure grid cells, 0.7% conjunctive grid cells, 3.8% direction-tuned cells; MEC layer III: 34.4% pure grid cells, 0.9% conjunctive grid cells, 1.1% internal direction cells; PaS: 4.8% pure grid cells, 12% conjunctive grid cells, 34.9% internal direction cells; 27,413 cells from 14 rats, 2-7 recording sessions per rat). Note that most pure grid cells in the sample were located in MEC layers II and III (3,377/3,861 or 87.4%), while most of the conjunctive grid cells (1,224/1,285 or 95.3%) and internal direction cells were located in parasubiculum (3,547/3,796 or 93.4%). **b**, Sagittal histological sections with probe tracks for four additional representative animals with probes in MEC-parasubiculum (reconstructions were performed for 14 of the 16 animals with MEC-parasubiculum implants). Animal identity and hemisphere are indicated above each section for the four animals. For each implanted probe shank, the section with the clearest track in MEC-parasubiculum is shown. Insets show, for each animal, the fraction of internal direction cells, conjunctive grid cells and pure grid cells along the probe shanks (as in **a**) across multiple recordings (range: 2–4). Red arrowheads indicate the most dorsal enabled recording site for each probe. Note that functional cell types are anatomically segregated in most animals. **c**, Histological sections (coronal and sagittal) showing two representative examples of recording locations in hippocampus. Arrowheads indicate the dorsoventral range of recording sites that were included in the study.

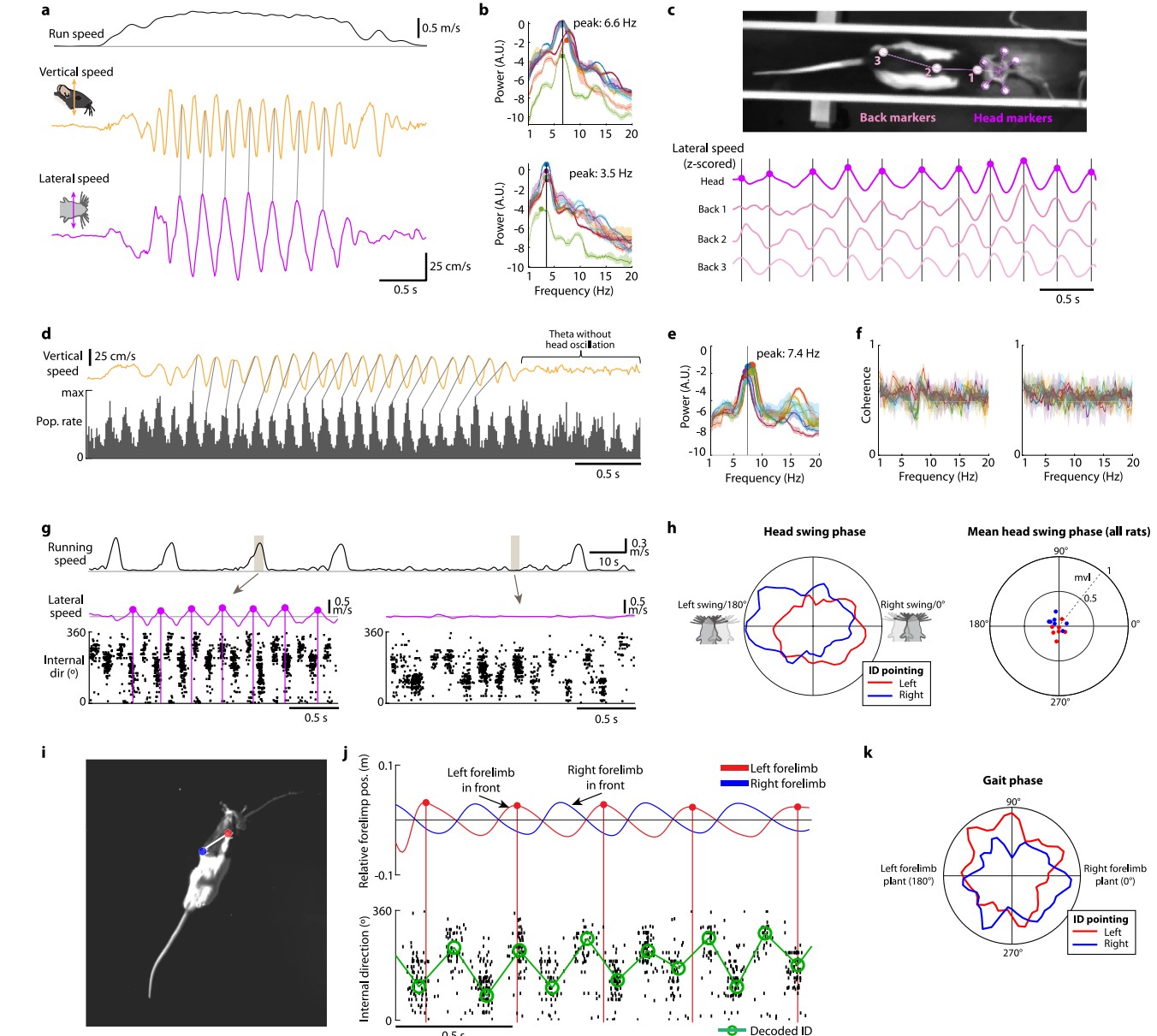

**Extended Data Fig. 2** | See next page for caption.

**Extended Data Fig. 2 | Neuronal theta oscillations are not coupled to oscillations in head movements or locomotion. a**, Running speed and vertical and lateral head speed during traversal of a linear track on a single trial (from central well to side well). Note characteristic vertical and lateral head oscillations during locomotion. **b**, Power spectra of vertical (top) and lateral (bottom) head speed for 7 rats during fast running (>40 cm/s) on a linear track. Note peaks (solid dots) in the theta frequency range[86,90], skewed towards lower values than theta oscillations in hippocampal neural activity. Solid lines show mean power across all 2-second time windows of running; shaded areas represent 95% bootstrap confidence intervals. **c**, Lateral head oscillations are coupled to lateral whole-body oscillations during locomotion. Top: image of a rat with three retroreflective markers (1, 2 and 3) on its back, in addition to standard head-mounted markers. Bottom: lateral speed of head and back markers during a lap on the linear track (from centre well to side well). Note that all markers oscillate together with a consistent phase offset. **d**, Vertical head oscillations are not in phase with neural theta oscillations. The plots show vertical head speed (top) and summed MEC-parasubiculum spiking activity (bottom) during a lap on the linear track. The rhythms have similar but not identical frequencies, resulting in phase drift between the two signals (visualized by lines connecting peaks of the two oscillations). Also note that theta-rhythmic neural activity starts before and ends after the bout of prominent head oscillations. **e**, Power spectrum of MEC-parasubiculum multi-unit activity for 7 rats during fast running ( > 40 cm/s) on linear tracks (plotted as in **b**). Note prominent peak in power at a higher frequency than the head oscillations in **b**. Solid lines show mean power across all 2-second time windows of running; shaded areas represent 95% bootstrap confidence intervals. **f**, Coherence between theta and vertical (left panel) or lateral head oscillations (right panel) across animals. Note absence of peaks in coherence spectrum, suggesting that the two oscillations were not phase-coupled. Solid lines show mean coherence across all 2-second time windows of running; shaded areas represent 95% bootstrap confidence intervals. **g**, There was a similar lack of coupling between lateral head oscillations and the left-right-alternating internal direction signal. Top: running speed during a segment of a linear track recording session. Bottom left: lateral head speed and spike raster from internal direction cells in MEC-parasubiculum during a period of running. Note that left-right-alternating packets of spiking activity in internal direction does not reliably lock with respect to peaks in lateral head speed (red vertical lines). Bottom right: same as bottom left, but during a period of immobility. Note weak but visible left-right-alternating packets of internal direction cell activity in the absence of movement. **h**. Left: angular distribution of left (red) and right (blue)-oriented internal direction signals (with respect to head-axis) across the lateral head swing cycle in an example animal. Note that the two distributions are nearly uniform and only weakly skewed to opposite phases of the lateral head swing cycle. Right: mean vectors for internal direction signals across all animals, plotted across the head swing cycle. Note low mean vector lengths, consistent with weak or nearly-absent modulation by rhythmic head movements. **i**, Paw position was tracked using DeepLabCut on under-floor video captured in a rat foraging in a 1 m open field with a transparent floor. Blue and red dots show tracked positions of left and right forelimbs. **j**, Left-right-alternating internal direction is not systematically coupled to left-right alternation of footsteps. Top panel: tracked position of left and right forelimb during a bout of straight running in the open field (**i**). Position is projected onto the axis of running. Left and right forelimbs move forwards in alternation. Left forelimb plants are shown with vertical blue lines. Bottom: raster from internal direction cells showing left-right-alternating packets of activity. Decoded internal direction is plotted in green. Note that the left-right-alternation of direction is not aligned to the alternation of footsteps. **k**, Left: angular distribution of internal direction signals (with respect to head axis) during left and right footsteps across the gait cycle in a single animal with paw tracking as in **i**. Right and left paw placements are defined as 0 and 180 deg, respectively. Note near-uniform distribution of left/right internal direction indicating a lack of phase coupling between internal direction alternation and footsteps. Credit: rats, scidraw.io/Gil Costa and Wenbo Tang.

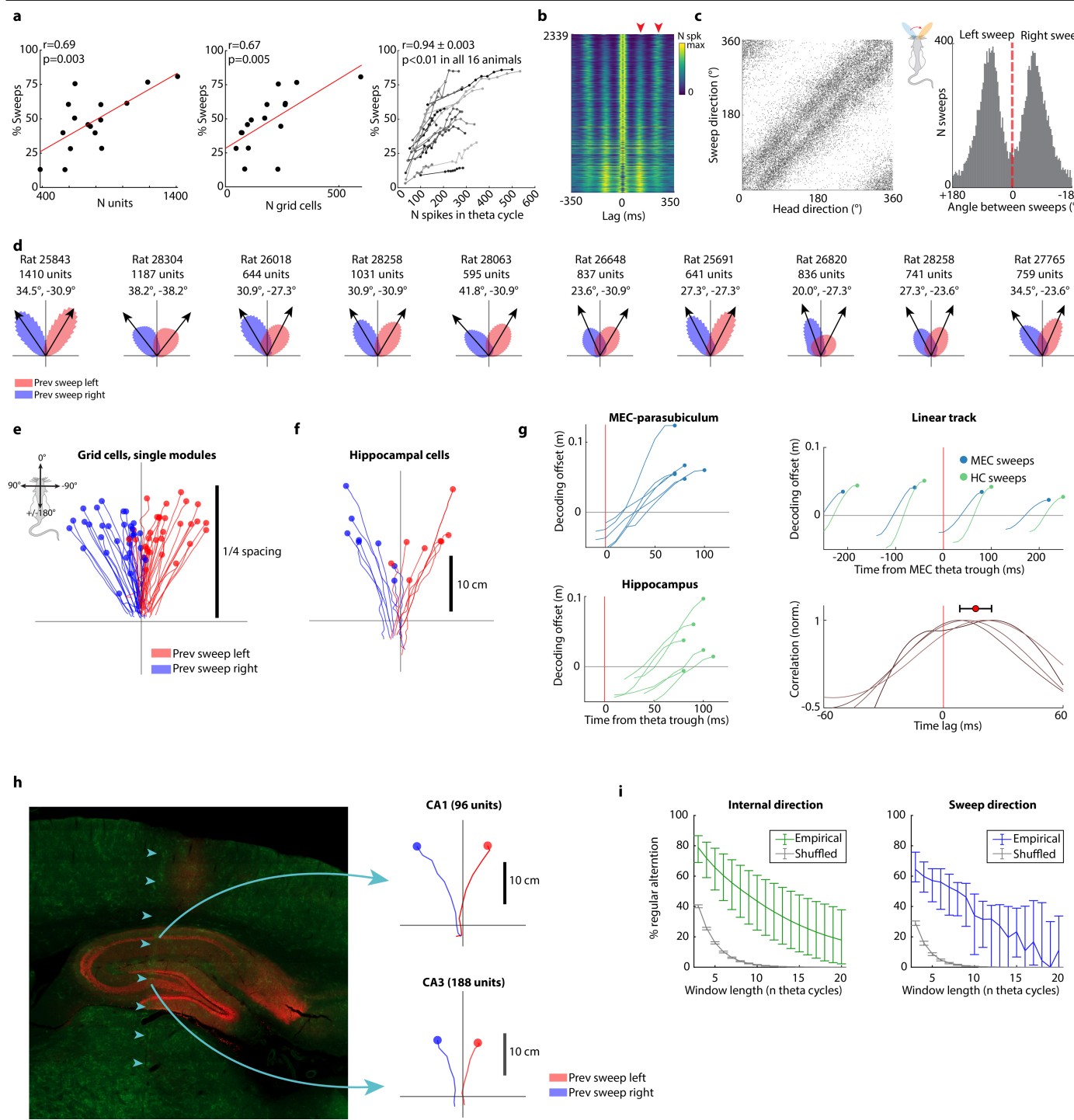

**Extended Data Fig. 3** | See next page for caption.

**Extended Data Fig. 3 | Left-right alternating sweeps in grid cells and place cells. a**, Number of detected sweeps scales with number of recorded MEC-parasubiculum cells (left; Pearson correlation: r = 0.69, p = 0.003, n = 16 animals, 1 session per animal), number of grid cells (middle; r = 0.67, p = 0.005, n = 16 animals, 1 session per animal), and number of spikes per theta cycle (right; r = 0.95 ± 0.01 (mean ± s.e.m.), p < 0.01 in all 16 animals, 1 session per animal), suggesting that the reported fraction of theta cycles with sweeps is underestimated. Data from individual animals are plotted as dots (left and middle) or lines (right). **b**, Stacked firing-rate autocorrelograms (±350 ms) for all theta-modulated grid cells (2,399/3,194 or 75.1% of grid cells were theta-modulated), sorted by tendency to fire on alternating theta cycles. Firing rates are colour-coded. Lags corresponding to repetitions of the ~8 Hz theta rhythm is indicated with red arrowheads. Note that a large fraction of theta-modulated grid cells (1,289/2,399 or 53.7%) display theta skipping, with prominent peaks at lags corresponding to every other theta cycle. **c**, Left: scatter of directions of all sweeps and tracked head direction at the corresponding times during one example recording session (same session as Fig. 1a). Sweep directions are distributed bimodally around the animal's head direction (circular correlation between sweep direction and head direction r = 0.55 ± 0.03, p < 0.001 in all 16 animals, absolute mean offset 3.1 ± 2.8 deg, mean ± s.e.m. across 16 animals). Right: histogram of angles between successive sweeps. Note that these angles are clustered around ±60 deg. Sweeps directed to the right or left of the previous sweep are defined as 'right' and 'left' sweeps, respectively. **d**, Circular histograms of head-centered sweep directions for the 10 animals with the highest fraction of theta cycles with detected sweeps. Sweeps are rotated to head-centred coordinates (head orientation is vertical) and sorted based on whether the previous sweep was directed to the right (blue) or left (red). Note clustering of sweeps around two principal head-centered directions offset ~30 deg to the left and right of the animal's head direction. **e**, Sweeps averaged across theta cycles for all cells of a grid module, as in Fig. 1b, but now showing all modules. Sweeps are plotted with reference to the lowpass-filtered decoded trajectory (origin) and rotated to head-centred coordinates (head orientation is vertical). Sweep lengths are normalized by the grid spacing of each module.

Note that sweeps have similar lengths relative to the scale of the grid cells. **f**, Averaged sweeps decoded from hippocampal ensemble activity (8 rats), plotted as in Fig. 1b. Hippocampal sweeps alternated from side to side in 78.5 ± 3.1% (mean ± s.e.m.) of successive triplets of theta cycles with detected sweeps, significantly more than when sweep directions were shuffled (>99.9th percentile for all animals). **g**, Left column: forward progression of decoded sweeps (projected onto the rats' head axis) as a function of time from the beginning of each theta cycle for MEC-parasubiculum (top) and hippocampus (bottom) for all 6 rats with dual HC/MEC implants during open field foraging, plotted as in Fig. 1i. Data from each of the rats are plotted as separate lines (1 session per rat). Note that hippocampal sweeps are delayed relative to MEC sweeps. Right column: top plot shows forward progression of entorhinal (blue) and hippocampal (green) sweeps during a linear track session from an example animal (plotted as in Fig. 1i). Bottom plot shows cross-correlations of decoded positions in MEC and hippocampus for 4 animals with dual HC/MEC implants that were tested on the linear track (1 session per animal), plotted as in Fig. 1k. Hippocampal sweeps are delayed by 16.2 ± 4.0 ms (mean ± s.e.m.) with respect to entorhinal sweeps on the linear track. **h**, Left: histological section from a rat with an implanted probe in the hippocampus. Probe track through hippocampal subregions is indicated by arrowheads. Green channel: autofluorescence; red channel: mCherry (expressed in hippocampus after a viral injection prior to implantation surgery). Right panel: left-right alternating sweeps from neurons recorded simultaneously in CA1 (top right) and CA3 (bottom right) during foraging in the open field task (data from 1 session from animal shown to the left; sweeps averaged across theta cycles, as in **e**,**f**). Mean sweep lengths were 28.8 cm and 22.6 cm; directions were 25.5 and 27.3 degrees to either side of the head-axis. Cells were recorded on different sites from the same probes (left). **i**, Panels show fraction of epochs with consistent left-right-alternation of internal direction (left) or sweep direction (right), with epoch lengths ranging from 3–20 theta cycles. Solid lines show the median across animals; whiskers show 25th and 75th percentiles (n = 16 animals, 1 session per animal). Shuffles were generated by randomly shuffling head-centered directions across theta cycles. Credit: rat, scidraw.io/Gil Costa.

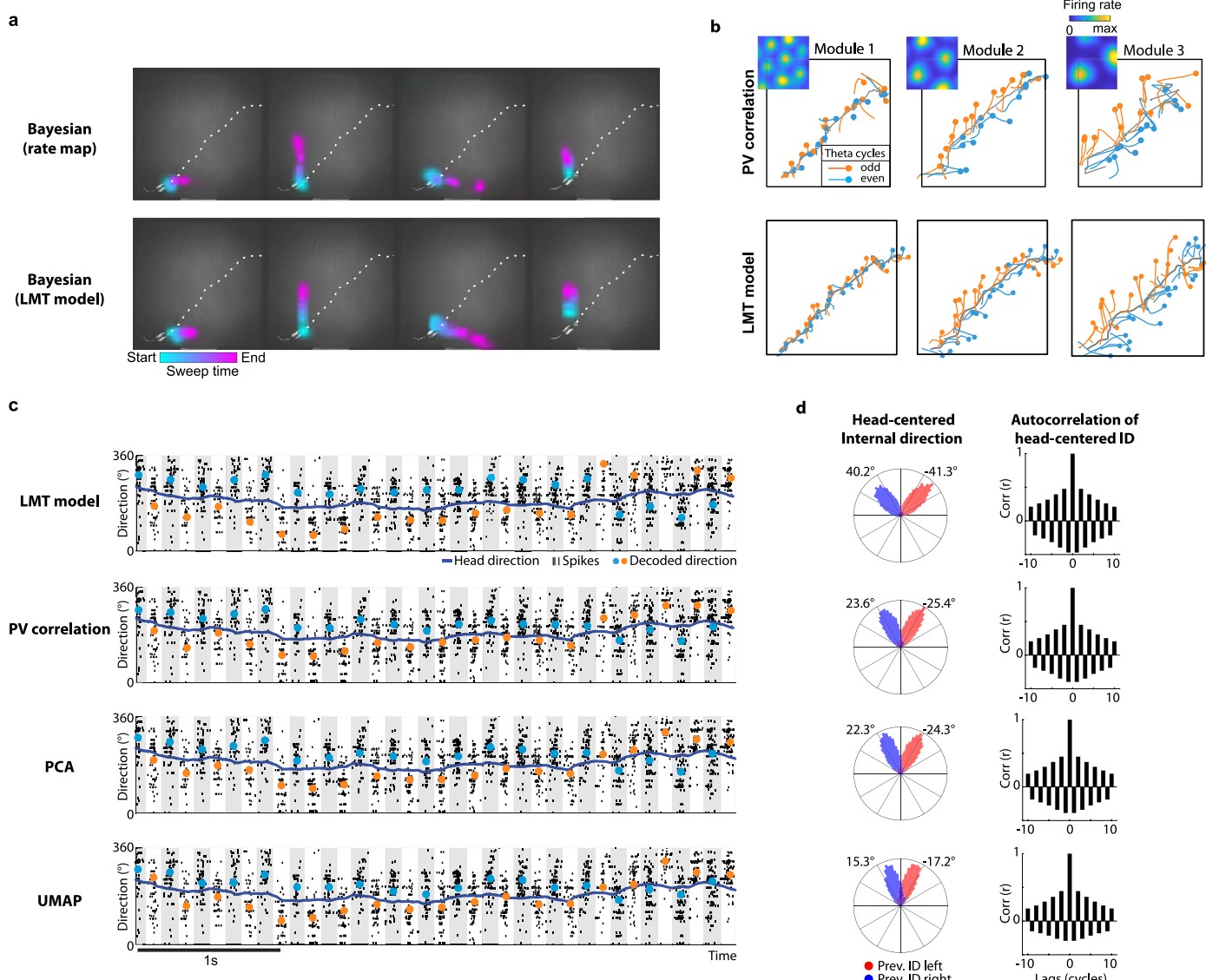

**Extended Data Fig. 4 | Alternating sweeps and directional signals can be decoded with several methods. a**, Example sweeps decoded from all co-recorded MEC/PaS cells (same time period as Fig. 1a) using different decoding methods. The Bayesian decoder was based on either standard rate maps (top row) or the fitted tuning curves from the latent manifold tuning model (bottom row; Fig. 4, Extended Data Fig. 10). Sweeps are similar with all methods but can be extracted with higher fidelity by the LMT model. **b**, Single-module sweeps extracted by PV-correlation decoding (top; as in Fig. 1e) and by the LMT latent variables (bottom). Note that the two methods yield similar results. **c**, Left: alternating sequences in decoded internal direction are similar regardless of

the decoding method used. In each plot, rows of black ticks indicate spike times from 533 rhythmic directional cells, with cells vertically sorted by preferred direction (as in Fig. 2c). Coloured dots show the extracted population signal using 4 different decoding methods (colours indicate odd and even-numbered theta cycles). Note that all methods extract a left-right alternating directional signal that follows the angles of the packets of spiking activity. **d**, Left: circular distribution of head-centred internal direction (as in Fig. 2d). Right: auto-correlograms of internal direction (as in Fig. 2e) for each of the decoding methods.

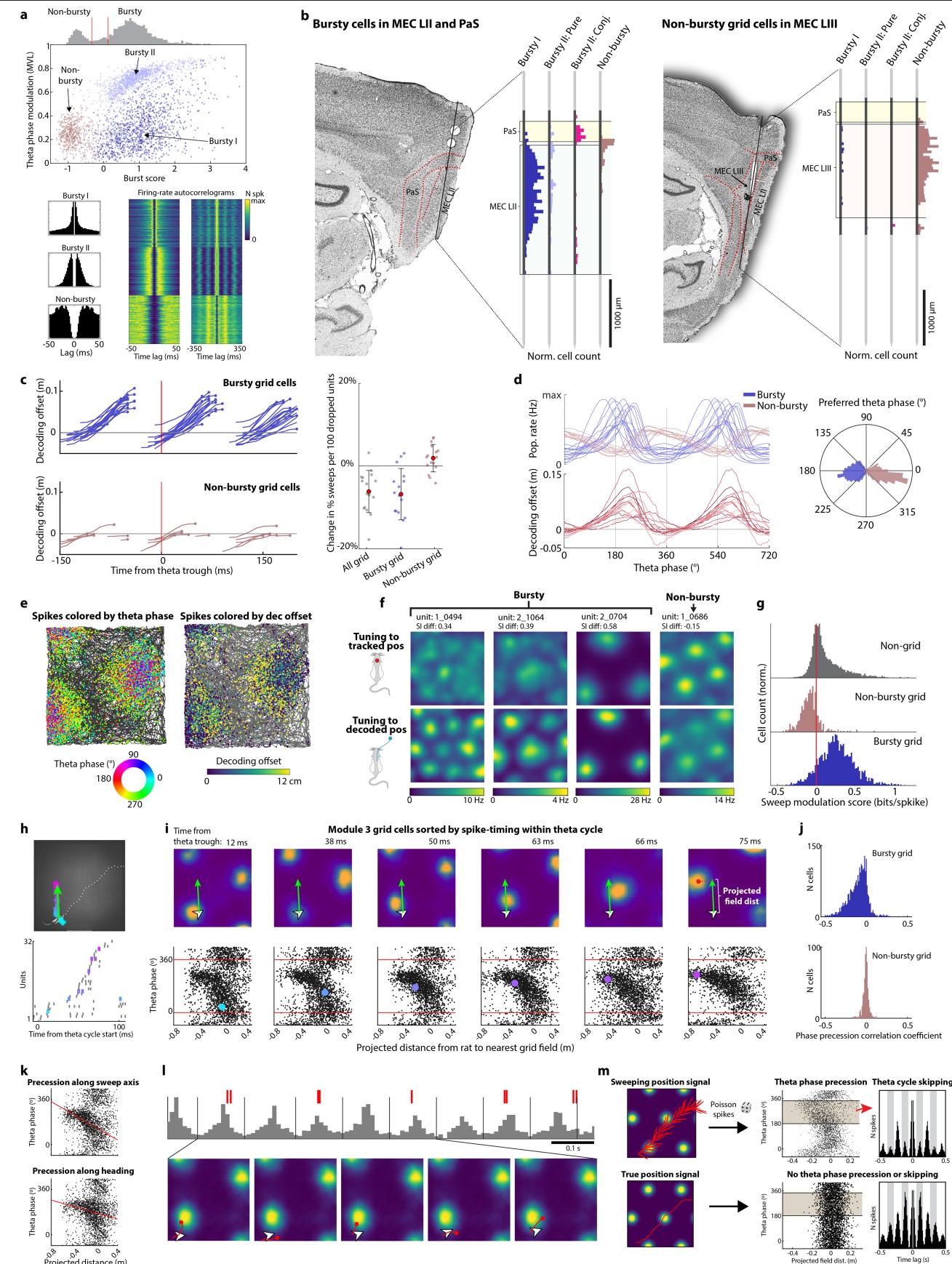

**Extended Data Fig. 5** | See next page for caption.

**Extended Data Fig. 5 | Bursty MEC layer II grid cells are strong carriers of the sweep signal. a**, Three subclasses of grid cells with distinct temporal dynamics are revealed by clustering of firing-rate temporal autocorrelograms. Top: scatter plot showing burst scores (Methods) and theta phase modulation for all recorded grid cells (dots). Note three separable clusters of grid cells. Two of the subclasses fire in bursts ('bursty I' and 'bursty II') and can be dissociated from each other based on autocorrelogram shape and theta modulation. 'Bursty I' grid cells include non-directional 'pure' grid cells in MEC layer II. 'Bursty II' grid cells include most conjunctive grid cells in parasubiculum and some non-directional 'pure' grid cells in MEC layer II. The third subclass ('non-bursty') does not fire in theta-rhythmic bursts. Histogram above scatter shows bimodal distribution of burst scores in the cell sample, with thresholds for binary classification (bursty or non-bursty) indicated by red vertical lines (cells with burst scores between the two lines were unclassified). Bottom: one example autocorrelogram (±50 ms) from each grid cell subclass is shown to the left. Middle and right: stacked autocorrelograms (±50 ms and ±350 ms) from 500 example grid cells in each subclass. Firing rate is colour-coded. Cells are sorted by cluster identity (top to bottom). Note prominent theta-rhythmic activity in bursting grid cells. **b**, Anatomical segregation of grid-cell subclasses (see **a** for reference). Left: histological section from rat 25843 with electrodes in MEC layer II and parasubiculum (labelled 'PaS'). The proportions of recorded cells belonging to each grid-cell subclass at each location along the probe are shown as vertical histograms. Bursty II grid cells with conjunctive direction tuning (labelled 'Conj.') are plotted separately from pure bursty II grid cells. Note predominance of bursty grid cells in MEC layer II and parasubiculum. Right: histological section from rat 25953 with electrodes in MEC layer III and PaS. Note predominance of non-bursty grid cells in MEC layer III. Based on inspection of probe tracks in 14 animals, 84.3% or 2,316/2,747 of bursty I-II grid cells (excluding conjunctive grid cells) were found to be in MEC layer II, 76.5% or 724/946 of non-bursty cells were in MEC layer III, and 92.7% or 1,240/1,337 of conjunctive grid cells were in parasubiculum (data from 2-7 sessions per animal). **c**, Left: progression of sweeps decoded separately from bursty grid cells (top) and non-bursty grid cells (bottom) visualized by plotting the offset between decoded position and lowpass-filtered trajectory projected onto the rats' head axis and averaged across all theta cycles in each session. Position was decoded from a random subsample of 100 bursty or non-bursty grid cells in all sessions with more than 100 co-recorded bursty grid cells (19 sessions from 10 animals, 1-3 sessions per animal) or non-bursty grid cells (5 sessions from 3 animals, 1-3 sessions per animal). Each line corresponds to one session. Sweeps are more prominent when decoding from bursty grid cells. Right: decrease in number of detected sweeps when omitting all grid cells or individual grid-cell subclasses from the decoder input data (all 16 animals, 1 session per animal, plotted as individual dots). Values are normalized by comparing the number of sweeps obtained when omitting a size-matched random selection of non-grid cells (repeated 100 times per session) and dividing by the number of dropped cells. Means and s.d. are shown as red dots and whiskers. Excluding all grid cells (47–650 cells) or bursty grid cells (27–567 cells) results in a significant drop in the number of detected sweeps (all grid cells: −6.2 ± 0.3% per 100 dropped units, p = 0.0008; bursty grid cells: −6.8 ± 0.4%, p = 0.002, Wilcoxon signed rank test, two-tailed). Excluding non-bursty grid cells (3-244 cells) did not result in a significant change in the number of detected sweeps (+1.9 ± 0.2%, p = 0.12). **d**, Bursty grid cells are maximally active when decoded position sweeps outwards. Top: firing rate for bursty and non-bursty grid cells during the course of the theta cycle (summed across all co-recorded cells, one line per session). Bottom: offset between decoded position and lowpass-filtered decoded position as a function of theta phase (each line corresponds to one animal). Right: polar histogram of preferred firing phase for all bursty and non-bursty grid cells. Note that bursty and non-bursty grid cells are active at opposing phases of the theta cycle[32,91]. **e**, Out-of-field spikes in bursting grid cells coincide with outgoing sweeps. Panels show firing locations for an example grid cell with dots corresponding to the animal's location at the time of individual spikes and colour corresponding to theta phase (left) or offset between decoded position and tracked position (right). Note that out-of-field spikes occur at phases where the decoded position sweeps ahead of the animal (180–270 deg with 0 deg as the phase of minimum activity). **f**, Spatial receptive fields of bursty grid cells can be sharpened by accounting for sweeps. Example

rate maps plotted with respect to tracked position (top row) and rate maps with respect to decoded position (bottom row) for three bursty grid cells and one non-bursty grid cell (columns). Note that for bursty cells, grid patterns are clearer and sometimes only visible when activity is plotted as a function of decoded position, which sweeps outwards from tracked position, indicating that the cells are tuned to the position of the sweep rather than the animal's tracked location. Difference in spatial information content ('SI diff') between the original and corrected rate maps is noted above each cell and indicates the strength of sweep-modulation. **g**, Sweep-modulation is primarily associated with grid-cell identity. Sweep modulation scores (difference in spatial information content between the original and corrected rate maps) are higher for bursty grid cells than for non-bursty grid cells (sweep modulation score for bursty grid cells: 0.27 ± 0.23, mean ± s.d. for n = 2,766 cells; non-bursty grid cells: −0.08 ± 0.13 (n = 460 cells); non-grid cells, i.e. all putative excitatory cells excluding grid cells; 0.14 ± 0.23 (n = 9,702 cells); 16 animals, 1 session per animal; two-tailed Mann-Whitney U = 31.5 p < 0.001 (2.1e-218) for bursty vs. non-bursty cells and Mann-Whitney U = 28.8 p < 0.001 (5.5e-182) for bursty vs. non-grid cells). **h**, Top: decoded position (colored blobs) and internal direction (green arrow) during a single theta cycle. Bottom: spikes emitted from 32 grid cells (grid module 3) during the same theta cycle. Cells are sorted based on their time of spiking within the theta cycle. **i**, Top row: session-averaged rate maps for six grid cells that spiked during the theta cycle in **h** (spikes of these cells highlighted by thicker ticks in bottom panel of **h**). Cells are sorted on their time of spiking within the theta cycle, with time of first spike during the theta cycle indicated above. White arrowhead indicates animal's position; green arrow shows internal direction (as in **h**). Note that grid cells with firing fields progressively further away from the animal are activated in sequence throughout the theta cycle. Bottom row: phase precession curves for the six grid cells shown in the top row. Black dots show spikes from the whole session (2000 random subsamples). To measure 2D phase precession, the distance from the rat to the nearest grid field was projected on to the internal-direction axis. The plots show theta phase as a function of this distance for successive spikes of individual cells. Note that all cells show reliable phase precession along the path of the sweep. Coloured dots show the first spike emitted by each cell during the theta cycle shown in **h**). **j**, Characterization of phase precession along the sweep axis in all grid cells across animals during open field foraging (3,194 cells from 16 animals). Histograms show circular-linear correlation coefficients for phase vs. position[73], with negative correlation coefficients indicating reliable phase precession. Note that phase precession is more prevalent in bursty grid cells than in non-bursty grid cells (significant precession in 1,635/2,465 or 66.3% of bursty grid cells and 83/569 or 14.6% of non-bursty grid cells; mean ± s.d. correlation coefficients of −0.10 ± 0.12 for bursty and −0.01 ± 0.04 non-bursty, two-tailed Mann-Whitney U = −22.5 p = 9.7e-112 for bursty vs. non-bursty grid cells). **k**, Phase precession of an example grid cell during open field foraging when field distance is projected along sweeps (top) or along the animal's movement direction (bottom). Regression line is shown in red. Note that precession is visible in both cases, but that the precession pattern is clearer along sweep direction. **l**, Left-right-alternating sweeps correlate with theta cycle skipping in individual grid cells. Top: population firing rate (grey bars) and spikes from an example grid cell (red ticks) over a period of nine theta cycles (separated by vertical lines). Note consistent cycle-skipping in the example grid cell during the first eight cycles (red ticks in every other cycle). Note also gradual phase precession of the example cell's spikes with respect to population theta. Bottom: the animal's position (white arrowhead) and decoded sweeps (red lines) during five of the theta cycles shown in top row. Rate map for the grid cell shown in background. Note that cycles with spikes correspond to cycles where the sweep goes through one of the grid fields. **m**, Phase precession and theta cycle skipping arise in simulated grid cells that encode a sweeping position signal. The activity of an idealized grid cell (left column) was predicted using a sweeping position signal (top left; rat's running trajectory with sweep-like spatial offsets in alternating directions). This results in reliable phase precession (top centre; second half of theta cycle is highlighted) as well as theta cycle skipping during the second half of the theta cycle (top right). Phase precession and cycle skipping did not emerge when the animal's running trajectory was used as input (bottom row). Credit: rat, scidraw.io/Gil Costa.

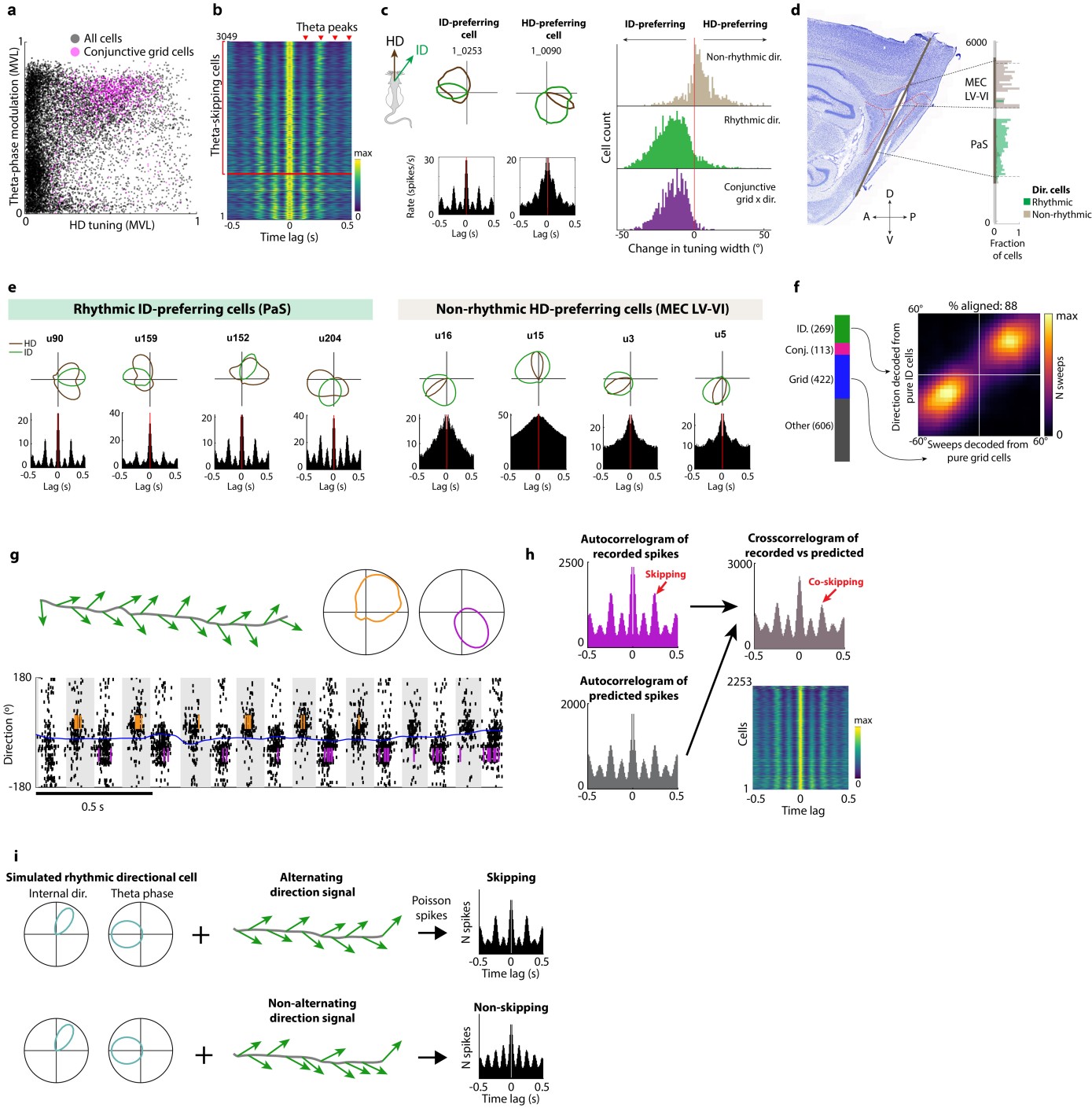

**Extended Data Fig. 6** | See next page for caption.

**Extended Data Fig. 6 | The internal direction signal is expressed by a discrete population of cells in parasubiculum. a**, Left: scatter plot of head direction (HD) and theta-phase tuning strength (mean vector length, MVL) for all recorded cells (12,300 cells, 16 animals, 1 session per animal). Each dot corresponds to one cell. Conjunctive grid cells are plotted in pink (23.3% or 848/3,632 of all direction-tuned cells were conjunctive grid cells). Note broad range of directional tuning and high proportion of theta-rhythmic cells (83.9% or 3,049/3,632 of all direction-tuned cells and 93.6% or 794/848 conjunctive grid cells were significantly modulated by theta oscillation phase). **b**, Stacked firing-rate autocorrelograms (±500 ms), colour-coded by firing rate, for all theta-modulated direction-tuned cells, sorted by tendency to fire on alternating theta cycles. Lags corresponding to repetitions of the ~8 Hz theta rhythm are indicated with red arrowheads. Cells above the red horizontal line have positive scores and are thus classified as 'skipping' cells. Note that most cells (73.9% or 2,253/3,049) display theta skipping, with prominent peaks at lags corresponding to every other theta cycle. Out of all 3,632 direction-tuned cells, 2,253 or 62.0% were both theta-modulated and cycle-skipping, 796 cells or 21.9% were non-skipping and theta-modulated, and 583 cells or 16.0% were non-skipping and non-rhythmic. **c**, Left: Directional tuning curves with respect to head direction (HD, brown) and decoded direction (green) for two example cells. Bottom: temporal autocorrelogram of cells' firing rates. The theta-rhythmic cell is sharply tuned to the decoded internal direction (ID) signal, while the non-rhythmic cell is more strongly tuned to HD. Right: histograms showing relative tuning width to HD vs ID from cells with significant tuning to either HD or ID (4,628 cells from 16 animals). Theta-rhythmic directional cells are more sharply tuned to ID than HD (90.5% or 3,527/3,897 of rhythmic directional cells were ID-preferring; difference in tuning width, defined as two standard deviations of the tuning curve, to ID vs HD: −15.4 ± 13.4 deg, mean ± s.d., p < 2.2e-16 Wilcoxon signed rank test, two-tailed), while non-rhythmic cells are more strongly tuned to HD (73.4% or 408/556 of non-rhythmic directional cells were HD-preferring; difference in tuning width: 4.8 ± 12.1 deg, p = 0.0004, Wilcoxon signed rank test, two-tailed). Conjunctive grid cells were mostly ID-preferring (96.5% or 950/985, tuning width difference: −14.9 ± 9.5 deg, mean ± s.d., p < 2.2e-16, Wilcoxon signed rank test). This indicates that there are separate populations of directional cells that encode distinct signals. **d**, Sagittal histological section from one example rat (out of 17 rats), 'rat 28304', with recording sites in parasubiculum and MEC layer V-VI, showing the track of a probe shank that went through deep layers of MEC before entering parasubiculum (PaS). Right: density of rhythmic and non-rhythmic directional cells along the probe shank (proportion of all cells at corresponding depths; data from 5 recording sessions in rat 28304 with different site configurations). Note the high proportion of non-rhythmic directional cells in MEC layer V-VI. **e**, Four example rhythmic directional cells recorded from parasubiculum (left) and four non-rhythmic directional cells recorded from the deep layers of MEC (right) in the animal shown in **d**. Tuning curves and autocorrelograms are plotted as in **d**. Note that the rhythmic cells are more strongly tuned to ID, while the non-rhythmic cells are sharply tuned to HD (resembling classical HD cells). **f**, Alternating sweep and direction signals are expressed by separate neural populations. Left:

stacked bar chart showing the number of functional cells recorded in an example animal (25843). Right: heat map showing alignment between direction of sweeps decoded from pure grid cells (n = 422) and internal direction decoded from simultaneously recorded pure internal direction ('ID') cells (n = 269), both expressed in head-centered coordinates as in Fig. 2g. Conjunctive grid cells were excluded from the decoder input data. Decoded sweep and direction signals remain aligned (both left or both right) in 88.1% of theta cycles where sweeps and direction signals were detected (circular correlation between sweep direction and internal direction signals: r = 0.84, p < 2.2e-16), suggesting that the observed dynamics is not caused by using the same cells to decode position and direction. **g**, Theta skipping in individual internal direction cells coincides with left-right-alternation of the internal direction signal. Top left: running trajectory (grey) and decoded internal direction (green arrows) during a brief running epoch. Top right: internal direction tuning curves for two example cells that were active during the epoch shown to the left. Bottom: raster showing population activity of all internal direction cells (black ticks) during the epoch shown in top left. Coloured ticks show spikes emitted by the two example cells. Note that the two cells fire in alternation when the animal runs along the indicated trajectory (on theta cycles where the internal direction signal aligns with the preferred direction of each cell). **h**, Theta skipping in individual internal direction cells can be predicted from left-right-alternation of the internal direction signal. Top left: firing-rate autocorrelogram for the internal direction cell shown in top right of panel **g**. Bottom left: firing-rate predictions for the cell was generated based on its tuning curves for internal direction and theta phase, along with decoded internal direction and measured theta phase. A Poisson spike train was drawn from the firing-rate prediction. The autocorrelogram of the prediction-based spike train is plotted and shows cycle skipping, as in the recorded cell. Top right: crosscorrelation of recorded spikes and prediction-based spikes. Note strong peaks every second theta cycle, implying that the predicted activity alternates in synchrony with the recorded activity. Bottom right: crosscorrelation of recorded spikes and prediction-based spikes for all theta skipping directional cells (2,253 cells from 16 animals). 'Co-skipping' between predicted and recorded spikes was present in the majority of theta skipping directional cells (1,768/2,253 or 78% of total), implying that a large fraction of the observed theta skipping in internal direction cells can be explained by the left-right-alternating dynamics of the internal direction population signal. **i**, Emergence of theta skipping in simulated rhythmic direction cells tuned to a left-right-alternating direction signal. Left: internal direction tuning and theta phase tuning for a simulated cell. Middle: the activity of the simulated cell was predicted given an alternating direction signal (top) or a non-alternating direction signal (bottom), computed by adding ± 30 degree offsets to the animal's head direction in an alternating or non-alternating pattern. Right: firing-rate predictions were made based on tuning curves, theta phase and simulated direction signals and Poisson spikes were drawn from the firing-rate prediction. Autocorrelograms of the simulated spike trains are plotted, with noticeable theta skipping for the left-right-alternating direction signal (top) and absence of skipping for the randomly shifting direction signal (bottom). Credit: rat, scidraw.io/Gil Costa.

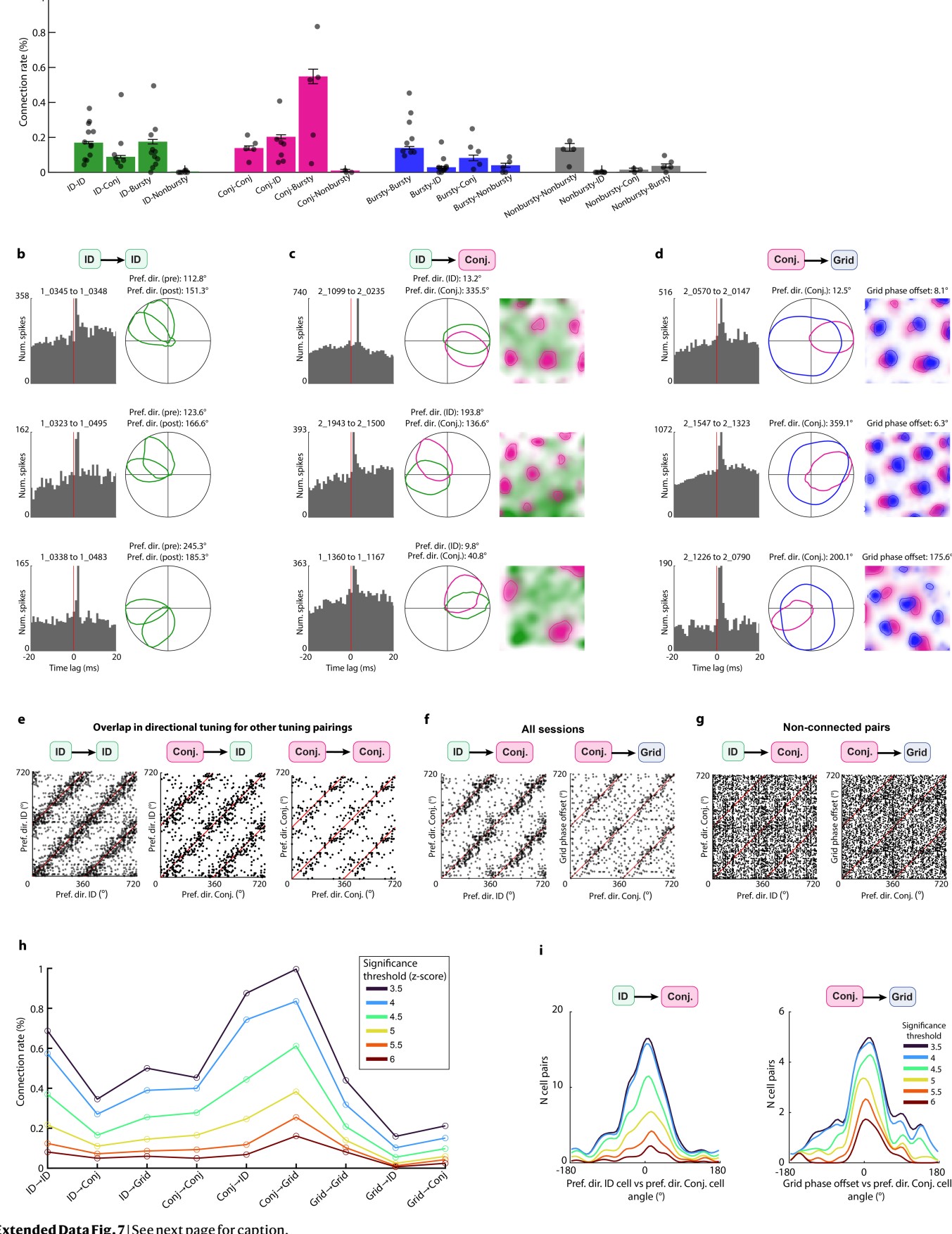

**Extended Data Fig. 7** | See next page for caption.

**Extended Data Fig. 7 | Connectivity between functional cell types.**
**a**, Estimated connection probability (in %, n = 16 animals, 1 session per animal) for projections originating from each of four functional classes (internal direction ('ID'), conjunctive grid ('conj'), bursty pure grid ('bursty') and non-bursty pure grid ('non-bursty'). Dots show connection rates in individual animals (only plotted for animals and pairings with >1500 total cell pairs: n = 14, 6, 11, 8, 6, 4, 6, 2, 11, 6, 9, 5, 8, 2, 5 and 4 animals for each category, left to right). Bars and whiskers show means and standard deviations of estimated binomial distributions (data pooled across 16 animals). Note connections from internal direction cells to conjunctive grid cells and from conjunctive grid cells to pure grid cells, in addition to recurrent connections within each class. Also note that internal direction cells and conjunctive grid cells have putative connections to bursty pure grid cells, but not non-bursty pure grid cells (connection rates for direction-tuned to bursty: 237/102,786 or 0.23%; ID to non-bursty: 3/55,900 or 0.003%; conjunctive to bursty: 183/28,387 or 0.645%; conjunctive to non-bursty: 2/18,085 or 0.011%). Although estimated connection rates varied from animal to animal, the general pattern of connections was preserved. **b**, Examples of putative recurrent connections between internal direction (ID) cells. Each of the three rows shows firing rate cross-correlogram (left column) and directional tuning curves (right column) for an example pair of putatively connected ID → ID cells. Preferred firing direction (PFD) of each cell is indicated above directional tuning curves. Note similar directional tuning between connected cells. **c**, Three example cell pairs illustrating putative connections between ID-tuned cells (green) and conjunctive grid cells (pink). Plotted as in Fig. 3b. **d**, Three example cell pairs showing putative connections between conjunctive grid cells (pink) and pure grid cells (blue). Plotted as in Fig. 3e. **e**, Alignment of directional tuning for other putatively connected cell pairs. Scatter plot shows preferred directions of pairs of pre- and postsynaptic cells, plotted as in Fig. 3c. Note that all combinations of connected direction-tuned cells have similar directional tuning (angle between preferred directions: 12.1 ± 58.0 deg, circular correlation: r = 0.49, p < 2.2e-16 for 707 putative ID → ID pairs; angle: 4.6 ± 58.2 deg, circular correlation: r = 0.48 p = 5.7e-13 for 322 putative conjunctive grid → ID pairs; and

angle: 11.1 ± 58.0 deg, circular correlation: r = 0.52 p = 5.5e-5 for 130 putative conjunctive grid → conjunctive grid pairs). **f**, Alignment of directional tuning is present across all recording sessions. Left: preferred directions for all pairs of connected ID cells to conjunctive grid cells (angle between preferred directions: 8.4 ± 51.6 deg, circular correlation: r = 0.56 p = 2.2e-16, n = 341 pairs from 40 sessions, 16 animals). Right: preferred direction of pre-synaptic grid cells and direction of grid phase offset between pre- and postsynaptic cell for all pairs of putatively connected conjunctive grid cells and 'pure' grid cells (angle: −0.7 ± 64.1 deg, circular correlation: r = 0.35 p = 1.4e-6, n = 213 pairs from 40 sessions). Each dot corresponds to one pair of cells. **g**, Left: tuning directions for randomly selected pairs of non-connected ID cells and conjunctive grid cells (1,600 pairs, 100 pairs per animal). Absolute angles between tuning directions were significantly smaller for connected ID → conjunctive grid pairs than for randomly selected cell pairs (mean absolute offset: 42.0 deg vs. 84.6 deg (chance 90 deg), p = 7.4e-17, two-tailed Mann-Whitney U test). Right: Tuning directions for randomly selected pairs of non-connected conjunctive grid cells and pure grid cells (1,600 pairs, 100 pairs per animal). Absolute angles between preferred directions were significantly smaller for connected conjunctive grid → pure grid cell pairs than for randomly selected cell pairs (mean absolute offset: 34.1 deg vs. 91.5 deg, p = 1.5e-11, Mann-Whitney U test). **h**, Connection rates between functional cell types estimated with different significance thresholds for identification of monosynaptic connections. Thresholds are specified in terms of standard deviations from the baseline firing-rate cross-correlograms. While connection probabilities are heavily dependent on detection thresholds (average connection probability ranging from 0.04% to 0.4%), the connection probability ratios are fairly stable. **i**, Alignment of tuning relationships of connected cell pairs is stable across significance thresholds (z-score of convolved cross-correlogram peak) for detecting monosynaptic connections. Left: directional tuning in ID → conjunctive grid cell pairs (mean angles: 12.7–16.3 deg, correlation coefficients 0.35–0.68); right: directional-tuning vs. grid-phase offset angle in conjunctive grid → pure grid cell pairs (mean angles between preferred directions: −2.0 to −0.4 deg, correlation coefficients 0.34–0.79).

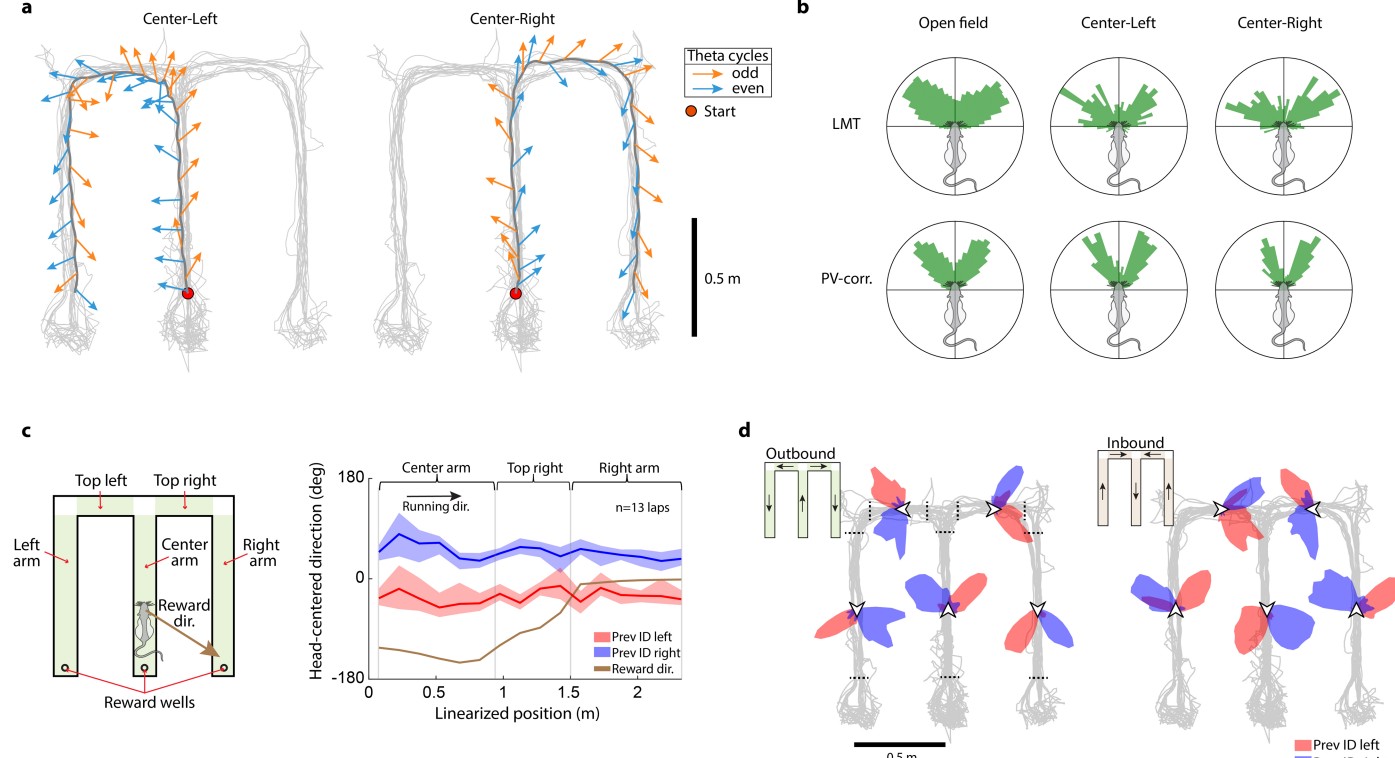

**Extended Data Fig. 8 | Left-right-alternating internal direction during goal-directed navigation. a**, Internal direction signals on an m-shaped spatial alternation task[10]. Panels show inferred internal direction (LMT based on MEC-parasubiculum population activity) on two outbound trials (left: outbound left; right: outbound right). Alternate theta cycles are plotted with different colours. The running trajectory from the trial segment is plotted in black, while the full trajectory across all trials is plotted in light grey. **b**, Polar histograms of internal direction inferred by LMT (top row) or decoded based on head-direction tuning curves (bottom row) during open field foraging (left column), on the central stem of the m-shaped maze preceding a left turn (middle column) or on the central stem preceding a right turn (right column). Same recording as in **a**. Note that the angular distributions look similar across conditions, indicating that internal direction is not biased towards the to-be-chosen path. **c**, Internal direction is not biased towards the current goal. Left: schematic of m-maze indicating maze segments and reward wells. Outbound running trajectories

were linearized by defining centre well and side well as beginning and end, respectively. Reward direction is defined by the vector from the rat's head to the to-be-chosen reward well. Right: internal direction averaged across all outbound right laps (n = 13 laps; same recording as in **a**). Solid lines show mean head-centred internal direction (inferred by LMT) following a left (red) or right (blue) internal direction in the preceding theta cycle. Shading shows 95% bootstrapped confidence intervals of the mean. Brown line shows the head-centred reward direction. Transitions between maze segments are indicated with vertical lines. Note that internal direction on alternating cycles is stable across the entire run despite substantial changes in reward direction. **d**, Polar histograms of head-centred internal direction in successive maze segments (centre arm, top arm, left or right arm) on outbound trials (left) and inbound trials (right). Note bimodal distribution of internal direction during all task phases. Credit: rat, scidraw.io/Gil Costa.

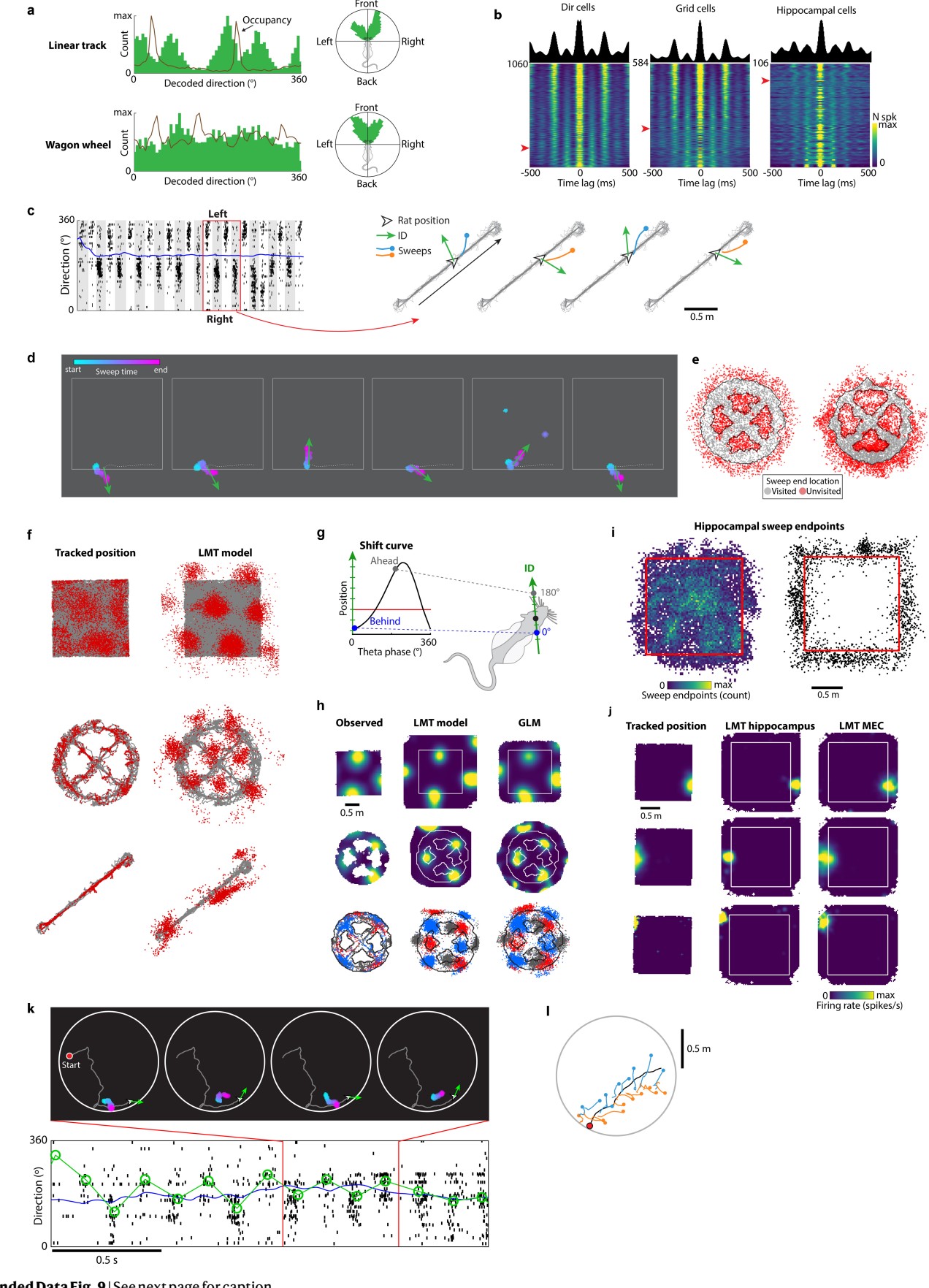

**Extended Data Fig. 9** | See next page for caption.

**Extended Data Fig. 9 | Sweeps extend into never-visited space and persist in novel environments and during darkness. a**, Left: histogram of decoded directions (green bars) and head-direction occupancy (solid line) during an example linear-track session (top) and wagon-wheel session (bottom). Although the occupancy on the linear track is biased to the axes of the track (visible as peaks), the sampling of other directions was sufficient for subsequent decoding of the direction to either side of the track. Right: circular histogram of decoded directions in head-centered coordinates. Note that the internal direction signal is bimodally distributed around the head axis also in 1D environments. **b**, Theta cycle skipping during linear-track running. Heat maps show firing-rate temporal autocorrelograms (±500 ms) from all internal direction cells, grid cells and hippocampal cells (left to right). Each row corresponds to one cell; cells are sorted by theta-skipping index. Cells above the red arrowhead have positive scores and are thus classified as 'skipping' cells. One example autocorrelogram from a skipping cell is shown above each plot. The presence of cycle skipping in this task indicates representation of alternating directions and locations, incompatible with coding for the running path, but compatible with representation of unvisited space on either side of the track. **c**, Alternating internal direction and sweeps during linear-track running extracted from fitted latent variables. Left: raster plot showing spike times of internal direction cells (sorted by preferred internal direction) during a lap on the linear track. Right: decoded sweeps (lines) and internal direction (arrows) during four successive theta cycles during the lap (indicated by red square in left panel). Note that sweep and direction signals point to the same side of the track in an alternating pattern. **d**, Decoded direction and sweeps from consecutive theta cycles during a period of running along one of the open-field walls (white square). Sweeps and decoded direction are plotted as in Fig. 4d, with colour indicating time within sweep. Note that sweeps travel through the opaque walls of the arena, in alignment with the internal direction signal. **e**, Sweeps span a 2D map even when navigation is confined to 1D paths. Scatter plot of sweep terminal positions during two recording sessions on the WW maze (left panel: rat 25843; right panel: rat 25691). Note that density of sweep terminal positions is similar for visited and unvisited locations. **f**, More examples of grid-cell tuning to unvisited locations in the three environments (top to bottom: open field, wagon wheel, linear track). Each column corresponds to one cell and shows the position of the rat at the time of each spike (left) or the latent position from the LMT model at the time of each spike (right). **g**, Because the LMT model, like other dimensionality-reduction methods, finds dense representations of the input data, in principle, grid-like tuning could emerge as a close-packing artifact during fitting. As a control, an alternative single-cell model was used to infer out-of-bounds tuning for each cell independently. The activity of each cell was fitted by a GLM-based model in which the animal's recorded position (black dot) was shifted along the axis parallel to internal direction ('ID') as a function of theta phase, according to a 'shift curve' which was fitted separately for each cell (see Methods for details). **h**, Firing-rate maps for a grid cell with respect to

tracked 2D position (left), inferred by the LMT model (middle) and by the GLM-based single-cell model described in **g** (right) during an open field experiment (top row) and on the wagon wheel track (middle row). Each plot in the bottom row shows the positions of spatial receptive fields (coloured patches) of three co-recorded cells on the wagon wheel track. Spikes from each cell is plotted with different colours. Note that the latent fields identified by the LMT and GLM models preserve similar phase offsets between the grid cells, as expected based on extrapolation of their grid patterns. **i**, Hippocampal sweeps extend into never-visited locations. Left: heatmap of sweep end positions as inferred by the LMT model when fitted to hippocampal data from one example session. Note that many sweeps terminate outside the confines of the open field arena (red box). A total of 17.6 ± 1.3% (mean ± s.e.m.) of hippocampal sweeps in 5 rats terminated outside the open arena (red box). Right: hippocampal sweep endpoints (black dots) during theta cycles where sweeps from co-recorded MEC-parasubiculum cells terminated outside the open field arena (red box). Out-of-bounds hippocampal sweeps were detected in the majority (65.7 ± 6.4%, mean ± s.e.m.) of theta cycles in which simultaneously decoded MEC sweeps terminated outside the open field arena, significantly more often than during the preceding or following theta cycle (difference in fraction of outside sweeps: 11.6 ± 2.7%, p = 0.031, two-tailed Wilcoxon signed-rank test). **j**, Place-cell maps include never-visited space. Plots show firing-rate maps for three hippocampal place cells during an open-field session based on (left) original tracked position, (middle) latent position extracted from hippocampal activity, and (right) latent position extracted from co-recorded MEC-parasubiculum cells. Note that the two LMT models infer similar place fields outside the walls of the arena. **k**, Left-right-alternating sweeps during the first traversal through a novel environment. Top: decoded sweeps and internal direction during four consecutive theta cycles after 7 s of exploration of a novel circular open field (in complete darkness). Tuning curves from a succeeding recording session with room lights on (same arena) were used to decode position and direction in the novel, dark condition. The rat's current position (white arrowhead) and running trajectory from the beginning of the session (grey line) is also shown. Note substantial spatial offset between the rat's position and the left-right-alternating sweeps, which may reflect inaccuracies in the animal's self-location estimate due to the novel and sensory-deprived conditions. Bottom: spikes from internal direction cells (black ticks), decoded internal direction (green) and tracked head direction (blue) during a 2-second epoch after 6 s of exploration. Note that left-right-alternating internal direction signals are expressed during the first traversal of the novel environment. **l**, Sweeps extracted from the LMT position latent variable during the novel open field session shown in k), during a 2.5-second epoch after 140 s of exploration (still in complete darkness). Alternating sweeps are visible, although the decoded position still deviates substantially from the rat's actual running trajectory (in black), as in **k**. Credit: rat, scidraw.io/Gil Costa.

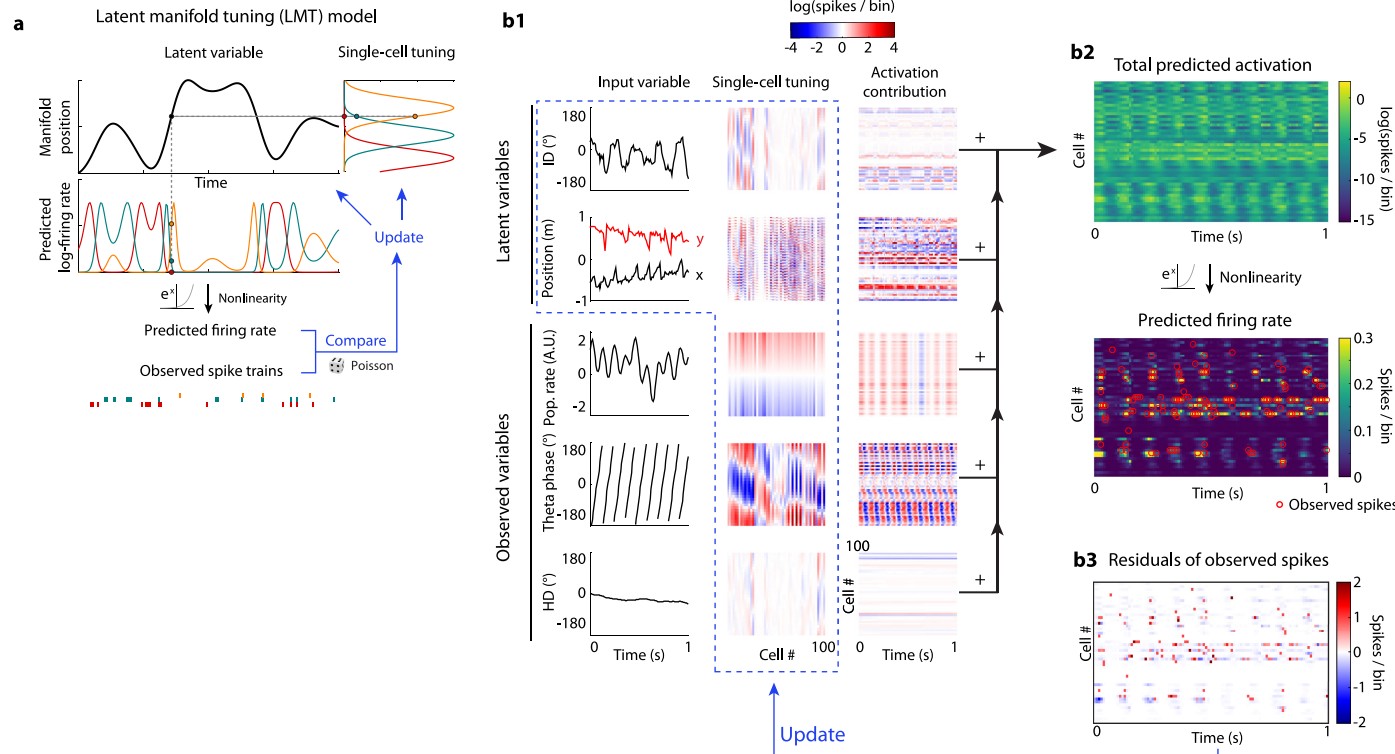

**Extended Data Fig. 10 | Latent variable model. a,** Illustration showing the basic principles of using the latent manifold tuning (LMT) model to extract a latent signal from neural population activity. A latent variable (top, black curve) evolves smoothly in time on a one-dimensional manifold. Individual cells are tuned to specific locations on the manifold (top right). At each point in time, the latent variable value predicts each cell's log-firing rate (second row), which is then transformed with an exponential nonlinearity into a firing-rate prediction. The latter is compared with the observed spikes fired by the cells, treating the spike counts as a Poisson process. The model is learned by iteratively optimizing the latent variable and the tuning curves to improve the prediction of the observed spikes. **b,** Schematic showing the design of the complete 'composite' model. **b1,** Neural activity is modelled as a function of five input variables (first column). The two latent variables of interest (internal direction ('ID') and 2D position; first two rows), are fitted while regressing out contributions of three observed variables known to modulate MEC neural activity (theta phase, HD and population firing rate; three bottom rows). Left column: example traces of the input variables. All variables are assigned with corresponding tuning for each cell (second column; 100 example cells are shown), which, in conjunction with the latent variable's value, predicts each cell's log-firing-rate at each time point (third column). The log-firing-rate predictions are linearly summed across all variables (**b2**, top), then the sum is exponentiated, yielding a net prediction of the population firing rates (bottom). For reference, observed spikes for each cell are overlaid (red circles). **b3,** The unaccounted-for ('residual') neural activity is calculated by subtracting the predicted firing rates from the observed firing. At each iteration of fitting the model, the residuals are used to calculate the next update to the latent variables and the tuning curves (enclosed by blue dashed box), leading to gradual improvement in the match between predicted and observed spiking.

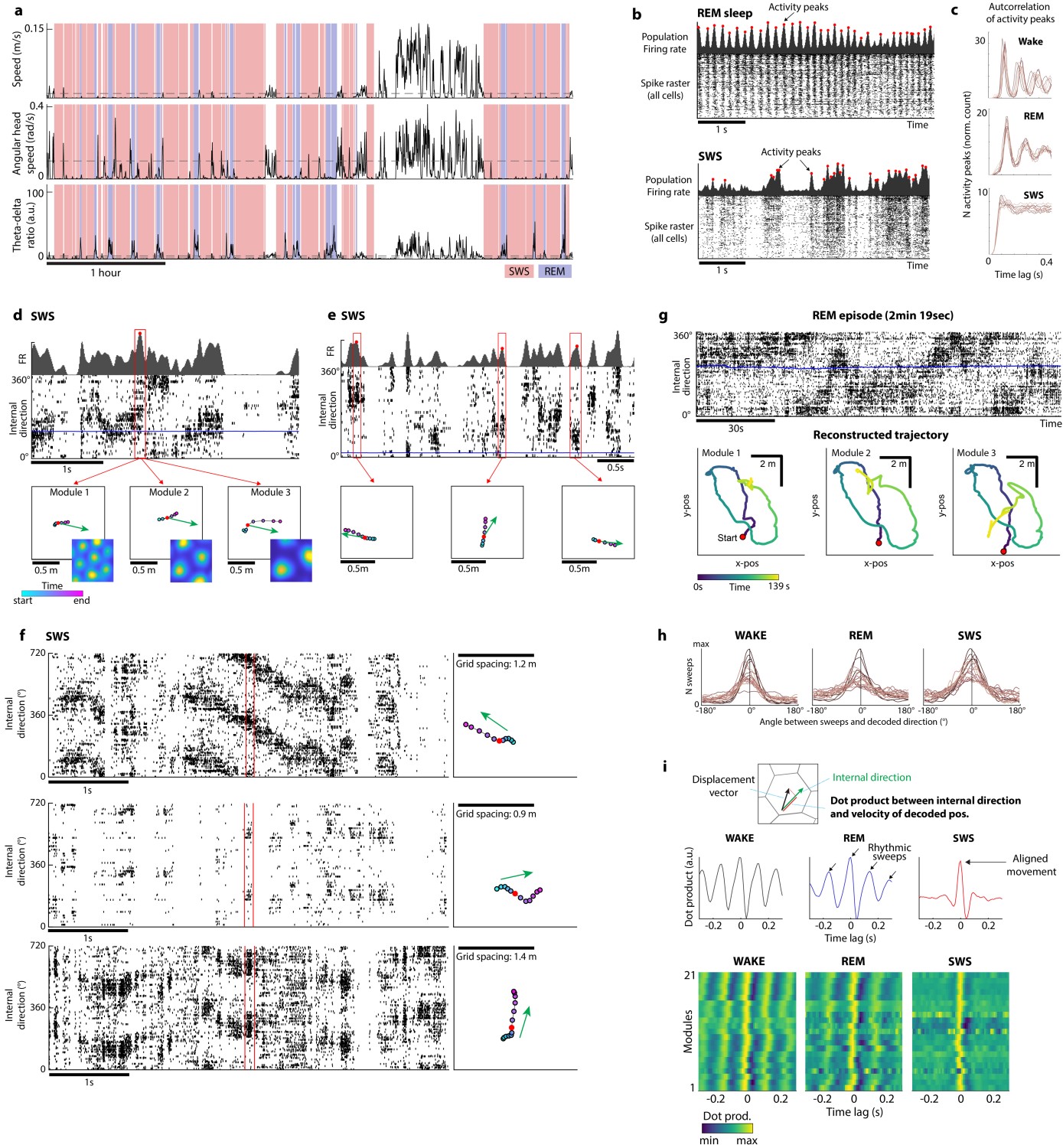

**Extended Data Fig. 11** | See next page for caption.

**Extended Data Fig. 11 | Internal direction and sweeps during sleep. a**, Sleep classification based on movement parameters and electrophysiological signatures, shown here for one example recording. Panels show head speed (top), angular head speed (middle) and the theta/delta ratio of multi-unit activity (bottom) over the course of a recording session (scale bar, 1 h). We used these parameters to identify episodes of slow-wave sleep (SWS; red background shading) and REM sleep (blue background shading). **b**, Spike rasters from all cells recorded simultaneously in MEC-parasubiculum during 5-s epochs of REM sleep (top) and SWS (bottom) in an example animal. Cells are sorted by mean firing rate. Population activity is dominated by theta oscillations during REM and by distinct UP and DOWN-states during SWS. Theta-rhythmic activity peaks can be detected in summed population activity (shown above rasters) during REM, while activity peaks occur at irregular intervals during SWS. **c**. Temporal autocorrelation of activity peaks across brain states. Activity peaks were identified from the summed activity of all direction-tuned cells. Panels show autocorrelation histograms of detected activity peaks during wake, REM and SWS (top to bottom), with counts of activity peaks (y-axis) at different time lags (x-axis). Note that peaks of activity occur at theta-rhythmic intervals during wake and REM, but not during SWS. **d**, Internal direction-aligned, non-rhythmic trajectories during SWS. Top: Summed firing rate of internal direction cells over a -3-second SWS epoch. Middle: sorted population activity of internal direction cells (black ticks) and tracked head direction (solid blue line) during the same period. Note sharp transitions between up and down states (with and without spiking activity, respectively). Bottom: decoded position (from three simultaneously recorded grid modules; colour-coded by time) centred around each of the highlighted peaks of direction-correlated activity (red filled circles in the top panel). Red dots show the decoded position at the time of the activity peaks. Green arrows indicate decoded direction at the time of the activity peaks. Note that the activity burst is accompanied by aligned sweeps in all three modules. **e**, Same as **d** but showing decoded position from a single grid module during three separate peaks of population activity (red filled circles in the top panel). Note that the population activity of direction-tuned cells is discretized in brief bursts and that the internal direction signal often resets between bursts of activity. **f**, More examples of decoded internal direction and sweeps during

SWS. Left: each panel shows spike rasters from internal direction cells (sorted by preferred direction) during a 5 s epoch of SWS activity from three different animals (first examples from same session as in Fig. 5a, b). Right: each panel shows a decoded sweep and internal direction (same as in **d,e**) during the period highlighted in the corresponding left panel. **g**, Decoded direction and sweep trajectories during a REM sleep episode of 2 min and 19 s. Top: raster plot showing spike times of internal direction cells during the REM episode, sorted by preferred firing direction during a separate session of open field foraging. Bottom: decoded position from grid cells during the same REM period based on fitted LMT tuning curves during wake. Position was decoded separately from grid cells belonging to each of three simultaneously recorded grid modules based on tuning curves from the open field session. Trajectories are smoothed in time with a wide gaussian kernel ($\sigma$ = 100 ms). Each panel shows the decoded trajectory for each of the three grid modules (left to right), colour-coded by time. Note that all modules play out similar trajectories (of several meters' length) over the course of the REM episode (minutes). **h**, Sweeps are aligned with direction signals in all brain states. Panels show distribution of angles between decoded direction and position (sweep) signals during wake, REM sleep and SWS (left to right). Note alignment in all states. Individual modules are plotted as separate lines. **i**, Direction-aligned sweeps are rhythmic during wake and REM, but not during SWS. Top: we computed the dot product between two vectors (arrows): internal direction (green) and sweep displacement vectors (black) at a range of time lags relative to directional activity peaks. The dot product is the length of the projection of one vector onto the other (red line). Middle: dot product across brain states for an example grid module. The dot product is positive at 0-lag during all states, indicating that sweeps and internal direction move synchronously in alignment. Note that the dot product oscillates at theta frequency during wake and REM (arrows), meaning that the grid module expresses rhythmic, direction-aligned trajectories that reset on every theta cycle. Direction-aligned trajectories are also present during SWS (positive dot product at zero lag), but they are not rhythmic. Bottom: dot product (colour-coded) as a function of time lag for all grid modules across animals. Each row shows one grid module (21 grid modules from 9 animals).

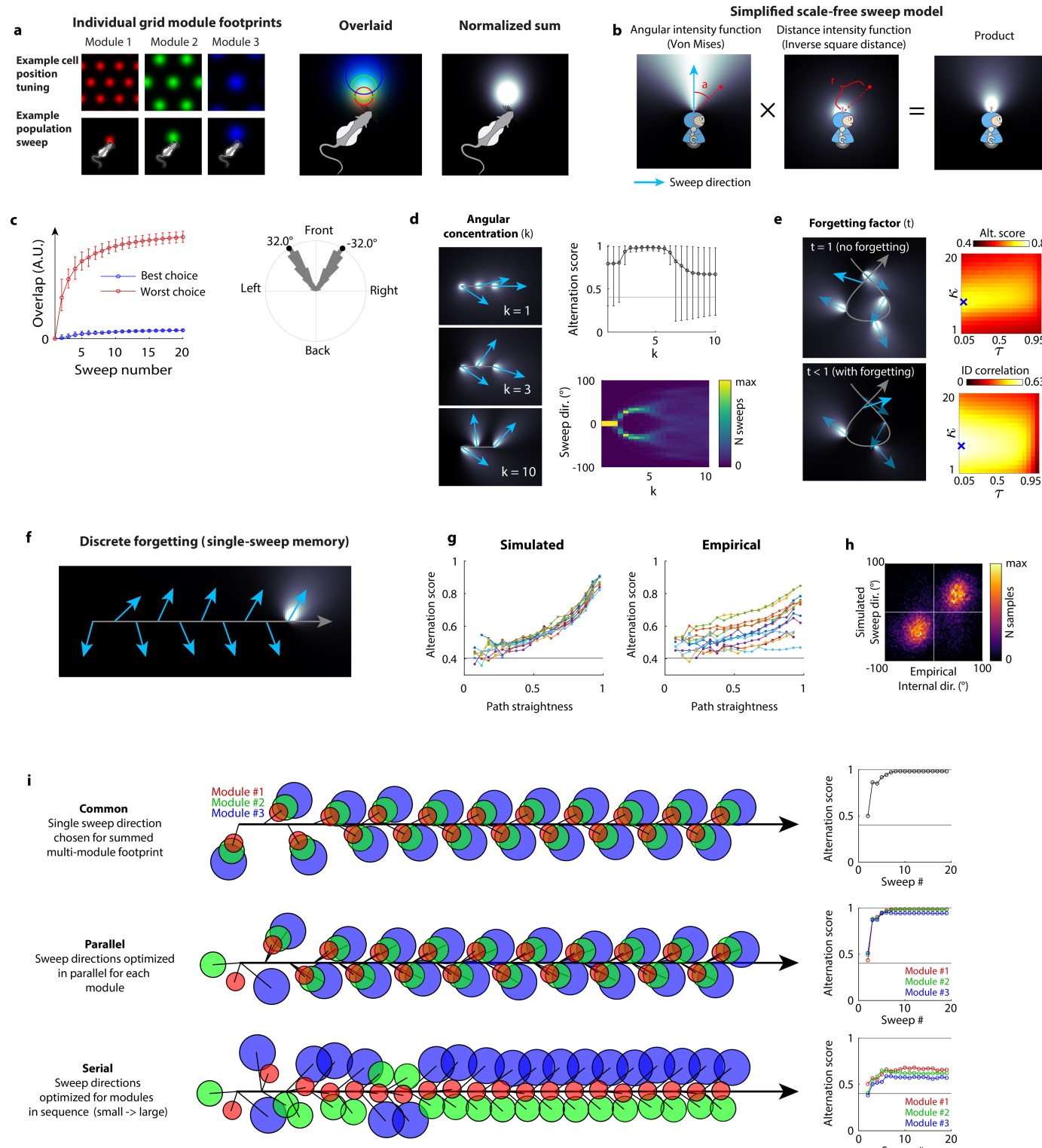

**Extended Data Fig. 12 | Left-right alternation is a stable regime for efficient spatial coverage. a**, The sum of sweeps across grid modules has a characteristic beam-like spatial footprint which expands and decays with distance. Left: simulated spatial firing patterns (top row) and footprints of sweep terminal positions (bottom row) for three simulated grid-cell modules with Gaussian fields. Note the proportionality between the grid field size, grid inter-field spacing, sweep footprint size and sweep distance. Centre: when the sweeps of the three modules are overlaid, the beam-like shape of the sweep becomes evident. Right: sum of sweep footprints across the three modules, weighted such that each module makes the same total contribution. **b**, Creating a simplified, scale-free model of the multi-module sweep footprint. For a given point in the agent's environment (red dot), the footprint intensity is calculated by multiplying two spatial functions. First, a beam-shaped angular intensity profile (left) is created by calculating the Von-Mises probability density function of the point's angular offset relative to the sweep direction (blue arrow). Next, a decaying radial profile (center) is created by calculating the inverse squared distance from the agent. Multiplying these two intensity profiles yields a decaying beam shape, similar to the multi-module sum shown in **a. c**, Optimality and bimodality of the agent's choices in 1,000 runs of the simulation in Fig. 6b. Left: average value of the largest ('worst') and smallest ('best') sweep overlap values at each time step, considering all possible sweep directions. The overlap value for a sweep at a given position and direction is defined as the product of the preexisting coverage trace with the footprint of the current sweep. Error bars indicate the $5^{th}$ and $95^{th}$ percentiles. Right: circular histogram of optimal sweep directions (smallest possible overlap with previous sweeps), relative to the direction of movement ('front'). **d**, Alternation is stable within a range of sweep widths. Left: the angular concentration of the sweep is set by the Von Mises distribution parameter κ. Top right: alternation score (mean and $5^{th}$–$95^{th}$ percentiles) for 1,000 repeated simulations across a range of sweep widths. Bottom right: colour-coded histograms (columns) of optimal sweep directions for different sweep widths. Note the bimodality of sweep directions for the intermediate range of sweep widths that also yielded high alternation scores. **e**, Rapid forgetting yields consistent alternation in the open field. Left: temporal decay of the cumulative coverage trace is achieved by introducing an exponential forgetting factor τ. When τ < 1 (bottom) the penalty trace fades over time, meaning that the agent is less influenced by earlier sweeps (transparent blue arrows). The direction of the final sweep (opaque blue arrow) changes depending on how much the penalty from the first sweep has decayed. Right panels: exploration of κ and τ parameter space in simulations where the agent was tasked with choosing sweep directions based on the recorded behavioral trajectories of each rat. Top right: average alternation score of the agent's chosen sweep directions, for each combination of κ and τ, based on the running trajectory of an animal during a recording in the open field (same session as Fig. 6e,f). Bottom right: circular correlation coefficient ('ID correlation') between decoded internal direction (ID) and the simulated sweep directions when the agent was run in 'empirically driven' mode (as illustrated in Fig. 6d). Blue crosses indicate the position of the maximum. Note that rapid forgetting and intermediate sweep widths results in robust alternation (τ: 0.01 ± 0.00, κ: 2.5 ± 0.67, mean ± s.e.m. across 13 animals) and high correlation with empirical sweep directions (τ: 0.04 ± 0.01, κ: 2.88 ± 0.11, mean ± s.e.m. across 13 animals). **f**, A single-sweep memory trace is sufficient to produce alternating sweeps. As in Fig. 6b, the agent moves along a linear path, generating a sweep at each time step. However, in this case the sweep memory trace retains only the most recent sweep. As in Fig. 6b, sweep directions (blue arrows) reliably alternate aside the direction of movement. **g**, Alternation of sweep directions increases with path straightness. Path straightness was computed for each time step of the recording by dividing net travel distance by cumulative travelled distance within a 2 s moving window (high values correspond to straight paths). Left: average alternation score of the directions chosen by the agent during an open field session, binned at different levels of path straightness. Each set of connected dots (with same color) shows data from a different animal (n = 13). Right: same, but for empirical decoded directions from the same animals. **h**, The agent accurately predicts empirical sweep directions. Heat map of decoded direction and predicted sweep directions (both in egocentric coordinates, relative to head axis) showing a high degree of correspondence between empirical and predicted sweep directions. **i**, Simulations based on multi-module sweep footprints. Sweeps were modelled as three Gaussian functions (corresponding to three grid modules, as illustrated in **a**) whose width and sweep length increased with a geometric ratio of 1.5. Each row shows a simulation in which the agent moves along a linear path, with sweeps (colored circles) generated by each module at every time step. In each simulation, the selection of sweep directions is governed by a different rule. Top ('common'): all modules are governed by a single common sweep direction, which minimizes the overlap of the summed multi-module footprint with the summed multi-module trace. Middle ('parallel'): same as top row, except that sweep directions are chosen individually for each module. On a given time step, the modules' respective sweep directions are chosen in parallel (i.e. modules are agnostic of each other's sweep directions within same time step). Note the preservation of coordinated alternation across modules, like in empirical data. Bottom ('serial'): same as middle row, except that module sweep directions are now chosen sequentially within each time step (from the smallest module to the largest module). In this serial version, note (1) the absence of strong alternation in any module, and (2) the closer packing of each module's footprints. These differences collectively suggest that alternating sweep directions in vivo are optimal for parallel processing in grid modules, rather than serial processing. Right of each row: mean alternation score for each sweep in the simulation, calculated over 1000 repetitions of the simulation with random initial conditions. Credit: rat, scidraw.io/Gil Costa; robot, openclipart.org/annares.

Edvard I. Moser

# Reporting Summary

## Statistics

For all statistical analyses, confirm that the following items are present in the figure legend, table legend, main text, or Methods section.

| n/a | Confirmed | |
|---|---|---|
| ☐ | ☒ | The exact sample size (*n*) for each experimental group/condition, given as a discrete number and unit of measurement |
| ☐ | ☒ | A statement on whether measurements were taken from distinct samples or whether the same sample was measured repeatedly |
| ☐ | ☒ | The statistical test(s) used AND whether they are one- or two-sided<br>*Only common tests should be described solely by name; describe more complex techniques in the Methods section.* |
| ☐ | ☒ | A description of all covariates tested |
| ☐ | ☒ | A description of any assumptions or corrections, such as tests of normality and adjustment for multiple comparisons |
| ☐ | ☒ | A full description of the statistical parameters including central tendency (e.g. means) or other basic estimates (e.g. regression coefficient) AND variation (e.g. standard deviation) or associated estimates of uncertainty (e.g. confidence intervals) |
| ☐ | ☒ | For null hypothesis testing, the test statistic (e.g. *F*, *t*, *r*) with confidence intervals, effect sizes, degrees of freedom and *P* value noted<br>*Give P values as exact values whenever suitable.* |
| ☒ | ☐ | For Bayesian analysis, information on the choice of priors and Markov chain Monte Carlo settings |
| ☒ | ☐ | For hierarchical and complex designs, identification of the appropriate level for tests and full reporting of outcomes |
| ☐ | ☒ | Estimates of effect sizes (e.g. Cohen's *d*, Pearson's *r*), indicating how they were calculated |

*Our web collection on statistics for biologists contains articles on many of the points above.*

## Software and code

Policy information about availability of computer code

| Data collection | Commercial software: Motive (OptiTrack) version 2.2.0; MATLAB (MathWorks) version r2019b<br>Open-source software: SpikeGLX (https://billkarsh.github.io/SpikeGLX) versions 20190724 and 20190919 |
|---|---|

| Data analysis | Commercial software: MATLAB (MathWorks) version r2020b, Python version 3.7 |
|---|---|

Open-source code (for MATLAB): Kilosort version 2.5 (https://github.com/MouseLand/Kilosort); UMAP version 1.4.1 (https://www.mathworks.com/matlabcentral/fileexchange/71902-uniform-manifold-approximation-and-projection-umap); CircStat toolbox (https://github.com/circstat/circstat-matlab); Chronux toolbox (http://chronux.org/); Latent manifold tuning model (https://github.com/waq1129/LMT); CCH deconvolution (https://github.com/EranStarkLab/CCH-deconvolution); Code for reproducing the analyses in this article are available at Zenodo, DOI: placeholder (will-be-provided-at-proofs-stage).

Open-source Python packages:
numpy          1.18.1
scanpy         1.7.2
pandas         1.1.5
anndata        0.7.8
scipy          1.4.1
scikit-learn   0.22.2
matplotlib     3.3.4
DeepLabCut     2.3.10

For manuscripts utilizing custom algorithms or software that are central to the research but not yet described in published literature, software must be made available to editors and reviewers. We strongly encourage code deposition in a community repository (e.g. GitHub). See the Nature Portfolio guidelines for submitting code & software for further information.

## Data

Policy information about availability of data

All manuscripts must include a data availability statement. This statement should provide the following information, where applicable:
- Accession codes, unique identifiers, or web links for publicly available datasets
- A description of any restrictions on data availability
- For clinical datasets or third party data, please ensure that the statement adheres to our policy

The datasets generated during the current study are available at EBRAINS, DOI: https://doi.org/10.25493/R5FR-EDG.

## Research involving human participants, their data, or biological material

Policy information about studies with human participants or human data. See also policy information about sex, gender (identity/presentation), and sexual orientation and race, ethnicity and racism.

| Reporting on sex and gender | N/A |
|---|---|
| Reporting on race, ethnicity, or other socially relevant groupings | N/A |
| Population characteristics | N/A |
| Recruitment | N/A |
| Ethics oversight | N/A |

Note that full information on the approval of the study protocol must also be provided in the manuscript.

## Field-specific reporting

Please select the one below that is the best fit for your research. If you are not sure, read the appropriate sections before making your selection.

☒ Life sciences  ☐ Behavioural & social sciences  ☐ Ecological, evolutionary & environmental sciences

For a reference copy of the document with all sections, see nature.com/documents/nr-reporting-summary-flat.pdf

## Life sciences study design

All studies must disclose on these points even when the disclosure is negative.

| Sample size | Samples included all available cells that matched the classification criteria for the relevant cell type. |
|---|---|
| Data exclusions | Cells with low firing-rates (<0.1Hz or 0.025Hz) were excluded from analyses because of their unsuitability for spike-train analysis. |
| Replication | In the results text we indicate for each result the number of animals or grid modules in which the effect was found. Multiple recordings were done in each animal and the recording session with the best unit yield and behavioral performance was included in the study. |

| Randomization | The study did not involve any experimental subject groups; therefore, random allocation did not apply and was not performed. |
| Blinding | The study did not involve any experimental subject groups; therefore, experimenter blinding did not apply and was not performed. |

# Reporting for specific materials, systems and methods

We require information from authors about some types of materials, experimental systems and methods used in many studies. Here, indicate whether each material, system or method listed is relevant to your study. If you are not sure if a list item applies to your research, read the appropriate section before selecting a response.

## Materials & experimental systems

| n/a | Involved in the study |
|---|---|
| ☒ ☐ | Antibodies |
| ☒ ☐ | Eukaryotic cell lines |
| ☒ ☐ | Palaeontology and archaeology |
| ☐ ☒ | Animals and other organisms |
| ☒ ☐ | Clinical data |
| ☒ ☐ | Dual use research of concern |
| ☒ ☐ | Plants |

## Methods

| n/a | Involved in the study |
|---|---|
| ☒ ☐ | ChIP-seq |
| ☒ ☐ | Flow cytometry |
| ☒ ☐ | MRI-based neuroimaging |

## Animals and other research organisms

Policy information about studies involving animals; ARRIVE guidelines recommended for reporting animal research, and Sex and Gender in Research

| Laboratory animals | Long Evans rats, male and female, age 3-4 months (300-500 g) |
| Wild animals | None |
| Reporting on sex | Findings apply to both sexes (17 males, 1 female rat). Sex-based analyses were not performed. |
| Field-collected samples | None |
| Ethics oversight | Protocols approved by the Norwegian Food Safety Authority (FOTS ID 18011) |

Note that full information on the approval of the study protocol must also be provided in the manuscript.

## Plants

| Seed stocks | N/A |
| Novel plant genotypes | N/A |
| Authentication | N/A |

