## [Peer Review File · Nature]

Left-right-alternating theta sweeps in entorhinal-hippocampal maps of space

Corresponding Author: Dr Edvard Moser

Version 0:

Reviewer comments:

Referee #1

(Remarks to the Author)

Here, Vollan, Gardner et al. record large populations of simultaneously recorded neurons in rat MEC, parasubiculum, and HPC to demonstrate theta sequences that sweep outwards at 30° CW and CCW angles relative to the animals current directional heading in an alternating fashion during free exploration. They show that the sweeps align with an internal direction signal, scale in length with grid modules, are putatively driven by internal direction sensitive cells and conjunctive grid cells, extend into unsampled locations, and persist during sleep. They show that sweeps in MEC temporally lead those in HPC providing a hypothetical circuit for the alternating spatial sweeps previously reported in HPC by the Frank lab (Kay et al., 2020). Finally, they employ a computational model to demonstrate that the alternating theta sequences sample proximal space optimally. As is common for the Moser lab, this work is impressive, well presented, with experiments and corresponding analyses that are appropriate for the experimental questions. That said, I have a few comments regarding interpretation and analyses that should be addressed in revision.

I am having difficulty with the spatial sampling hypothesis related to spatial sweeps. Namely, it is unclear if the manuscript in current form provides evidence that the theta sequences observed here are in actuality sampling anything. A task-elicited bias in sweeping would be useful in this regard. It is hard for me to see how it is optimal to sample spaces outside of the linear track that the animal has no ability to see (e.g. beyond opaque walls) or navigate (e.g. drop offs on a maze)? Why wouldn't the sweep adjust to sample spaces in front or behind the animal that are navigationally afforded in these scenarios (as in work from the Redish/Frank labs)?

The points above make me wonder how much the alternating sweeps and internal direction are related to movement. Recent work has demonstrated that the theta oscillation and hippocampal representations therein can be coupled to motor actions like stepping (Joshi et al., 2023). Is it possible that the alternating sweeps are related to something like this (perhaps even micro head movements), rather than spatial sampling? Obviously the observation of spatial sweeps during REM sleep is good evidence against this explanation but it is possible alternating sweeps are 'wired up' during behavior and then persist during sleep. I think some effort should be made to exclude fine grain behavior as a potential source of the lateral spatial sweeping. This should be doable given the animal tracking system the authors have in place.

This sweep delay between MEC and HPC is an interesting component of the manuscript and adds new insight into previous reports related to HPC spatial sweeping. From reading the methods it appears that theta phase is estimated using multiunit activity in MEC/PS rather than HPC. Does the delay change at all if using multiunit estimates from HPC instead and/or just local field potentials from each area independently?

I'm also curious if HPC sweeps are delayed relative to MEC on the linear tracks. I'm wondering if the delay arises partly from distinct firing dynamics in 2D spaces that would be potentially more similar between the two structures on the track. Is it possible these sweep lags would go away if spike counts were normalized between the two structures?

Minor.

I really like the internal direction element of this work - the correspondence with spatial sweeps is quite nice. Very cool analysis and really interesting to think about. I did wonder about the percentage of direction coding neurons that were non-cycle-skipping (I believe the text says 'most') and whether they dominate the population code when the internal direction and observed direction were aligned?

L104-107 - It is interesting to think about why grid modules would be aligned with respect to sweep direction. Wouldn't it be more optimal to have them desynchronized to effectively sample space on both sides simultaneously? Did the authors consider using a modified version of the simulation to explore this?

Were sweeps observed in multiple hippocampal sub-regions? Did they have similar properties?

L181-193 + Figure 3b - These percentages are very small, can the authors report total numbers? Are these percentages of all cell pairs or cell pairs that were putatively monosynaptic?

L200 - 'ciruclar' typo

L564-565 - repetition of $(0.01 \text{ rad} < \sigma < 1 \text{ rad})$

L724 - I'm confused by the weighting by the correlation value for internal direction estimation. Do you mean that estimates within the theta cycle that have lower correlations to the reference vector are down weighted? Does this apply to negative correlation values and what happens to the estimate if the downweighting isn't applied?

L728 - Is this intended to be $2d$?

(Remarks on code availability)

Referee #2

(Remarks to the Author)

The authors describe a new phenomenon of theta-cycle alternating forward-right, forward-left sweeps of the MEC/parasubicular/hippocampal position and direction representation during rodent locomotion. They perform a comprehensive set of analyses to quantify and further explore this phenomenon. For instance they found that grid cell sweeps are aligned across different grid modules, the sweeps look forward within a distance of half of the grid spacing in each module, hippocampal place cell sweeps follow simultaneously measured grid cell sweeps by ~20 millisecond which suggests a grid cell origin for sweeps, and the MEC/parasubicular direction signal is also aligned in direction with the grid sweeps. In addition, cross-correlation analysis suggests a multisynaptic circuit from direction cells to grid cells underlying the shift in internal position representation of grid cells during sweeps, which also constitutes exciting evidence of a mechanism for moving on the continuous grid cell attractor. They also show that sweeps extend to unexplored space such as on the sides of linear tracks, that alternating sweeps occur during REM sleep, and that in SWS trajectories are aligned with the direction signal.

Overall this is an excellent study that describes a fundamental finding of the rodent spatial system and explores this finding in great detail.

I have one major area of comment and it revolves around understanding the analysis on sweeps extending to never visited locations. First, could this analysis be extended to see if the alternating sweeps are present during the first traversal when an animal is running in a novel space? Second, could the authors explain and clarify their LMT (latent manifold tuning) analysis approach more? I can imagine how grid cell firing patterns could be extended in space and be used to infer positions outside of explored regions. However, while I might have missed it, I didn't see any periodic constraint on individual neuron tuning curves, just gaussian and smoothness constraints. The hippocampal place fields that are inferred in Fig4i look like they are extensions of fields that start inside the walls and thus I can understand how these tuning curves are inferred. But some inferred grid tuning curves in Fig4g look like some of their fields might be completely outside the explored regions of the environments. In such a case I would assume periodicity would provide the constraint to help infer those fields. On the other hand is it the case that all inferred grid fields already have some part of the field rising at the edge of the explored environment which would not then require a periodic constraint, and does this relate to their finding that grid cell sweeps essentially always extend less than half of the grid spacing? If so, it would help if the authors explicitly state that there is always some part of the inferred field inside the regions the animal visited, and that the LMT model builds the new fields that are centered outside of the visited regions from those small "foothills" of fields. Also if there is no periodic-like boundary condition in the LMT analysis, the authors should emphasize this point in the main text or methods since I was assuming it was there and other readers might do the same.

A second comment relates to the sleep trajectory analysis. Can the sleep trajectories be analyzed with the population vector approach (versus LMT approach) as used for the initial sweep analysis since all reactivated locations have been visited before, and what would the results look like?

(Remarks on code availability)

Referee #3

(Remarks to the Author)

The manuscript's major finding about the left-right alternation of population vector activity across theta cycles in the

entorhinal and related areas is indeed intriguing. Forward sweeps in theta cycle are known. The novel observation that forward-going population vector activity alternates across theta cycles offers new insights into the dynamics of neural representation during theta oscillations. However, there are several critical points that need to be addressed to strengthen the conclusions drawn from these findings.

Single-Cell Analysis of Alternating Theta Sweeps:

Current work: The population vector analysis, while comprehensive, does not specify which individual cells contribute to the observed alternation effect. This leaves room for potential amplification of random effects by population vectors. Suggestion: The authors could conduct an analysis at the level of single cells. By identifying whether individual cells exhibit systematic phase-precessed firing patterns that align with the observed population-level alternation, they can provide stronger evidence for the phenomenon. Techniques like spike-phase coupling analysis or single-cell autocorrelation within theta cycles could be insightful.

Relation Between Theta Skipping and Theta Sweeps:

Current Explanation: The authors suggest that theta skipping might be related to the observed alternating theta sweeps, but this connection seems speculative and unclear.

Clarification Needed: The authors could provide details of how theta cycle skipping could lead to systematic sweeping. Specifically, they should investigate if cells that participate in theta skipping also contribute consistently to one side of the alternation. Evidence from spike-timing and phase-precession analysis could support this hypothesis.

Mechanism of Theta Sweep Phenomenon: Current Mechanism (Figure 3):

The proposed monosynaptic connections between subgroups of neurons as a mechanism for theta sweeps raises concerns due to the expected synaptic delays.

Mechanism Feasibility: The authors need to provide evidence or simulations demonstrating that presynaptic spikes can reliably induce postsynaptic spikes within the tight time frame of 2 milliseconds, considering the axonal, dendritic delays, and membrane time constants.

Allocentric vs. Head-Centered Coordinate System:

Current Observation: The medial entorhinal cortex is known for its allocentric spatial mapping, yet the observed sweeps are head-centered.

Reconciliation of Observations: The authors could discuss potential mechanisms by which these head-centered sweeps integrate into the broader allocentric map of space. One possibility is that head-centered sweeps could contribute to updating the allocentric map based on directional movement or heading. Including a theoretical or computational framework could help reconcile these observations.

Robustness of Theta Sweep Analysis:

Current Analysis: The significance of sweeps is assessed based on triplets of theta cycles.

Suggestion for Robustness Testing: To demonstrate that the alternation is a robust and continuous process, the authors could extend their analysis to include both smaller (2 cycles) and larger numbers of cycles. This could help confirm that the observed alternation is not only for the chosen cycle window and is indeed a persistent feature of the neural activity.

Conclusion: The findings reported in the manuscript offer a fascinating glimpse into the dynamics of neural activity during theta oscillations. However, addressing the points mentioned above through additional analyses and more detailed explanations will significantly strengthen the conclusions and provide a clearer understanding of the underlying mechanisms. This will help reconcile the novel observations with existing knowledge about entorhinal cortex function and theta oscillations.

(Remarks on code availability)

The data and codes will be made available prior to publication. This is oK.

Version 1:

Reviewer comments:

Referee #1

(Remarks to the Author)

The authors have sufficiently addressed my concerns. This is a very cool paper and I appreciate the hard work that went into the original manuscript and the revision. Nice job. I have only a few minor comments.

I commend the authors for their in depth analysis of the relationship between micro movements and sweeps. A few comments:

Extended Data 2 - can we see the frequency and coherence between theta and lateral head movements as in e and f above.

Extended Data 2g what is the y-axis for the lateral speed plots?

Extended Data 2h right plot - are these directions rotated 90? It looks so compared to the left plot.

I think Figure 4 for reviewers is an important feature of the data. The sweep lags and decoding offsets seem modulated by the shape of the environment/task. I would encourage the authors to include this point and figure in the manuscript.

(Remarks on code availability)

Referee #2

(Remarks to the Author)

The new data showing the alternating sweeps exist at the very beginning of a novel exposure is a valuable addition, and the authors have addressed the rest of my comments. Overall I believe this work would be a very nice addition to the literature.

(Remarks on code availability)

Referee #3

(Remarks to the Author)

The authors have responded to all of my observations in a satisfactory manner.

(Remarks on code availability)

Response to Reviewers

We thank all Reviewers for their thoughtful comments. Their comments have helped us improve the manuscript in several important ways. First, we have improved the description and implementation of analyses, intending to address all points of the Reviewers. Interpretations regarding mechanism and function should now be clearer. Second, we have added new experiments to address questions about whether sweeps and direction signals are determined by the location of salient landmarks such as rewards and obstacles, and we have used data from these new experiments as well as new analyses on the original data to determine whether the left-right alternation pattern follows rhythms in the animal's fine-grained behavior (such as lateral and vertical head movements, or forelimb stepping).

Reviewer comments are kept in *blue italic*, our responses in black standard font.

Referee #1 (Remarks to the Author)

Here, Vollan, Gardner et al. record large populations of simultaneously recorded neurons in rat MEC, parasubiculum, and HPC to demonstrate theta sequences that sweep outwards at 30° CW and CCW angles relative to the animals current directional heading in an alternating fashion during free exploration. They show that the sweeps align with an internal direction signal, scale in length with grid modules, are putatively driven by internal direction sensitive cells and conjunctive grid cells, extend into unsampled locations, and persist during sleep. They show that sweeps in MEC temporally lead those in HPC providing a hypothetical circuit for the alternating spatial sweeps previously reported in HPC by the Frank lab (Kay et al., 2020). Finally, they employ a computational model to demonstrate that the alternating theta sequences sample proximal space optimally. As is common for the Moser lab, this work is impressive, well presented, with experiments and corresponding analyses that are appropriate for the experimental questions. That said, I have a few comments regarding interpretation and analyses that should be addressed in revision.

I am having difficulty with the spatial sampling hypothesis related to spatial sweeps. Namely, it is unclear if the manuscript in current form provides evidence that the theta sequences observed here are in actuality sampling anything. A task-elicited bias in sweeping would be useful in this regard. It is hard for me to see how it is optimal to sample spaces outside of the linear track that the animal has no ability to see (e.g. beyond opaque walls) or navigate (e.g. drop offs on a maze)? Why wouldn't the sweep adjust to sample spaces in front or behind the animal that are navigationally afforded in these scenarios (as in work from the Redish/Frank labs)?

We thank the Reviewer for asking us to clarify our spatial sampling hypothesis. We recognize that we did not explain well what we meant by spatial sampling. To address the Reviewer's concern, we have clarified the terminology as well as our interpretation of function in the Discussion section. In addition, although the Reviewer does not directly ask for it, we have included new data from a task that further probes the question about biases towards salient regions of the environment.

First, regarding terminology and our intended meaning of “spatial sampling” and the optimality of such sampling, we originally used “sampling” in a statistical sense, meaning the process of choosing a subset of elements from the full set. Each sweep covered a small region of the full (internally represented) 2D space, and hence choosing a series of sweeps could be viewed as drawing a statistical sample of locations from the full space. We did not intend to imply anything about the behavioral relevance or sensory features of the “sampled” swept-to locations. On a comparable note, we used the term “efficient” to describe a set of sweeps that covers a large portion of the surrounding 2D space in few sweeps (which, intuitively, is achieved by placing sweeps in non-overlapping locations), and by “optimal” we meant “optimally efficient” – the most efficient possible combination of sweeps. Importantly, since “sampling” relates to the toroidal manifold rather than physical space, our definitions of “efficient” and “optimal” did not strictly imply anything about navigational utility in the physical environment. With this intended meaning of the terms, we found that the data were well explained by an efficient spatial sampling interpretation, referring to sampling of the manifold space, consistent with the observation of sweeps through inaccessible locations. To make this point clearer in the revised manuscript, we have now changed the wording from “optimal/efficient sampling” to “optimal/efficient coverage” throughout the manuscript. In addition we have pointed out directly, in the second paragraph of the Discussion, and in the Abstract, that coverage (or sampling) refers to the manifold space, and not external physical or sensory space. We introduce explicitly the possibility that the reference for sweeps and direction signals is location on the toroidal manifold rather than the animal’s physical environment, suggesting that the mechanisms are primarily internal and not caused by the location of particular landmarks or bottom-up sensory input. We hope this clarification addresses the Reviewer’s queries.

Because the question about what is sampled is a fundamental one, we have added new supplementary data relevant to the Reviewer’s question. The Reviewer is right that the original manuscript leaves somewhat open the question of whether sweeps are biased towards (or away from) navigationally relevant locations such as goals or obstacles. In the original manuscript, we had data from a linear track and a wagon-wheel track, in which the goals had distinct locations (ends of the linear track; centre of cross-linking arms of the wagon-wheel track). Rigid left-right alternation persisted in these tasks, suggesting – together with wall-crossing sweeps in the open field – that sweeps (and directional signals) ignored salient locations in the environment. In the revised manuscript, we have corroborated these findings by adding data from a third goal navigation task, an m-shaped extended T-maze, in which navigational decisions at the junction have consequences for whether the animal finds reward, unlike in the linear track and the wagon-wheel track, where the animal finds reward on each section, independently of its navigational decisions at the junctions (the m task is similar to the task used in studies of place cells from the Frank lab (Frank, Brown, and Wilson 2000; Kay et al. 2020). Due to this task difference, demands on planning and memory retrieval may be stronger in the m-maze. In the revised manuscript we add data from one new animal, in which 1,055 MEC-parasubiculum cells and 409 hippocampal cells were recorded in one session on the m-maze. The data from this animal suggests that the left-right alternation pattern persists across the entire environment, also

ahead of the decision point. Internal direction continued to reliably alternate between head-centered angles ~ 30 degrees to either side in all parts of the maze (regardless of the upcoming choice and direction to the goal), resembling the stereotypical pattern observed in the open field in the original data. This observation reinforces the original claim that the alternation pattern is hardwired in the circuit and expressed independently of the location of upcoming goals or obstacles. The maintained left-right alternation in the m-maze is shown in Extended Data Fig. 8 of the revised manuscript (and referred in the “Sweeps extend to never-visited locations” section of the main text).

[REDACTED]

REDACTED

[REDACTED]

[REDACTED]

[REDACTED]

[REDACTED]

[REDACTED]

[REDACTED]

The points above make me wonder how much the alternating sweeps and internal direction are related to movement. Recent work has demonstrated that the theta oscillation and hippocampal representations therein can be coupled to motor actions like stepping (Joshi et al., 2023). Is it possible that the alternating sweeps are related to something like this (perhaps even micro head movements), rather than spatial sampling? Obviously the observation of spatial sweeps during REM sleep is good evidence against this explanation but it is possible alternating sweeps are 'wired up' during behavior and then persist during sleep. I think some effort should be made to exclude fine grain behavior as a potential source of the lateral spatial sweeping. This should be doable given the animal tracking system the authors have in place.

We thank the Reviewer for bringing up this question. Indeed, several previous studies have identified relationships between the hippocampal theta rhythm and fast rhythmicity in the animal's behaviors (Grion et al. 2016; Kleinfeld, Deschênes, and Ulanovsky 2016), including locomotion-related vertical head oscillations (Ledberg and Robbe 2011) and footsteps (Joshi et al. 2023). Since locomotion and theta rhythms often have similar frequencies and because left-right-alternating sweeps are most strongly expressed during running, we agree with the Reviewer that a more thorough analysis of their potential coupling is warranted. In response to the Reviewer's request, we have added Extended Data Fig. 2 to the revised manuscript, in which we show – partly with further analysis of old data and partly by adding new data – that oscillatory head and body movements during locomotion are not reliably coupled to theta oscillations or left-right-alternation in internal direction signals. The new data and analyses show that lateral head or body movements are neither sufficient nor necessary for left-right-alternation of sweeps and internal direction signals.

We addressed the Reviewer's questions by quantifying the rhythmicity of body movements during locomotion (and in the absence of locomotion). Analyses were performed on existing data from the linear track as well as new data where footsteps were videorecorded from below while the animal was running on a transparent floor. Head, foot and body positions were tracked using marker-based 3D motion capture (OptiTrack) and marker-free pose estimation (DeepLabCut) from behavioral videos. In addition to the previously reported ~6 Hz vertical head oscillations during linear track running (Ledberg and Robbe 2011), we observed that most animals displayed ~3 Hz lateral head swings (i.e. the head moves back and forth in a U-shaped pattern in the frontal plane) during locomotion. In one animal with reflective markers on the back (in addition to head-mounted markers), we found that the lateral head swings were coupled to left-right-alternating body movements. In another animal – the animal that was added specifically for the revision – we tracked limb movements during open field foraging by capturing video from below, through a transparent floor (Extended Data Fig. 2h). In the new experiment we found that forelimb movements were synchronized with lateral head movements (Extended Data Fig. 2c). Taken together, these recordings suggest that left-right-alternating movements of the head, body and limbs are all part of the same locomotor rhythm. This does not necessarily mean that they are synchronized with the left-right alternation of direction signals and sweeps, however. Although the oscillation frequency was quite similar during fast running, we found no

significant coherence between theta oscillations and vertical or lateral head oscillations (Extended Data Fig. 2d-g). Furthermore, we did not find any consistent phase locking of left-right-alternating direction signals to the left-right-alternating gait cycle (Extended Data Fig. 2h), mirroring observations in (Joshi et al. 2023), where left/right forelimb plants did not reliably coincide with left/right sweeps in their hippocampal recordings. Lastly, we show that theta – and left-right-alternating sweeps and direction signals – both preceded and succeeded running bouts with head oscillations (Extended Data Fig. 2d,g). We thus conclude that left-right-alternating sweeps cannot be explained by locomotor-related head and body movements.

In some animals (e.g. Extended Data Fig. 2i), left-right-alternation of the internal direction signal appeared to be weakly modulated by lateral head movements. This modulation may arise because of phase-amplitude coupling between theta and head/body oscillations (Ledberg and Robbe 2011; Joshi et al. 2023) or because of transient phase-phase coupling between the two rhythms (Grion et al. 2016). However, our analyses show that any such impact is small and we believe detailed analysis of this topic is better suited for a follow-up study. The data included in the new Extended Data Fig. 2 show clearly that head, body and foot-step movements are neither necessary nor sufficient for the expression of left-right theta sweeps and internal direction signals.

While the new analyses are presented primarily in Extended Data Fig. 2, a summary of these findings – with reference to the new figure – has been added in the main text at the end of the second paragraph (line 53-55). We kept the summary short, given that the data showed so clearly that neural and behavioral rhythms were by and largely independent.

This sweep delay between MEC and HPC is an interesting component of the manuscript and adds new insight into previous reports related to HPC spatial sweeping. From reading the methods it appears that theta phase is estimated using multiunit activity in MEC/PS rather than HPC. Does the delay change at all if using multiunit estimates from HPC instead and/or just local field potentials from each area independently?

We thank the Reviewer for the interest in the temporal delay between entorhinal and hippocampal sweeps. We want to emphasize that this delay was observed without reference to theta oscillations in either region. To quantify the temporal delay, we first decoded position independently from HPC and MEC. Next, we computed, for each region, the offset between decoded position and the rat's position, projected onto the rat's head axis. Lastly, we computed the temporal cross-correlation between head-centered decoded position from each region and detected the peak closest to zero-lag. Thus, there is no reference to MEC or HPC theta oscillations in this calculation.

We suspect that the miscommunication was caused by our use of MEC theta troughs as a reference in the example plot in Fig. 1i (and again in Extended Data Fig. 3g). Below, as Figure 2 for Reviewers, we show the same plot using HPC theta as a reference. It can be seen that the relative timing between MEC and HPC sweeps in the two plots is unchanged, which is expected given that the choice of theta reference (MEC or HPC) should not affect the relative timing between decoded position in HPC and MEC sweeps.

Figure 2 for Reviewers: Plots show the forward progression of decoded sweeps referenced to entorhinal (left) or hippocampal (right) theta during an example open field session (same as in Fig. 1i). The offset between decoded position and lowpass-filtered decoded trajectory was projected onto the rat's head axis and averaged across all theta cycles in the session. Theta phase was extracted from multiunit spiking activity from either region. Note that the temporal delay between hippocampal sweeps and MEC sweeps is unaffected by choice of theta reference region.

To remove any doubts about how we calculated the delay between sweeps in MEC and hippocampus, we have added a paragraph in the Methods where we more specifically describe the calculation of the delay between sweeps (see *Temporal delay between hippocampal and entorhinal sweeps*). Importantly, we emphasize in this paragraph that theta phase (trough) is used in Fig. 1i and Extended Data Fig. 3g only for visualization purposes.

Finally, since theta waves travel along the longitudinal axis of both MEC and hippocampus, with phase shifts of up to 180 degrees from dorsal to ventral (Lubenov and Siapas 2009; Patel et al. 2012; Hernández-Pérez, Cooper, and Newman 2020), the reviewer may be concerned about whether recordings were performed at homologous dorsoventral levels in the two regions. We cannot entirely rule out some difference in dorsoventral locations; however, the fact that the temporal delay was present also in two animals with recordings from the very dorsal tip of hippocampus speaks against such differences as an explanation of the decoding offset (see Figure 3 for Reviewers below).

Figure 3 for Reviewers: Recording locations in hippocampus (left), recording locations in MEC-parasubiculum (middle) and temporal alignment of sweeps decoded from each region (right, plotted as in Fig. 2 for Reviewers) for two animals with recording sites in the septal pole of hippocampus (right above anterior thalamus). Note that hippocampal sweeps are delayed relative to entorhinal sweeps also in these animals.

I'm also curious if HPC sweeps are delayed relative to MEC on the linear tracks. I'm wondering if the delay arises partly from distinct firing dynamics in 2D spaces that would be potentially more similar between the two structures on the track. Is it possible these sweep lags would go away if spike counts were normalized between the two structures?

We have now repeated the analysis of HPC-MEC sweep delay with the linear track data (and also the new m-maze data). Also in this condition, we observe a consistent temporal lag between MEC and HPC sweeps (Figure 4a-b for Reviewers, below, referring to data from linear track and m-maze, respectively). We have not included these data in the manuscript, considering that the rebuttal letter will be published, but can do so if Reviewers and Editors prefer so.

Figure 4 for Reviewers: **a)** Left plot shows forward progression of entorhinal (blue) and hippocampal (green) sweeps during a linear track session from an example animal (data plotted as in Fig. 1i and Figure 1 for Reviewers). Right plot shows cross-correlations of decoded positions in MEC and hippocampus for 4 animals with dual hippocampal and entorhinal recordings that were tested on the linear track (1 session per animal).

Hippocampal sweeps are delayed by 16.2 ± 4.0 ms with respect to entorhinal sweeps. **b)** Same as above, but for m-maze experiment ($n=1$ session). Hippocampal sweeps were delayed by 7.3 ms with respect to entorhinal sweeps. **c)** Forward progression and sweep delay in the open field arena, plotted as in **a)**, when position is decoded from the same number of cells in MEC and hippocampus (downsampling cells in the region with most cells to match the number in the other region; $n=7$ open field sessions from 7 animals).

In response to the question about whether normalization of spike counts would remove the delay effect, we have downsampled units in the more active region of hippocampus or MEC-parasubiculum to balance the amount of data from each region. With sweeps in each region

decoded from the same number of units, the same delays were still observed (Figure 4c for Reviewers). A one-sentence statement about this has been added in the Methods (line 757-759).

Minor.

I really like the internal direction element of this work - the correspondence with spatial sweeps is quite nice. Very cool analysis and really interesting to think about. I did wonder about the percentage of direction coding neurons that were non-cycle-skipping (I believe the text says 'most') and whether they dominate the population code when the internal direction and observed direction were aligned?

We thank the Reviewer for these important questions. The large majority of the direction coding neurons in our cell sample (mostly from superficial MEC and parasubiculum) were both theta-modulated and cycle-skipping (2,253/3,632 or 62.0% of all direction-tuned cells). The remaining 1,379 non-skipping cells (38.0% of total) were either theta-modulated (796 cells or 21.9% of total) or not (583 cells or 16.0% of total). For most analyses, population decoding was performed using all cells (including non-directional cells) but with samples dominated by the large fraction of cycle-skipping internal direction cells. When analyses were restricted to non-rhythmic, non-skipping head direction cells (often recorded outside parasubiculum), we saw a non-alternating direction signal that faithfully tracked the animal's head direction relative to the environment (Figure 5 for Reviewers below).

Figure 5 for Reviewers: head-centered distribution of decoded direction from simultaneously recorded directional cells in MEC-parasubiculum, anterodorsal thalamus and presubiculum. Thalamic and presubicular recordings are not included in the current dataset. Note that head direction cells in anterodorsal thalamus and presubiculum precisely track the animal's head direction (defined relative to the external environment).

However, since the yield of non-skipping head direction cells was quite low in the current dataset and the focus of the paper is on the internal direction signal, we have chosen not to put the above figure in the manuscript. If Reviewers or Editors would like us to do so, we can do it, with the addition of a cautionary statement about sample size (it would also require the addition of methods for recording in presubiculum and anterior thalamus where such cells were abundant; cells from these regions are not part of the current dataset).

The current references to non-rhythmic directional cells can be found in two places in the revised manuscript:

L179-183: "... the smaller sample of non-rhythmic directional cells faithfully followed tracked head direction ('head direction cells'; Extended Data Fig. 6c). The latter cells were generally located outside the area of internal direction cells: in deep layers of MEC, presubiculum or postrhinal cortex (Extended Data Fig. 6d-e)."

L162-165: "The rigid left-right alternation of the population direction signal provides an explanation for the phenomenon of theta cycle skipping in individual cells, which was present in most directional cells and grid cells (Fig. 2a; Extended Data Figs. 3b and 6b)."

In addition, to leave no doubt that non-skipping cells are the minority, we have now included the number of non-skipping and non-rhythmic cells in the legend of Extended Data Fig. 6b: "Out of all 3,632 direction-tuned cells, 2,253 or 62.0% were both theta-modulated and cycle-skipping, 796 cells or 21.9% were non-skipping and theta-modulated, and 583 cells or 16.0% were non-skipping and non-rhythmic".

L104-107 - It is interesting to think about why grid modules would be aligned with respect to sweep direction. Wouldn't it be more optimal to have them desynchronized to effectively sample space on both sides simultaneously? Did the authors consider using a modified version of the simulation to explore this

We agree this is an interesting question – indeed, it might seem more efficient if modules were to sweep in different directions, thus spanning a larger total area with every sweep. To approach this question, we have added new simulation data (Extended data fig. 10i) based on a modified version of the artificial agent that explicitly incorporates multi-module sweeps.

We found that even if sweep directions are chosen individually and in parallel for each module, the directions converge on a coordinated alternating pattern just like in the original model (and the empirical data). We identified one condition in the simulations in which modules chose different directions: If sweeps were performed in series (from the smallest to the largest module), rather than in parallel, the simulation often converged on solutions without regular left-right-alternation in each module. Our data suggest that this latter condition is not implemented in the entorhinal-hippocampal network, however.

In summary, the optimal strategy depends on the objective: minimizing overlap *within* scale levels (modules) favors coordinated alternation, whereas minimizing overlap *between* scale levels favors modules sweeping in different directions. Our empirical results, which show reliable coordination of sweep directions across modules, suggest that the underlying objective of grid-cell sweeps is for each grid module to avoid overlapping with its own former sweeps – in other words, favoring parallel processing in modules rather than serial processing. This is now explained in Extended Data Fig. 12i, and in the final sentence of the model section in the Results (line 348-349).

Were sweeps observed in multiple hippocampal sub-regions? Did they have similar properties?

Yes, sweeps were observed in CA3 as well as CA1. We have added examples from each region in Extended Data Fig. 3h. We have not compared CA3 and CA1 sweeps quantitatively as we see this as beyond the scope of the study.

L181-193 + Figure 3b - These percentages are very small, can the authors report total numbers? Are these percentages of all cell pairs or cell pairs that were putatively monosynaptic?

The percentages refer to the percentage of putatively connected cell pairs (of a given type) among all cell pairs (of that type) that were tested for the presence of a putative connection (e.g. the percentage of all internal direction to grid cell pairs that satisfied criteria for a putative connection). The observed connectivity rates of 0.1-0.3 % are in line with several previous reports: connectivity rates of 0.2% by (Barthó et al. 2004), connectivity rates of 0.12-0.27% between putative principal cells in medial prefrontal cortex by (Fujisawa et al. 2008), and 0.17% excitatory connections in Allen institute Neuropixels recordings analyzed by (Ren et al. 2022). As requested by the Reviewer, total numbers of putative connections (in addition to percentages) are now reported in lines 188-199 of the main text. We specify that the percentages refer to the total number of possible cell pairs in the recorded sample.

Note that the percentage of putatively connected cell pairs depends on detection thresholds (Extended Data Fig. 7h) as well as several other parameters (reviewed by Stevenson 2023), including recording duration (Schwindel et al. 2014) and distance between cells (connections in MEC-parasubiculum are topographically organized and confined along the dorsoventral axis, not always matching the spatial layout of recording sites). In the present study, we chose to use fairly strict thresholds to avoid false positive connections (at the expense of increasing false negatives). We did so because we are mainly interested in the functional properties of connected pairs rather than the exact rates of connectivity. Nonetheless, we do show that the main results hold across different detection thresholds (Extended Data Fig. 7i).

L200 - 'circular' typo

Thanks, now corrected.

L564-565 - repetition of $(0.01 \text{ rad} < \sigma < 1 \text{ rad})$

Thanks, now corrected.

L724 - I'm confused by the weighting by the correlation value for internal direction estimation. Do you mean that estimates within the theta cycle that have lower correlations to the reference vector are down weighted? Does this apply to negative correlation values and what happens to the estimate if the downweighting isn't applied?

The decoded estimate of internal direction is taken as the “center of mass” of the posterior probability distribution (in the case of Bayesian decoding) or correlation values (in the case of population-vector decoding). While this method is typically used in Bayesian decoding

(e.g. Widloski and Foster 2022) where the posterior probability is positive and sums to one, the Reviewer points out that the presence of negative values in the population vector correlations might cause problems. We have now looked into this issue and find that the presence of negative values does not affect the location of the mean (see below), and so should thus not affect the decoded direction (Figure 6 for Reviewers, below).

Figure 6 for Reviewers. Figure shows a skewed distribution of angular values (sum of two Von Mises distributions with different means and kappa values). The two distributions are identical, with values in the right panel shifted down by 0.5. Note that the two distributions have identical angular means despite presence of negative values in the right panel.

We have added a statement in the Methods to clarify this (line 769-771).

Note that similar results were obtained using Bayesian decoding of internal direction with LMT tuning curves (no negative values) or when internal direction was extracted with dimensionality reduction methods (Extended Data Fig. 4c-d).

L728 - Is this intended to be 2d?

Yes, figure reference now corrected to 2d.

Referee #2 (Remarks to the Author):

The authors describe a new phenomenon of theta-cycle alternating forward-right, forward-left sweeps of the MEC/parasubicular/hippocampal position and direction representation during rodent locomotion. They perform a comprehensive set of analyses to quantify and further explore this phenomenon. For instance they found that grid cell sweeps are aligned across different grid modules, the sweeps look forward within a distance of half of the grid spacing in each module, hippocampal place cell sweeps follow simultaneously measured grid cell sweeps by ~20 millisecond which suggests a grid cell origin for sweeps, and the MEC/parasubicular direction signal is also aligned in direction with the grid sweeps. In addition, cross-correlation analysis suggests a multisynaptic circuit from direction cells to grid cells underlying the shift in internal position representation of grid cells during sweeps, which also constitutes exciting evidence of a mechanism for moving on the continuous grid cell attractor. They also show that sweeps extend to unexplored space such as on the sides of linear tracks, that alternating sweeps occur during REM sleep, and that in SWS trajectories are aligned with the direction signal.

Overall this is an excellent study that describes a fundamental finding of the rodent spatial system and explores this finding in great detail.

I have one major area of comment and it revolves around understanding the analysis on sweeps extending to never visited locations. First, could this analysis be extended to see if the alternating sweeps are present during the first traversal when an animal is running in a novel space?

In Extended Data Fig. 9f of the original manuscript, we showed examples of left-right-alternating sweeps during foraging in a dark open field arena that the rat was exploring for the first time. However, the example was taken from around 2.5 minutes into the session. Since previous studies have found that hippocampal theta sequences may not be present during the very first traversal of a novel linear track (Feng, Silva, and Foster 2015), we agree with the Reviewer that it would be interesting to see whether alternating MEC-parasubiculum sweeps and internal direction signals are present under those conditions, during the very first traversal of novel space. In the revised manuscript, we now show examples from the very beginning of a recording session in a new environment (approximately 7 s into the session, when the animal is certainly navigating into novel space; data from 1 session in 1 animal). The internal direction signal displays striking left-right-alternation already during those first seconds of exploration and we also see examples of left-right-alternating sweeps through unexplored space at this time. This result is now shown in Extended Data Fig. 9k,l and Supplementary Video 4.

In the above experiment, the rat was placed in a novel circular arena in total darkness (first exposure to both arena and recording room). After an hour of exploration in total darkness, the rat was taken out and room lights were turned on, before the rat was placed back in the arena to explore for another hour (see Waaga et al. 2022 for details). Grid cell rate maps appeared stable during the light session but were less clear during the dark session (presumably due to both novelty and darkness). We therefore used rate maps from the light session to decode position during the dark session. We find that for most of the darkness

session, the lowpass-filtered decoded position roughly followed the animal's running trajectory (with noticeable drift as described by Waaga et al. 2022), with left-right-alternating sweeps emanating from the lowpass-filtered trajectory, in synchrony with the left-right alternation of direction signals. During the first seconds of the session, the output of the position decoder was substantially more noisy than later and it was difficult to reconstruct a coherent spatial trajectory even on a slow timescale that averaged across theta cycles (Video 1 for Reviewers). After the rat had spent 5-7 seconds in the environment, however, the position decoding became more consistent and sweeps could be observed (Videos 1 and 2 for Reviewers, see below for legends).

Video 1 for Reviewers: Lowpass-filtered ($\sigma = 200$ ms gaussian smoothing) decoded position from all recorded MEC-parasubiculum cells during the first minute of exploration in the novel environment (based on LMT tuning curves from the succeeding open field session with room lights on). Purple blob shows decoded position with intensity corresponding to posterior probability. The rat's current position (white arrowhead) and running trajectory (grey line) is also plotted. Note position decoding is incoherent for the first 5-7 seconds.

Video 2 for Reviewers: Decoded position and decoded internal direction from all recorded MEC-parasubiculum cells during the first 30 seconds of exploration in the novel open field arena. The lowpass-filtered decoded position is indicated by a green dot and decoded internal direction is shown with green arrow (length proportional to population firing rate). Note that internal direction clearly alternates within a few seconds after the start of the session, while left-right-alternating sweeps can be seen from 6-7 seconds, corresponding to the time where the lowpass-filtered decoding becomes reliable.

The video shows that, after 6-7 s, the rat was obviously still exploring novel parts of the arena and we thus conclude that sweeps can be present on the very first traversal through a novel portion of space. The details of their implementation in a novel environment must remain for future study, however. Although we found examples of left-right-alternating sweeps during the first seconds of behavior, they appeared to be less clear and less prevalent compared to periods later in the session (and they were completely absent for the first 5 seconds). There are several potential explanations for this slight change in the nature of the sweeps, including novelty-induced fluctuations in firing rates, less vigorous behavior (left-right-alternation is more prominent with fast and straight running), and map instability. It is possible that the relative anchoring of grid modules takes some time to settle, which would lead to catastrophic decoding errors in our whole-population decoder because it includes cells from several grid modules and relies on the relative grid anchoring from the light session (which may take time to settle). With the current low sample size (1 rat, 1 session), we thus cannot rule out that the animal needs to visually explore the environment for a few seconds and find its bearings before sweeps emerge. A thorough study with many more animals, sessions and cells is necessary to fully answer this question.

Second, could the authors explain and clarify their LMT (latent manifold tuning) analysis approach more? I can imagine how grid cell firing patterns could be extended in space and be used to infer positions outside of explored regions. However, while I might have missed it, I didn't see any periodic constraint on individual neuron tuning curves, just gaussian and smoothness constraints. The hippocampal place fields that are inferred in Fig4i look like they are extensions of fields that start inside the walls and thus I can understand how these tuning curves are inferred. But some inferred grid tuning curves in Fig4g look like some of their fields might be completely outside the explored regions of the environments. In such a case I would assume periodicity would provide the constraint to help infer those fields. On the other hand is it the case that all inferred grid fields already have some part of the field rising at the edge of the explored environment which would not then require a periodic constraint, and does this relate to their finding that grid cell sweeps essentially always extend less than half of the grid spacing? If so, it would help if the authors explicitly state that there is always some part of the inferred field inside the regions the animal visited, and that the LMT model builds the new fields that are centered outside of the visited regions from those small "foothills" of fields. Also if there is no periodic-like boundary condition in the LMT analysis, the authors should emphasize this point in the main text or methods since I was assuming it was there and other readers might do the same.

We thank the Reviewer for highlighting some important aspects of the LMT method that we should have described more clearly. Regarding constraints on the fitted spatial tuning, there is no periodic 'gridness' baked into the model's assumptions – the only constraints are (1) Gaussian smoothness and (2) sparsity enforced by an L1 regularization term. The absence of intrinsic periodicity in the LMT position model means that the inferred grid-periodic tuning outside the box is non-trivial. We agree that this methodological detail is important for interpreting the fitted tuning, so we have now emphasized this in the relevant Methods section (line 968-969).

The Reviewer also asked whether externally located fields must have edges that fall within the box in order to be successfully reconstructed by the LMT model. The answer, in principle, is again 'no', because the LMT and similar latent-variable models can fully reconstruct a latent neural manifold (and the corresponding single-neuron tuning) without any prior observation of the tuned-to variable. This has been demonstrated with the LMT model (Wu et al. 2017) and with other similar methods (Zhao and Park 2016; Gao et al. 2016).

In our case, instead of inferring the latent variables from an unbiased initial state, we initialized the latent variables with tracked head position and direction. We made this choice because we are examining much finer-timescale dynamics than the studies cited above, which increases the impact of noise and makes convergence on a local minimum highly likely. Initializing the latent variables at positions close to the expected solution greatly increases the chance of well-behaved convergence.

In principle, the fitted latent position might spread far from the initial tracked position, allowing for fields to occur entirely outside the environment. We found this to be relatively uncommon in our data, and there are several possible explanations for this. First, sweep lengths are limited to a fraction of the grid spacing (a result which we verified with the GLM method; hence this is not a peculiarity of the LMT model); therefore, fields can only occur

within this distance of the environment's walls. We believe that this limitation similarly applies to place cells, because place-cell sweep lengths scaled together with the average receptive field size at the corresponding dorsoventral recording location. Second, because of scattering, data points become increasingly sparse at greater distances outside boundaries. With scarce data, tuning at these far-away locations is unlikely to be resolved, even if it were to exist. Third, the temporal smoothness constraint on the latent position penalizes large, sharp deviations, meaning that longer sweeps (which would be needed to reach far outside the environment) would be more strongly penalized, and would therefore be less likely to be resolved.

A second comment relates to the sleep trajectory analysis. Can the sleep trajectories be analyzed with the population vector approach (versus LMT approach) as used for the initial sweep analysis since all reactivated locations have been visited before, and what would the results look like?

In the original manuscript, we used LMT rate maps from an open field session to decode position (Bayesian approach) during sleep recorded on the same day. As requested by the Reviewer, we now show that the results are largely the same if standard rate maps (with respect to the animal's tracked position) and population-vector decoding methods are used instead. A comparison between LMT and population-vector decoding is included in the revised manuscript as Supplementary Video 5, with accompanying quantification across animals in the video legend. Additionally, we have reproduced the main sweep panels from Fig. 5 with the population-vector decoding method and include these in Figure 8 for Reviewers below.

Figure 8 for Reviewers: **a)** Decoded trajectories for 100 example sweeps from one example grid module (as in Fig. 5d), sorted based on whether the previous decoded direction pointed to the right (blue) or left (red). Note that during wake and REM most sweeps are directed forward to the opposite side of the previous sweep (few red sweeps on left side, few blue sweeps on right side) and resemble the results of the LMT approach. **b)** Averaged sweeps across brain states for all grid modules across animals. Sweeps were referenced, rotated and sorted as in the top panel, normalized by the grid module's spacing, and then averaged across all theta cycles. Each pair of red and blue lines corresponds to one module. **c)** Sweeps are aligned with direction signals in all brain states. Panels show distribution of angles between decoded direction and position (sweep) signals during wake, REM sleep and SWS (left to right). Individual modules are plotted as separate lines. Note alignment in all states, like with the LMT approach (Fig. 5e).

As the Reviewer points out, all locations on the grid-cell unit tile are usually covered by the rat's running trajectory during foraging in a 1.5m x 1.5m box. However, although the population-vector approach may be simpler, we have chosen to use the LMT approach in the main section of the manuscript, whereas population vector-based analyses are reported in the legend of Supplementary Video 5, for a few reasons. Firstly, in grid modules with large spacing (1-2 visible grid fields in arena), the grid unit tile extends beyond the box and the rat will not physically cover it. These grid modules are hard to even detect with conventional methods, since the grid pattern only becomes apparent when extending the map beyond the walls of the arena (e.g. as in Extended Data Fig. 9h). Second, although grid patterns are visible in standard rate maps of most grid cells, a subset of grid cells (bursty II) display sweep-distorted grid patterns that are hard to discern in standard rate maps but can be

recovered by the LMT model. LMT rate maps are generally sharper than standard rate maps for most grid cells, since the standard rate maps are smeared out by sweeps to some degree (Extended Data Fig. 5f-g).

Referee #3 (Remarks to the Author):

The manuscript's major finding about the left-right alternation of population vector activity across theta cycles in the entorhinal and related areas is indeed intriguing. Forward sweeps in theta cycle are known. The novel observation that forward-going population vector activity alternates across theta cycles offers new insights into the dynamics of neural representation during theta oscillations. However, there are several critical points that need to be addressed to strengthen the conclusions drawn from these findings.

Single-Cell Analysis of Alternating Theta Sweeps:

Current work: The population vector analysis, while comprehensive, does not specify which individual cells contribute to the observed alternation effect. This leaves room for potential amplification of random effects by population vectors. Suggestion: The authors could conduct an analysis at the level of single cells. By identifying whether individual cells exhibit systematic phase-precessed firing patterns that align with the observed population-level alternation, they can provide stronger evidence for the phenomenon. Techniques like spike-phase coupling analysis or single-cell autocorrelation within theta cycles could be insightful.

We agree with the Reviewer that the proposed analyses would strengthen the link between single-cell and population analyses. In response to the Reviewer's suggestions, we have added new panels to Extended Data Fig. 5, referenced in lines 93-95 in the manuscript, in which we demonstrate the relationship between population-level sweeps and theta sequences, and theta phase precession and theta cycle skipping in individual grid cells. We show that grid cells with firing fields progressively further away from the animal are activated in sequence during the course of the theta cycle, in coordination with the trajectory of the sweep (Extended Data Fig. 5i, top). The grid cells thus display phase precession along the "sweep axis" (Extended Data Fig. 5i, bottom), which alternates from side to side across theta cycles. Phase precession is visible also when the in-field activity is projected onto the running direction of the animal (Extended Data Fig. 5k), mirroring the results from decades of studies where activity is decoded only with reference to visited positions in linear environments. Finally, in agreement with these experimental observations, we demonstrate the emergence of phase precession in simulations of grid cells that are tuned to a sweeping position signal (Extended Data Fig. 5m), but not in grid cells that are tuned to true position (with no sweeps).

The link between sweeps and sequences at the population level and theta phase modulation in individual cells is apparent also from several findings reported in the original version of the manuscript. We showed, at the population level, that subclasses of grid cells, identified by their bursting properties, are differentially involved in the expression of sweeps (Extended Data Fig. 5) and that sweeps could be decoded from bursty grid cells but not from non-bursty grid cells (Extended Data Fig. 5c). At the single-cell level, we found that bursty grid cells were theta phase modulated (preferentially active during the second half of theta cycles) and often displayed theta cycle skipping, while non-bursty grid cells showed only weak theta phase modulation and preferred the opposite phase of the theta cycle (Extended Data Fig. 5a,d). We also showed an example bursty grid cell that displayed out-of-field firing at the end of the theta cycle when the sweep signal was far away from the animal (Extended

Data Fig. 5e), which further hints at the link between phase precession and sweeps. Our interpretation of these findings, taken together, is that sweeps are expressed most strongly in bursty grid cells and that, in individual bursty grid cells, sweep expression is directly related to theta phase precession and theta cycle skipping (see our reply to the second reviewer point for discussion of theta skipping).

However, as the Reviewer points out, the link between sweeps, on one hand, and phase precession and cycle skipping, on other hand, was not sufficiently emphasized in the original manuscript. In the revised manuscript, we have added figure panels and analyses to address the link better. First, while the path plot with spike positions colored by theta phase in the original manuscript (Extended Data Fig. 5e) indicates the presence of 2D phase precession, we have now provided more direct demonstrations of phase precession plotted in the same way as in conventional 1D theta phase precession plots (in-field position vs. theta phase: Extended Data Fig. 5i,k). Specifically, we show that bursty grid cells display phase precession along the axis of the sweep (which alternates from side to side across theta cycles), while non-bursty grid cells do not (Extended Data Fig. 5j).

Through the new simulations, we also provide a potential explanation for how population-level sweeps can give rise to single-cell phase precession. In Extended Data Fig. 5g of the initial submission, we showed that bursty grid cells were more sharply tuned to the sweeping position signal decoded from population activity compared to the tracked head position of the animal, while non-bursty grid cells were more strongly tuned to tracked head position. In the revised manuscript, we have added simulations that demonstrate the emergence of phase precession in grid cells that are tuned to a sweeping position signal, but not in grid cells that are simply tuned to true position (Extended Data Fig. 5m).

We hope that these new analyses address the requests of the Reviewer.

Relation Between Theta Skipping and Theta Sweeps:

Current Explanation: The authors suggest that theta skipping might be related to the observed alternating theta sweeps, but this connection seems speculative and unclear.

Clarification Needed: The authors could provide details of how theta cycle skipping could lead to systematic sweeping. Specifically, they should investigate if cells that participate in theta skipping also contribute consistently to one side of the alternation. Evidence from spike-timing and phase-precession analysis could support this hypothesis.

We agree with the Reviewer that the relationship between theta cycle skipping in single cells and population-level left-right-alternating sweeps and internal direction signals could be demonstrated more clearly.

In the revised manuscript, as suggested by the Reviewer, we show in the recorded data that theta-skipping cells fire preferentially on one side of the head axis. An example is shown for internal direction cells in a new figure panel in Extended Data Fig. 6g. Here the animal runs from west to east, such that one cell (the left one) fires with internal direction pointing to the left side of the trajectory whereas the other cell (the right one) fires when internal

direction points to the right, on alternating theta cycles. Note that which cells are activated depends on the animal's running trajectory. If the animal's path was from south to north, for example, the left cell would fire on alternating cycles on the right side of the animal, whereas the right cell would be mostly silent. A similar relationship between side of sweeps and theta skipping applies for grid cells, as shown in Extended Data Fig. 5l, where the running path is to the right of the grid field and the cell therefore sweeps through the field only on every second theta cycle.

As further support for the link between left-right alternating sweeps and theta skipping, we have added simulations in which we show that theta skipping emerges, along with phase precession, in individual grid cells when the grid-cell population exhibits left-right-alternating sweeps, but not when the population consists of non-sweeping grid cells. Similarly, we show that theta skipping emerges in simulations of internal direction cells when the population direction signal displays left-right-alternation, but not when it shifts randomly between left and right. The results of these simulations are shown in Extended Data Fig. 5m and 6h-i (referenced in lines 173-174 in the manuscript). The simulations suggest that theta skipping emerges simply because cells are activated only (or mostly) on one side of the animal's head axis.

Finally, we support the link between left-right alternation and theta skipping by showing, in internal direction cells, that it is possible to predict theta cycle skipping based on the cells' tuning curves and decoded internal direction. The prediction-based spike trains displayed theta cycle skipping with autocorrelation plots very similar to those observed from the recorded data, with strong peaks on every second theta cycle. Importantly, crosscorrelograms of recorded and predicted spike trains also had strong peaks every second theta cycle, implying that the predicted activity alternated in synchrony with the recorded activity. Such "co-skipping" between predicted and recorded spikes was present in the majority of theta skipping directional cells (1,768/2,253 or 78% of total). These results indicate that a large fraction of the observed theta skipping in internal direction cells can be explained by the left-right-alternating dynamics of the internal direction population signal. The analyses are shown in Extended Data Fig. 6h and further described in the legend of that figure. The results are referenced in line 173-174 of the main part of the manuscript.

Based on these converging observations and simulations, we propose that theta cycle skipping in both grid cells and internal direction cells is caused by left-right-alternation of their respective population signals (sweeps and internal direction).

Mechanism of Theta Sweep Phenomenon: Current Mechanism (Figure 3):

The proposed monosynaptic connections between subgroups of neurons as a mechanism for theta sweeps raises concerns due to the expected synaptic delays.

Mechanism Feasibility: The authors need to provide evidence or simulations demonstrating that presynaptic spikes can reliably induce postsynaptic spikes within the tight time frame of 2 milliseconds, considering the axonal, dendritic delays, and membrane time constants.

We used cross-correlations of firing rates to identify putative monosynaptic connections. The Reviewer asks whether it is biologically possible for postsynaptic spikes to be triggered within only 2 ms after a presynaptic spike. Setting up new experiments to address these questions in this paper is not feasible (which we have discussed with the Editors); however, the length of biologically plausible intervals has been addressed with quite sophisticated experiments and analyses in several published works, with converging results. We now refer to those studies in the manuscript: ‘ground-truth’ juxtacellular stimulation and recording studies in the hippocampus (English et al. 2017), and paired patch-clamp recordings from MEC-parasubiculum (Winterer et al. 2017; Sammons, Tzilivaki, and Schmitz 2022), all suggesting that the reported 2-ms latencies are biologically plausible.

Particularly relevant among the above works is the study by English et al., where the authors paired juxtacellular stimulation and recording from CA1 pyramidal (PYR) neurons with extracellular recordings from inhibitory interneurons (INT). This preparation allowed the authors to evoke spikes in single PYR neurons and identify INT neurons that fired reliably shortly after the evoked PYR spikes. Importantly, PYR-INT pairs with precise spike-timing after stimulation displayed similarly sharp spike-timing during spontaneous activity. With this method, the authors found synaptic connections with typical postsynaptic spike latencies of 0.8-3 ms with respect to the presynaptic spikes (evoked or spontaneous). The authors also found that PYR-INT latencies increased with anatomical distance, but even at intersomatic distances of 500 μm , the latencies were $\sim 2\text{ms}$ (Fig. 3 of English et al.).

These conclusions are further supported by paired patch-clamp experiments from the Schmitz lab showing that pyramidal-pyramidal connections in parasubiculum (Sammons et al. 2022, Fig. 2) and pyramidal-stellate and stellate-stellate connections in MEC (Winterer et al. 2017, Fig. 3) have latencies of 1-1.5 ms and rise times of ~ 2 ms. While crosscorrelogram shapes are influenced by synaptic and cell-intrinsic parameters (E. E. Fetz and Gustafsson 1983; Cope, Fetz, and Matsumura 1987; E. Fetz, Toyama, and Smith 1991) – such as peak conductance, decay time and reversal potential - as well as background activity and connectivity, as explored in simulations by Ostojic et al. (Ostojic, Brunel, and Hakim 2009), we find it reasonable based on the published work to assume that those shape variations are small and that ~ 2 ms is a fair estimate of the spike-time latency of monosynaptic activation in MEC-parasubiculum.

Thus, in agreement with the Editors, we have chosen not to add new data or simulations but instead we are now citing the work justifying the choice of 2 ms delays. A brief sentence with reference to English et al, Sammons et al and Winterer et al has been added to the Methods section (line 872-873).

Allocentric vs. Head-Centered Coordinate System:

Current Observation: The medial entorhinal cortex is known for its allocentric spatial mapping, yet the observed sweeps are head-centered.

Reconciliation of Observations: The authors could discuss potential mechanisms by which these head-centered sweeps integrate into the broader allocentric map of space. One possibility is that head-centered sweeps could contribute to updating the allocentric map based on directional movement or heading. Including a theoretical or computational framework could help reconcile these observations.

As the Reviewer points out, sweeps display an interesting duality in its allocentric and head-centered properties. While sweeps traverse locations in the allocentric entorhinal-hippocampal map, they also have a fixed geometric relationship with respect to the animal's head, pointing to sweeps as a possible bridge between head-centered signals and allocentric spatial representations.

How such a bridge function could be implemented remains to be determined, however. It is widely believed that animals are capable of updating their allocentric self-location estimate by integrating self-motion signals (most of which are based on head-centered sensory inputs). Grid cells have been proposed to play a pivotal role in this operation by either integrating velocity input signals (Fuhs and Touretzky 2006; McNaughton et al. 2006). However, an unresolved problem with this idea is that path integration relies on velocity input (i.e. movement direction and speed) whereas the directional input signals described until now are more correlated with the animal's head direction than its movement direction (Sargolini et al. 2006; Raudies et al. 2015). This could result in catastrophic path integration errors when head direction does not correspond to movement direction (Raudies et al. 2015). One possibility is that left-right-alternating sweeps provide a mechanism for integrating the animal's path of movement, perhaps by averaging across many sweeps. This would be an interesting topic to explore in future modelling work along with Neuropixels recordings and fine-grain behavioral analysis. However, given its speculative nature and the amount of analysis required to address it, we concluded, after discussing with the Editors, that it would be beyond the scope of the present paper to try to develop a theoretical framework for the coordination of heading direction and allocentric map, and to relate our data to such a framework. However, the Reviewer's point is a good one and we have instead added a sentence in the second paragraph of the Discussion to air the possibility that sweeps can serve as the basis for a movement direction signal. The Raudies-Hasselmo paper is cited to direct readers to the frequent non-alignment of heading and head direction.

Robustness of Theta Sweep Analysis:

Current Analysis: The significance of sweeps is assessed based on triplets of theta cycles.

Suggestion for Robustness Testing: To demonstrate that the alternation is a robust and continuous process, the authors could extend their analysis to include both smaller (2 cycles) and larger numbers of cycles. This could help confirm that the observed alternation is not only for the chosen cycle window and is indeed a persistent feature of the neural activity.

We quantified left-right-alternation by detecting triplets of theta cycles with left-right-left or right-left-right alternation. We agree with the Reviewer that it would be interesting to see

how robust the alternation is beyond windows of three theta cycles. To some extent, persistent alternation can be seen in the original manuscript in the form of autocorrelograms extending to ± 5 theta cycles (e.g. Fig. 1c, 2e) and in examples that extend for up to 24 theta cycles (e.g. Fig. 2c; see also Fig. 9 for Reviewers below). However, to fully answer the Reviewer's question, we have now repeated the alternation analysis over a range of window lengths and show extended periods of alternation across animals. The new findings have been added in Extended Data Fig. 3i and are also reported in the main text, lines 67-68.

In Extended Data Fig. 3i, we now show the percentage of theta-cycle windows with persistent alternation of internal direction or sweep direction, with window lengths spanning 3-20 theta cycles. We compared the measured alternation rates with shuffled controls where head-centered directions were randomly shuffled. We show that both internal direction and sweep direction display robust alternation over extended periods of time, with rates that exceed chance even over 20-cycle windows. We also measured how frequently extended bouts of left-right-alternation occurred, defining an alternation bout as a block of successive theta cycles with consistent left-right-alternation (example bout in Fig. 9 for Reviewers below). The number of alternation bouts that exceeded 10 theta cycles was higher than chance for both internal direction (25.1 ± 2.0 extended bouts per 1000 theta cycles, mean \pm s.e.m. across 16 rats; shuffled: 3.6 ± 0.2 bouts/1000 theta cycles) and sweep direction (6.7 ± 1.5 bouts/1000 theta cycles; shuffled: 0.4 ± 0.1 bouts/1000 theta cycles). For reference, 25 extended bouts of ≥ 10 theta cycles sum to ≥ 250 theta cycles out of 1,000 theta cycles, implying that a quarter of the time was spent in an extended bout of alternation. In general, the estimation of internal direction is more precise and reliable than sweep direction, since it requires fewer cells and is more straightforward to measure.

Figure 9 for Reviewers: Example of an extended alternation bout with consistent left-right-alternation. Note that alternation is present also before the alternation bout, but the left-right pattern is interrupted by a single pair of right-right cycles.

Conclusion: The findings reported in the manuscript offer a fascinating glimpse into the dynamics of neural activity during theta oscillations. However, addressing the points mentioned above through additional analyses and more detailed explanations will significantly strengthen the conclusions and provide a clearer understanding of the underlying mechanisms. This will help reconcile the novel observations with existing knowledge about entorhinal cortex function and theta oscillations.

Thanks – we have added additional analyses to address the points raised by the Reviewer, hoping that the conclusions are now fully justified.

Referee #3 (Remarks on code availability):

The data and codes will be made available prior to publication. This is OK.

References

- Barthó, Peter, Hajime Hirase, Lenaïc Monconduit, Michael Zugaro, Kenneth D. Harris, and György Buzsáki. 2004. 'Characterization of Neocortical Principal Cells and Interneurons by Network Interactions and Extracellular Features'. *Journal of Neurophysiology* 92 (1): 600–608. <https://doi.org/10.1152/jn.01170.2003>.
- Cope, T. C., E. E. Fetz, and M. Matsumura. 1987. 'Cross-Correlation Assessment of Synaptic Strength of Single Ia Fibre Connections with Triceps Surae Motoneurons in Cats'. *The Journal of Physiology* 390 (September):161–88. <https://doi.org/10.1113/jphysiol.1987.sp016692>.
- English, Daniel Fine, Sam McKenzie, Talfan Evans, Kanghwan Kim, Euisik Yoon, and György Buzsáki. 2017. 'Pyramidal Cell-Interneuron Circuit Architecture and Dynamics in Hippocampal Networks'. *Neuron* 96 (2): 505-520.e7. <https://doi.org/10.1016/j.neuron.2017.09.033>.
- Feng, Ting, Delia Silva, and David J. Foster. 2015. 'Dissociation between the Experience-Dependent Development of Hippocampal Theta Sequences and Single-Trial Phase Precession'. *Journal of Neuroscience* 35 (12): 4890–4902. <https://doi.org/10.1523/JNEUROSCI.2614-14.2015>.
- Fetz, E. E., and B. Gustafsson. 1983. 'Relation between Shapes of Post-synaptic Potentials and Changes in Firing Probability of Cat Motoneurons'. <https://doi.org/10.1113/jphysiol.1983.sp014812>.
- Fetz, Eberhard, Keisuke Toyama, and Wade Smith. 1991. 'Synaptic Interactions between Cortical Neurons'. In *Normal and Altered States of Function*, edited by Alan Peters and Edward G. Jones, 1–47. Boston, MA: Springer US. https://doi.org/10.1007/978-1-4615-6622-9_1.
- Frank, Loren M, Emery N Brown, and Matthew Wilson. 2000. 'Trajectory Encoding in the Hippocampus and Entorhinal Cortex'. *Neuron* 27 (1): 169–78. [https://doi.org/10.1016/S0896-6273\(00\)00018-0](https://doi.org/10.1016/S0896-6273(00)00018-0).
- Fuhs, Mark C., and David S. Touretzky. 2006. 'A Spin Glass Model of Path Integration in Rat Medial Entorhinal Cortex'. *Journal of Neuroscience* 26 (16): 4266–76. <https://doi.org/10.1523/JNEUROSCI.4353-05.2006>.
- Fujisawa, Shigeo, Asohan Amarasingham, Matthew T. Harrison, and György Buzsáki. 2008. 'Behavior-Dependent Short-Term Assembly Dynamics in the Medial Prefrontal Cortex'. *Nature Neuroscience* 11 (7): 823–33. <https://doi.org/10.1038/nn.2134>.
- Gao, Yuanjun, Evan Archer, Liam Paninski, and John P. Cunningham. 2016. 'Linear Dynamical Neural Population Models through Nonlinear Embeddings'. arXiv.Org. 26 May 2016. <https://arxiv.org/abs/1605.08454v2>.
- Grion, Natalia, Athena Akrami, Yangfang Zuo, Federico Stella, and Mathew E. Diamond. 2016. 'Coherence between Rat Sensorimotor System and Hippocampus Is Enhanced

- during Tactile Discrimination'. *PLOS Biology* 14 (2): e1002384.
<https://doi.org/10.1371/journal.pbio.1002384>.
- Hernández-Pérez, J Jesús, Keiland W Cooper, and Ehren L Newman. 2020. 'Medial Entorhinal Cortex Activates in a Traveling Wave in the Rat'. Edited by Neil Burgess and Laura L Colgin. *eLife* 9 (February):e52289. <https://doi.org/10.7554/eLife.52289>.
- Johnson, Adam, and A. David Redish. 2007. 'Neural Ensembles in CA3 Transiently Encode Paths Forward of the Animal at a Decision Point'. *Journal of Neuroscience* 27 (45): 12176–89. <https://doi.org/10.1523/JNEUROSCI.3761-07.2007>.
- Joshi, Abhilasha, Eric L. Denovellis, Abhijith Mankili, Yagiz Meneksedag, Thomas J. Davidson, Anna K. Gillespie, Jennifer A. Guidera, Demetris Roumis, and Loren M. Frank. 2023. 'Dynamic Synchronization between Hippocampal Representations and Stepping'. *Nature*, April, 1–7. <https://doi.org/10.1038/s41586-023-05928-6>.
- Kay, Kenneth, Jason E. Chung, Marielena Sosa, Jonathan S. Schor, Mattias P. Karlsson, Margaret C. Larkin, Daniel F. Liu, and Loren M. Frank. 2020. 'Constant Sub-Second Cycling between Representations of Possible Futures in the Hippocampus'. *Cell* 180 (3): 552-567.e25. <https://doi.org/10.1016/j.cell.2020.01.014>.
- Kleinfeld, David, Martin Deschênes, and Nachum Ulanovsky. 2016. 'Whisking, Sniffing, and the Hippocampal θ -Rhythm: A Tale of Two Oscillators'. *PLOS Biology* 14 (2): e1002385. <https://doi.org/10.1371/journal.pbio.1002385>.
- Ledberg, Anders, and David Robbe. 2011. 'Locomotion-Related Oscillatory Body Movements at 6-12 Hz Modulate the Hippocampal Theta Rhythm'. *PloS One* 6 (11): e27575. <https://doi.org/10.1371/journal.pone.0027575>.
- Lubenov, Evgueniy V., and Athanassios G. Siapas. 2009. 'Hippocampal Theta Oscillations Are Travelling Waves'. *Nature* 459 (7246): 534–39. <https://doi.org/10.1038/nature08010>.
- McNaughton, Bruce L., Francesco P. Battaglia, Ole Jensen, Edvard I. Moser, and May-Britt Moser. 2006. 'Path Integration and the Neural Basis of the "Cognitive Map"'. *Nature Reviews Neuroscience* 7 (8): 663–78. <https://doi.org/10.1038/nrn1932>.
- Ostojic, Srdjan, Nicolas Brunel, and Vincent Hakim. 2009. 'How Connectivity, Background Activity, and Synaptic Properties Shape the Cross-Correlation between Spike Trains'. *Journal of Neuroscience* 29 (33): 10234–53. <https://doi.org/10.1523/JNEUROSCI.1275-09.2009>.
- Patel, Jagdish, Shigeyoshi Fujisawa, Antal Berényi, Sébastien Royer, and György Buzsáki. 2012. 'Traveling Theta Waves along the Entire Septotemporal Axis of the Hippocampus'. *Neuron* 75 (3): 410–17. <https://doi.org/10.1016/j.neuron.2012.07.015>.
- Raudies, Florian, Mark P. Brandon, G. William Chapman, and Michael E. Hasselmo. 2015. 'Head Direction Is Coded More Strongly than Movement Direction in a Population of Entorhinal Neurons'. *Brain Research* 1621 (September):355–67. <https://doi.org/10.1016/j.brainres.2014.10.053>.
- Ren, Naixin, Ganchao Wei, Abed Ghanbari, and Ian H. Stevenson. 2022. 'Predictable Fluctuations in Excitatory Synaptic Strength Due to Natural Variation in Presynaptic Firing Rate'. *The Journal of Neuroscience* 42 (46): 8608–20. <https://doi.org/10.1523/JNEUROSCI.0808-22.2022>.
- Sammons, Rosanna P, Alexandra Tzilivaki, and Dietmar Schmitz. 2022. 'Local Microcircuitry of PaS Shows Distinct and Common Features of Excitatory and Inhibitory Connectivity'. *Cerebral Cortex* 32 (1): 76–92. <https://doi.org/10.1093/cercor/bhab195>.

- Sargolini, Francesca, Marianne Fyhn, Torkel Hafting, Bruce L. McNaughton, Menno P. Witter, May-Britt Moser, and Edvard I. Moser. 2006. 'Conjunctive Representation of Position, Direction, and Velocity in Entorhinal Cortex'. *Science* 312 (5774): 758–62. <https://doi.org/10.1126/science.1125572>.
- Schwindel, C. Daniela, Karim Ali, Bruce L. McNaughton, and Masami Tatsuno. 2014. 'Long-Term Recordings Improve the Detection of Weak Excitatory–Excitatory Connections in Rat Prefrontal Cortex'. *Journal of Neuroscience* 34 (16): 5454–67. <https://doi.org/10.1523/JNEUROSCI.4350-13.2014>.
- Stevenson, Ian H. 2023. 'Circumstantial Evidence and Explanatory Models for Synapses in Large-Scale Spike Recordings'. *Neurons, Behavior, Data Analysis, and Theory*, December. <https://doi.org/10.51628/001c.90831>.
- Waaga, Torgeir, Haggai Agmon, Valentin A. Normand, Anne Nagelhus, Richard J. Gardner, May-Britt Moser, Edvard I. Moser, and Yoram Burak. 2022. 'Grid-Cell Modules Remain Coordinated When Neural Activity Is Dissociated from External Sensory Cues'. *Neuron* 110 (11): 1843-1856.e6. <https://doi.org/10.1016/j.neuron.2022.03.011>.
- Widloski, John, and David J. Foster. 2022. 'Flexible Rerouting of Hippocampal Replay Sequences around Changing Barriers in the Absence of Global Place Field Remapping'. *Neuron* 110 (9): 1547-1558.e8. <https://doi.org/10.1016/j.neuron.2022.02.002>.
- Winterer, Jochen, Nikolaus Maier, Christian Wozny, Prateep Beed, Jörg Breustedt, Roberta Evangelista, Yangfan Peng, Tiziano D'Albis, Richard Kempster, and Dietmar Schmitz. 2017. 'Excitatory Microcircuits within Superficial Layers of the Medial Entorhinal Cortex'. *Cell Reports* 19 (6): 1110–16. <https://doi.org/10.1016/j.celrep.2017.04.041>.
- Wu, Anqi, Nicholas A. Roy, Stephen Keeley, and Jonathan W Pillow. 2017. 'Gaussian Process Based Nonlinear Latent Structure Discovery in Multivariate Spike Train Data'. In *Advances in Neural Information Processing Systems*. Vol. 30. Curran Associates, Inc. https://papers.nips.cc/paper_files/paper/2017/hash/b3b4d2dbedc99fe843fd3dedb02f086f-Abstract.html.
- Zhao, Yuan, and Il Memming Park. 2016. 'Variational Latent Gaussian Process for Recovering Single-Trial Dynamics from Population Spike Trains'. arXiv.Org. 11 April 2016. https://doi.org/10.1162/NECO_a_00953.

Response to final comments from Reviewers and Editors

Your manuscript, "Left-right-alternating theta sweeps in the entorhinal-hippocampal spatial map", has now been seen by our referees, and in the light of their advice I am delighted to say that we can in principle offer to publish it.

Thank you very much for the willingness to publish our paper, and for a constructive and helpful review process. We are delighted!

Reviewer comments

Referee #1 (Remarks to the Author):

The authors have sufficiently addressed my concerns. This is a very cool paper and I appreciate the hard work that went into the original manuscript and the revision. Nice job. I have only a few minor comments.

We thank the Reviewer for their insightful comments during revision and hope the remaining comments have been addressed in the revised manuscript.

I commend the authors for their in depth analysis of the relationship between micro movements and sweeps. A few comments:

Extended Data 2 - can we see the frequency and coherence between theta and lateral head movements as in e and f above.

We have now computed coherence spectra between MEC theta oscillations (derived from multi-unit activity) and lateral head speed, and added the plot to Extended Data Fig. 2f. Note absence of peaks in coherence spectrum also for lateral head movements, suggesting that the two oscillations are not phase-coupled. Power spectra for vertical and lateral head speed was already shown in panel b of the same figure.

Extended Data 2g what is the y-axis for the lateral speed plots?

Vertical scale bar for lateral head speed and a horizontal line indicating zero has now been added to Extended Data Fig. 2g to improve readability.

Extended Data 2h right plot - are these directions rotated 90? It looks so compared to the left plot.

Thanks for pointing this out. We have now checked and the two plots are in fact aligned. Since there is no consistent phase locking across animals, the dots in the right plot are more or less randomly scattered.

I think Figure 4 for reviewers is an important feature of the data. The sweep lags and decoding offsets seem modulated by the shape of the environment/task. I would encourage the authors to include this point and figure in the manuscript.

The two key panels from Figure 4 for Reviewers are now added in Extended Data Figure 3g.

Referee #2 (Remarks to the Author):

The new data showing the alternating sweeps exist at the very beginning of a novel exposure is a valuable addition, and the authors have addressed the rest of my comments. Overall I

believe this work would be a very nice addition to the literature.

We thank the Reviewer for their insightful comments during revision and are glad to hear that the comments have been addressed.

Referee #3 (Remarks to the Author):

The authors have responded to all of my observations in a satisfactory manner.

We thank the Reviewer for their insightful comments during revision and are glad to hear that the comments have been addressed.